# SMART: A Modular Two-Stage Framework for Structural Representation Attribution

## Abstract

Recovering latent signals via representation learning is fundamental to analyzing high-dimensional complex systems. However, existing approaches predominantly focus on static single-environment representations, overlooking mechanism shifts across environments, which are crucial for causal discovery, change point detection, and transfer learning, among others. We propose SMART (Sparse Mechanism Attribution for RepresenTation), a modular two-stage framework for structural representation attribution. Our two-stage framework first performs signal recovery, then applies sparse regularization on structural differences to attribute distributional shifts to specific mechanisms, yielding an attribution that circumvents the notorious identifiability issue. Theoretically, we prove that the estimated row support equals the true support with probability tending to one under some conditions. Supporting this, the estimation errors for structure and representation inherit the convergence rates of the first stage. A consistent information criterion is also introduced to determine latent dimensionality. Furthermore, we develop a one-step estimation method for additive noise representations and extend SMART to multi-environment scenarios with theoretical guarantees. Extensive simulations and applications to two real datasets demonstrate the effectiveness and practical utility of SMART.

## 1 Introduction

High-dimensional data are often driven by low-dimensional latent structures. Distilling valid signals from noise constitutes the foundational step in comprehending complex systems, with prominent methodologies ranging from matrix factorization and factor analysis to modern representation learning techniques like Variational Autoencoders (VAEs) (Mnih & Salakhutdinov, 2007; Kingma & Welling, 2013; Liu et al., 2023).

However, in scientific and engineering domains, the central challenge lies in change. For instance, causal discovery requires identifying mechanisms altered by interventions; change point detection seeks to localize specific components drifting during state transitions; and transfer learning distinguishes invariant knowledge from domain specific variations (Aminikhanghahi & Cook, 2017; Rothenhäusler et al., 2021; Zhu et al., 2023). Essentially, these distinct tasks focus on a common objective: structural attribution. Consequently, the shared imperative extends beyond accurate estimation to precise structural attribution. The goal is to disentangle structural drifts from distributional shifts and to name the features whose loading structures have changed. This row support is the primary target of inference in what follows.

Despite this shared goal, existing approaches encounter a fundamental mismatch. First, methods optimizing reconstruction error or distributional distance detect whether a change occurred but fail to localize it. Second, the identifiability barrier arises from rotational ambiguity in latent variable models (Bai & Li, 2012; Cui & Xu, 2025). Comparing estimators across environments directly results in dense loading differences due to arbitrary rotations, rendering the attribution uninterpretable.

This mismatch also distinguishes mechanism attribution from change detection at the coordinate level. A feature may undergo a genuine structural change in its latent loading while its marginal distribution remains nearly unchanged; conversely, a sparse structural perturbation can propagate through dense latent correlations and create broad distributional changes in the observed coordinates. Thus, directly testing observed variables is not sufficient in general for localizing the mechanisms that changed.

In this paper, we address these challenges by proposing **SMART** (**S**parse **M**echanism **A**ttribution for **R**epresen**T**ation), a modular two-stage framework designed to reframe structural attribution as an identifiability problem after estimation. Adopting a two-stage strategy, SMART first performs signal recovery via representation learning, and subsequently shifts the focus from minimizing reconstruction error to explicitly optimizing the parsimony of structural differences. Rather than accepting the dense discrepancies resulting from separate estimations, our framework seeks a latent alignment that yields the sparsest differential structure. By enforcing a row sparsity constraint on the loading differences, we recover the true, sparse locus of the mechanism shift from dense artifacts. Our main theoretical result is a support selection consistency theorem. The estimated set of changed features equals the true one with probability tending to one, provided the regularization parameter lies in an explicit range and the smallest changed row carries enough signal. The error bounds for the representation and for the loading matrices are supporting results and show that both inherit the convergence rate of the first stage. We also propose a consistent information criterion to determine the latent dimensionality. Furthermore, we develop a one-step algorithm for additive noise models and extend the framework to multi-environment scenarios via a fused group penalty, establishing rigorous theoretical guarantees for both extensions.

The contribution of SMART is therefore not another single-environment signal recovery model, but an inferential layer that turns recovered representations into identifiable structural attributions. In contrast to transfer-learning methods that usually treat differences across environments as nuisance terms to be corrected for, SMART treats these differences as the primary objects of inference and provides support-level identifiability under the rotational equivalence class of latent factor models.

## 2 Related Work

**Representation Learning and Factor Analysis.** Dimensionality reduction via representation learning is fundamental to analyzing high-dimensional data, distilling informative latent variables from observed features (Bengio et al., 2013). Prominent approaches include statistical factor analysis (Fan et al., 2013; Chen et al., 2020; Liu et al., 2023), positing a linear generative process, and AEs or VAEs, capturing complex non-linear dependencies (Kingma & Welling, 2013; Vahdat & Kautz, 2020). While these methods excel in static environments, they face challenges under distribution shifts. Standard multi-group latent factor models typically assume either identical structures across groups or use tensor techniques (Han et al., 2024; Bolivar et al., 2025). However, existing frameworks fail to explicitly model the sparse structural shifts within the latent generative mechanism, limiting their interpretability in dynamic environments.

**Sparse Difference from Transfer Learning.** Our work draws inspiration from transfer learning, where source and target domains are assumed to share a dominant common structure with sparse disparities. Pioneering statistical works leverage this sparsity assumption to transfer knowledge (Bastani, 2021; Tian & Feng, 2023). In the machine learning community, transferability is rigorously analyzed through representation learning, leveraging task diversity to guarantee generalization to target domains (Tripuraneni et al., 2020; Zhu et al., 2023). Crucially, most existing methods assume shifts in downstream task labels. By contrast, SMART addresses the fundamental challenge of mechanism shifts within the loading structures themselves.

**Relation to Identifiable Causal Representation Learning.** Identifiable causal representation learning and SMART place change and invariance in opposite parts of the model (Khemakhem et al., 2020; Li et al., 2023). That line of work holds the observation mechanism $p(x \mid z)$ fixed and identifies latent variables or subspaces from sufficient variation of $p(z \mid u)$, whereas SMART holds the latent coordinates fixed and lets the observation mechanism change, so the two targets are complementary rather than competing. The shared $\mathbf{F}$ is what makes the comparison meaningful, because separate $\mathbf{F}^{(u)}$ would confound a change of coordinates with a change of loadings, whereas a single $\mathbf{F}$ leaves only one global transform and that transform preserves the zero row set. When the environmental change acts mainly on $p(\mathbf{F} \mid u)$, standard identifiable causal representation learning is the natural tool and SMART does not claim to cover that case. Because that literature solves the harder latent identification problem, its guarantees typically require the number of auxiliary values or domains to grow with the latent dimension, while SMART needs only two aligned environments. Our framework is better suited to scenarios where environmental changes affect specific modular mechanisms.

## 3 Methodology

In this section, we describe the two stages of SMART.

### 3.1 Stage I: Signal Recovery via Representation Learning

Let $\mathbf{X} = (x_{ij}) \in \mathbb{R}^{n \times p}$ denote the high-dimensional observation matrix. We assume each entry $x_{ij}$ is drawn from a distribution $f(x_{ij}; m_{ij})$, where $m_{ij}$ represents the parameter of interest. Let $\mathbf{M} = (m_{ij}) \in \mathbb{R}^{n \times p}$ denote the latent parameter matrix. We extract the low-dimensional representation $\mathbf{F} = (\boldsymbol{f}_1, \ldots, \boldsymbol{f}_n)^\top \in \mathbb{R}^{n \times r}$ ($r \ll p$) from $\mathbf{X}$, where $\boldsymbol{f}_i \in \mathbb{R}^r$ denotes the latent vector for the $i$-th observation, and subsequently use $\mathbf{F}$ to recover the underlying parameter matrix $\mathbf{M}$. Formally, this procedure consists of two mappings:

Representation Learning: We first extract latent structures from the observations via an encoding function $h : \mathbb{R}^{n \times p} \to \mathbb{R}^{n \times r}$, yielding the estimated representation

$$\hat{\mathbf{F}} = h(\mathbf{X}).$$

Signal Reconstruction: We then reconstruct the parameter matrix via a decoding function $g : \mathbb{R}^{n \times r} \to \mathbb{R}^{n \times p}$, yielding the estimated signal

$$\tilde{\mathbf{M}} = g(\hat{\mathbf{F}}).$$

We will provide two examples to illustrate this procedure.

*Example* 1 (Additive Noise Models). Consider the additive noise model $\mathbf{X} = \mathbf{M} + \mathcal{E}$, where $\mathcal{E}$ is a noise matrix with $\mathbb{E}[\mathcal{E}] = \mathbf{0}$. Here, the parameter of interest $\mathbf{M}$ corresponds to the expectation $\mathbb{E}[\mathbf{X}]$. Under the low-rank assumption, we estimate $\mathbf{M}$ by solving the constrained optimization problem:

$$\tilde{\mathbf{M}} = \underset{\mathbf{M}:\text{rank}(\mathbf{M}) \leq r}{\arg\min} \|\mathbf{X} - \mathbf{M}\|_F^2.$$

In this setting, the solution corresponds to Principal Component Analysis (PCA) or Singular Value Decomposition (SVD). Specifically, $\hat{\mathbf{F}}$ is derived from the top-$r$ left singular vectors (scaled by singular values).

*Example* 2 (Generalized Linear Models). Assume $x_{ij}$ follows an exponential family distribution with density

$$f(x_{ij}; m_{ij}) = \exp\left[\phi^{-1}\{x_{ij}m_{ij} - \psi(m_{ij})\} + c(x_{ij})\right],$$

where $m_{ij}$ is the natural parameter, $\psi(\cdot)$ and $c(\cdot)$ are prespecified functions. The scale parameter $\phi$ is known. Assuming that $\mathbf{M}$ can be linearly represented by $\mathbf{F}$, i.e., $\mathbf{M} = \mathbf{F}\mathbf{B}^\top$, where $\mathbf{B} \in \mathbb{R}^{p \times r}$ is the loading matrix. Then this is known as Generalized Factor Models (GFM). The estimation of $\mathbf{F}$ and $\mathbf{B}$ is typically achieved by maximizing the log-likelihood via alternating optimization algorithms (Chen et al., 2020; Liu et al., 2023).

Beyond these, our framework naturally accommodates other dimensionality reduction methods. For instance, deep generative models such as VAEs can be employed, where $h(\cdot)$ and $g(\cdot)$ are parameterized by neural networks (encoder and decoder, respectively) to capture complex dependencies in $\mathbf{M}$ (see Girin et al., 2022, and references therein). Moreover, the approach remains robust to incomplete observations. When only $\mathbf{X} \circ \boldsymbol{\Omega}$ is observed, where $\boldsymbol{\Omega}$ denotes a binary mask matrix, variants of the aforementioned methods can be readily adapted to perform effective signal recovery (Gondara & Wang, 2018; Chen & Li, 2022).

### 3.2 Stage II: Sparse Mechanism Attribution

Consider data matrices $\mathbf{X}^{(1)} = (x_{ij}^{(1)})$ and $\mathbf{X}^{(2)} = (x_{ij}^{(2)})$ from two environments with parameter matrices $\mathbf{M}^{(1)} = (m_{ij}^{(1)})$ and $\mathbf{M}^{(2)} = (m_{ij}^{(2)})$, respectively. To facilitate attribution estimation and enhance interpretability, we postulate that the true parameters share a linear structure induced by a common low-dimensional representation that is invariant across environments. Specifically, we model

$$\mathbf{M}^{(1)} = \mathbf{F}(\mathbf{B}^{(1)})^\top, \quad \mathbf{M}^{(2)} = \mathbf{F}(\mathbf{B}^{(2)})^\top,$$

where $\mathbf{F} = (\boldsymbol{f}_1, \ldots, \boldsymbol{f}_n)^\top \in \mathbb{R}^{n \times r}$ denotes the low-dimensional sample representation, and $\mathbf{B}^{(k)} = (\boldsymbol{b}_1, \ldots, \boldsymbol{b}_p)^\top \in \mathbb{R}^{p \times r}(k \in \{1, 2\})$ captures the interaction between representations and features in environment $k$. We focus on the shift in interactions by attributing changes to the difference $\Delta\mathbf{B} = \mathbf{B}^{(2)} - \mathbf{B}^{(1)} =$

$(\Delta\boldsymbol{b}_1, \ldots, \Delta\boldsymbol{b}_p)^\top \in \mathbb{R}^{p \times r}$. Assuming $\Delta\mathbf{B}$ is row sparse, identifying its non-zero rows reveals the features undergoing variation across two environments. Let $\mathcal{S}^* = \left\{ j : \left\| \Delta\boldsymbol{b}_j^* \right\|_2 \neq 0 \right\}$ denote the support of the structural differences, and let $s = |\mathcal{S}^*|$ be its cardinality, where a superscript $*$ marks a true parameter value.

Two modeling choices define the scope of this formulation. First, SMART targets matrix data whose samples are paired or aligned across environments, so that row $i$ refers to the same unit in $\mathbf{X}^{(1)}$ and in $\mathbf{X}^{(2)}$. Under alignment, $\mathbf{F}$ carries the sample level heterogeneity common to both environments and $\mathbf{B}^{(k)}$ the environment specific association between that representation and the features. Second, the linear decomposition acts on the Stage I parameter matrix rather than on the raw observations, and it keeps every row of $\mathbf{B}^{(k)}$ tied to a single feature, which is what makes a nonzero row of $\Delta\mathbf{B}$ readable as one feature level mechanism.

The target $\mathcal{S}^*$ is a support of mechanism changes rather than a support of marginal distributional changes in the raw variables. In an additive factor model, the covariance difference contains terms such as $\mathbf{B}^{(1)}\Sigma_{\boldsymbol{f}}(\Delta\mathbf{B})^\top$ and $\Delta\mathbf{B}\Sigma_{\boldsymbol{f}}(\mathbf{B}^{(1)})^\top$; when $\mathbf{B}^{(1)}$ is dense, even a sparse $\Delta\mathbf{B}$ can induce dense changes in observed covariances. Conversely, as illustrated in Appendix D.7, a nonzero row of $\Delta\mathbf{B}$ can be constructed so that the corresponding observed marginal distribution remains unchanged. This is why SMART uses the shared latent representation as an anchor for structural attribution instead of relying on separate coordinate-wise distribution tests.

The proposed framework accommodates diverse scenarios characterized by mechanism shifts. Representative applications include, among others, causal inference for event shocks, where determining the sparse support of $\Delta\mathbf{B}$ reveals sensitive features; transfer learning, which exploits shared latent structures to isolate domain specific variations; and change point detection, which facilitates the simultaneous localization of temporal break points and structural coordinates.

To estimate the model parameters, we adopt a two-stage strategy. In the first stage, we obtain preliminary estimates $\tilde{\mathbf{M}}^{(k)}$ for each environment. This preliminary estimation proceeds without imposing a low-rank constraint, thereby enhancing flexibility in handling complex data structures. Subsequently, we construct the objective function for the second stage by aggregating the low-rank reconstruction losses of $\tilde{\mathbf{M}}^{(k)}$ and imposing a sparsity penalty on the environmental shift matrix

$$\mathcal{Q}(\mathbf{F}, \mathbf{B}^{(1)}, \mathbf{B}^{(2)}; \tilde{\mathbf{M}}^{(1)}, \tilde{\mathbf{M}}^{(2)}) = \left\| \tilde{\mathbf{M}}^{(1)} - \mathbf{F}(\mathbf{B}^{(1)})^\top \right\|_F^2 + \left\| \tilde{\mathbf{M}}^{(2)} - \mathbf{F}(\mathbf{B}^{(2)})^\top \right\|_F^2 + \lambda \left\| \mathbf{B}^{(2)} - \mathbf{B}^{(1)} \right\|_{2,1},$$

where the $\|\cdot\|_{2,1}$ norm of a matrix denotes the sum of the $\ell_2$-norms of its rows. Recalling the shift $\Delta\mathbf{B} = \mathbf{B}^{(2)} - \mathbf{B}^{(1)}$, we reformulate the optimization problem as

$$\mathcal{Q}(\mathbf{F}, \mathbf{B}^{(1)}, \Delta\mathbf{B}; \tilde{\mathbf{M}}^{(1)}, \tilde{\mathbf{M}}^{(2)}) = \left\| \tilde{\mathbf{M}}^{(1)} - \mathbf{F}(\mathbf{B}^{(1)})^\top \right\|_F^2 + \left\| \tilde{\mathbf{M}}^{(2)} - \mathbf{F}(\mathbf{B}^{(1)} + \Delta\mathbf{B})^\top \right\|_F^2 + \lambda \|\Delta\mathbf{B}\|_{2,1} \tag{1}$$

with the estimator given by

$$\left( \hat{\mathbf{F}}, \hat{\mathbf{B}}^{(1)}, \hat{\Delta\mathbf{B}} \right) = \underset{\mathbf{F}, \mathbf{B}^{(1)}, \Delta\mathbf{B}}{\arg\min} \mathcal{Q}(\mathbf{F}, \mathbf{B}^{(1)}, \Delta\mathbf{B}; \tilde{\mathbf{M}}^{(1)}, \tilde{\mathbf{M}}^{(2)}).$$

*Remark* 3.1. The parameter set $\{\mathbf{F}, \mathbf{B}^{(1)}, \mathbf{B}^{(2)}, \Delta\mathbf{B}\}$ is not uniquely identifiable. Specifically, for any invertible matrix $\mathbf{Q} \in \mathbb{R}^{r \times r}$, the transformed set $\{\mathbf{FQ}, \mathbf{B}^{(1)}(\mathbf{Q}^{-1})^\top, \mathbf{B}^{(2)}(\mathbf{Q}^{-1})^\top, \Delta\mathbf{B}(\mathbf{Q}^{-1})^\top\}$ constitutes an equivalent solution. Imposing the constraint $n^{-1}\mathbf{F}^\top\mathbf{F} = \mathbf{I}_r$ can restrict $\mathbf{Q}$ to an orthogonal matrix. While the individual embeddings and loading matrices are not identifiable, their column spaces are estimable. Moreover, the row sparsity support of $\Delta\mathbf{B}$ is invariant under orthogonal rotation. Consequently, the identified key features remain unique and identifiable despite the rotational ambiguity.

To clarify the last statement, the ambiguity in Theorem 3.3 concerns the numerical values of $\Delta\mathbf{B}$, not its row support. For any orthogonal matrix $\mathbf{O}$ and any row vector $\Delta\boldsymbol{b}_j^*$, $\|\Delta\boldsymbol{b}_j^*\mathbf{O}\|_2 = \|\Delta\boldsymbol{b}_j^*\|_2$; hence $\Delta\boldsymbol{b}_j^* = \mathbf{0}$ if and only if $\Delta\boldsymbol{b}_j^*\mathbf{O} = \mathbf{0}$. The support $\mathcal{S}^*$ is therefore identifiable without choosing a particular coordinate system in the latent space. If a specific Stage-I factor model imposes stronger normalization and sign conventions, the remaining rotation may be further fixed (Bai & Li, 2012; Liu et al., 2023); SMART does not require this stronger identifiability of coefficient values because its inferential target is the support of the structural shift.

### 3.2.1 Estimation

Given the non-convex nature of the objective function, we employ an alternating minimization strategy to solve for the parameters. Following Chen et al. (2020), the estimation proceeds over a bounded feasible region

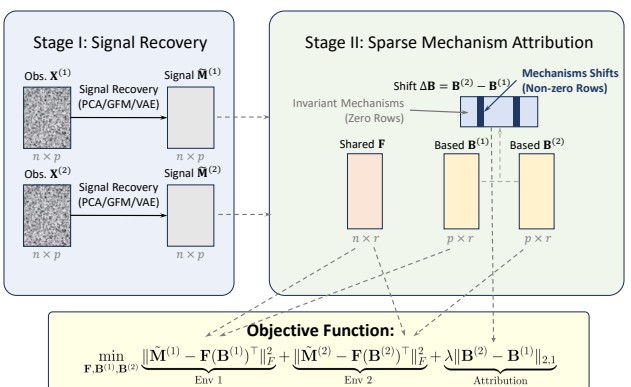

Figure 1: **Overview of the SMART framework.** The proposed method operates in two stages: Stage I recovers the denoised signals $\tilde{\mathbf{M}}^{(k)}$ from high-dimensional observations $\mathbf{X}^{(k)}$ using standard representation learning methods (e.g., PCA, GFM, or VAE). Stage II performs structural disentanglement by minimizing the reconstruction error regularized by a row-sparse penalty on the shift matrix $\Delta\mathbf{B}$. This formulation explicitly identifies the locus of mechanism shifts (non-zero rows) while aligning invariant structures (zero rows). defined as

$$\mathcal{F}(C) = \left\{ \mathbf{F}, \mathbf{B}^{(k)} : \{\|\boldsymbol{f}_i\|_2, \|\boldsymbol{b}_j\|_2\} \le C, n^{-1}\mathbf{F}^\top\mathbf{F} = \mathbf{I}_r \right\},$$

where $C$ is a sufficiently large constant. The boundedness constraint ensures solution stability and facilitates theoretical analysis.

**Step 0: Initialization.** We initialize the parameters by performing SVD on the concatenated matrix $(\tilde{\mathbf{M}}^{(1)}, \tilde{\mathbf{M}}^{(2)}) \in \mathbb{R}^{n \times 2p}$. We set the initial representation $\mathbf{F}^{[0]}$ as the leading $r$ left singular vectors scaled by $\sqrt{n}$, and initialize the structural shift matrix as $\Delta\mathbf{B}^{[0]} = \mathbf{0}$.

**Step 1: Update $\mathbf{B}^{(1)}$.** With $\mathbf{F}^{[t-1]}$ and $\Delta\mathbf{B}^{[t-1]}$ fixed, we minimize the Equation (1) with respect to $\mathbf{B}^{(1)}$. Omitting terms independent of $\mathbf{B}^{(1)}$, the problem becomes

$$\underset{\mathbf{B}^{(1)}}{\arg\min} \left\{ \left\| \tilde{\mathbf{M}}^{(1)} - \mathbf{F}^{[t-1]}(\mathbf{B}^{(1)})^\top \right\|_F^2 + \left\| \tilde{\mathbf{M}}^{(2)} - \mathbf{F}^{[t-1]}(\Delta\mathbf{B}^{[t-1]})^\top - \mathbf{F}^{[t-1]}(\mathbf{B}^{(1)})^\top \right\|_F^2 \right\},$$

yielding the closed form solution:

$$\breve{\mathbf{B}}^{(1)[t]} = \frac{1}{2}\left( \tilde{\mathbf{M}}^{(1)} + \tilde{\mathbf{M}}^{(2)} \right)^\top \mathbf{F}^{[t-1]} \left( (\mathbf{F}^{[t-1]})^\top \mathbf{F}^{[t-1]} \right)^{-1} - \frac{1}{2}\Delta\mathbf{B}^{[t-1]}.$$

**Step 2: Update $\Delta\mathbf{B}$.** Given $\mathbf{F}^{[t-1]}$ and $\mathbf{B}^{(1)[t]}$, the optimization for $\Delta\mathbf{B}$ decomposes into $p$ independent component-wise subproblems. Rewriting Equation (1), we have

$$\breve{\Delta\boldsymbol{b}}_j^{[t]} = \underset{\Delta\boldsymbol{b}_j}{\arg\min} \left( \sum_{i=1}^n \left[ \left\{ \tilde{m}_{ij}^{(1)} - (\boldsymbol{b}_j^{(1)[t]})^\top \boldsymbol{f}_i^{[t-1]} \right\}^2 + \left\{ \tilde{m}_{ij}^{(2)} - (\boldsymbol{b}_j^{(1)[t]})^\top \boldsymbol{f}_i^{[t-1]} - \Delta\boldsymbol{b}_j^\top \boldsymbol{f}_i^{[t-1]} \right\}^2 \right] + \lambda\|\Delta\boldsymbol{b}_j\|_2 \right).$$

This formulation corresponds to a group Lasso problem; refer to Hastie et al. (2015) for detailed solving algorithms.

**Step 3: Update $\mathbf{F}$.** Fixing $\mathbf{B}^{(1)[t]}$ and $\Delta\mathbf{B}^{[t]}$, we solve for $\mathbf{F}$ by minimizing the low-rank reconstruction loss:

$$\breve{\mathbf{F}}^{[t]} = \underset{\mathbf{F}}{\arg\min} \left\{ \left\| \tilde{\mathbf{M}}^{(1)} - \mathbf{F}(\mathbf{B}^{(1)[t]})^\top \right\|_F^2 + \left\| \tilde{\mathbf{M}}^{(2)} - \mathbf{F}(\mathbf{B}^{(1)[t]} + \Delta\mathbf{B}^{[t]})^\top \right\|_F^2 \right\}.$$

The closed form update is given by

$$\breve{\mathbf{F}}^{[t]} = \left\{ \tilde{\mathbf{M}}^{(1)}\mathbf{B}^{(1)[t]} + \tilde{\mathbf{M}}^{(2)}\mathbf{B}^{(2)[t]} \right\} \times \left\{ (\mathbf{B}^{(1)[t]})^\top\mathbf{B}^{(1)[t]} + (\mathbf{B}^{(2)[t]})^\top\mathbf{B}^{(2)[t]} \right\}^{-1}.$$

**Step 4: Orthogonalization and Projection.** We first enforce the orthogonality condition via a normalization step. Let $\mathbf{S} = n^{-1}(\breve{\mathbf{F}}^{[t]})^\top \breve{\mathbf{F}}^{[t]}$ with eigendecomposition $\mathbf{S} = \mathbf{U}\boldsymbol{\Lambda}\mathbf{U}^\top$. Define $\mathbf{Q} = \mathbf{U}\boldsymbol{\Lambda}^{-\frac{1}{2}}$ and update the parameters as $\mathbf{F}^{[t]} = \breve{\mathbf{F}}^{[t]}\mathbf{Q}$, $\mathbf{B}^{(1)[t]} = \breve{\mathbf{B}}^{(1)[t]}(\mathbf{Q}^{-1})^\top$, $\Delta\mathbf{B}^{[t]} = \Delta\breve{\mathbf{B}}^{[t]}(\mathbf{Q}^{-1})^\top$. Subsequently, to ensure the solution resides within $\mathcal{F}(C)$, we project the row vectors of the updated matrices. Specifically, any row vector $\boldsymbol{v} \in \{\boldsymbol{f}_i^{[t]}, \boldsymbol{b}_j^{(1)[t]}, (\boldsymbol{b}_j^{(1)[t]} + \Delta\boldsymbol{b}_j^{[t]})\}$ is rescaled to $C\boldsymbol{v}/\|\boldsymbol{v}\|_2$ if $\|\boldsymbol{v}\|_2 > C$; otherwise, it remains unchanged.

**Convergence.** We iterate Steps 1–4 until convergence. The complete procedure is summarized in Algorithm 2 in Appendix A.

### 3.3 Theoretical Property for SMART

For notational clarity, we use the superscript "$*$" to denote the true parameter values hereafter.

*Condition* (C1). $\|\boldsymbol{f}_i^*\|_2 \le C$, $\left\|\boldsymbol{b}_j^{(k)*}\right\|_2 \le C$, and $n^{-1}(\mathbf{F}^*)^\top \mathbf{F}^* = \mathbf{I}_r$.

*Condition* (C2). $\sigma_1\left(\mathbf{M}^{(1)*}\right) \asymp \sigma_{r^*}\left(\mathbf{M}^{(1)*}\right)$ and $\sigma_{r^*}\left(\mathbf{B}^{(k)*}\right) \asymp \sigma_{\mathbf{B}}$.

Condition (C1) requires the true parameters to be bounded, a standard assumption in theoretical analysis (e.g., Zhang et al., 2020; Chen & Li, 2022). Condition (C2) stipulates that the non-zero singular values of the true parameter matrices share the same order of magnitude, consistent with Bai & Ng (2002) and Fan et al. (2022).

We first establish the estimation error bound for the parameter matrices in the following theorem.

**Theorem 3.2.** *Suppose that the estimation error in the first stage satisfies $\left\|\tilde{\mathbf{M}}^{(k)} - \mathbf{M}^{(k)*}\right\|_F \lesssim err$ with probability tending to one for $k \in \{1, 2\}$. Under Condition (C1) and assuming $r \ge r^*$, it holds that*

$$\left\|\mathbf{M}^{(k)} - \hat{\mathbf{F}}\left(\hat{\mathbf{B}}^{(k)}\right)^\top\right\|_F \lesssim err + \sqrt{s}\lambda$$

*for $k \in \{1, 2\}$ with probability tending to one.*

Theorem 3.2 quantifies the overall estimation error, which is dominated by two components: the error inherited from the preliminary stage and the error induced by the regularization parameter $\lambda$ and sparsity level $s$. The second component is the price of regularization, and it is not a cost that every support recovery method must pay. When $s\lambda \ll n \vee p$, the term $\sqrt{s}\lambda$ is of lower order than the inherited Stage I error under the generalized factor model rates of Corollary 3.4, so SMART matches the first order reconstruction rate of the unpenalized truncated singular value decomposition while gaining the localization that the latter cannot provide. We do not claim an advantage over the unpenalized benchmark in pure reconstruction. Accordingly, the experiments report reconstruction accuracy and support recovery as separate criteria rather than as a single score.

Building on this result, we further characterize the estimation accuracy of the latent representation $\mathbf{F}$, the loading matrices $\mathbf{B}^{(k)}$, and the attribution matrix $\Delta\mathbf{B}$.

**Theorem 3.3.** *Suppose that the estimation error in the first stage satisfies $\left\|\tilde{\mathbf{M}}^{(k)} - \mathbf{M}^{(k)*}\right\|_F \lesssim err$ with probability tending to one for $k \in \{1, 2\}$. Under Conditions (C1) and (C2), and assuming $r = r^*$, there exists an orthogonal rotation matrix $\mathbf{O}$ such that the following convergence rates hold with probability tending to one:*

$$\frac{1}{\sqrt{n}}\left\|\hat{\mathbf{F}}\mathbf{O} - \mathbf{F}^*\right\|_F \lesssim \frac{err + \sqrt{s}\lambda}{\sqrt{n}\sigma_{\mathbf{B}}}, \qquad \frac{1}{\sqrt{p}}\left\|\hat{\mathbf{B}}^{(k)}\mathbf{O} - \mathbf{B}^{(k)*}\right\|_F \lesssim \frac{err + \sqrt{s}\lambda}{\sqrt{n}\sigma_{\mathbf{B}}}, \quad k \in \{1, 2\},$$

*and*

$$\frac{1}{\sqrt{s}}\left\|\hat{\Delta\mathbf{B}}\mathbf{O} - \Delta\mathbf{B}^*\right\|_F \lesssim \sqrt{\frac{p}{\sigma_{\mathbf{B}}^2}} \cdot \frac{err + \sqrt{s}\lambda}{\sqrt{ns}}.$$

Theorem 3.3 indicates that a larger $\sigma_{\mathbf{B}}$ leads to improved estimation precision. While standard factor analysis typically assumes a strong signal regime with $\sigma_{\mathbf{B}} \asymp \sqrt{p}$ (Bai & Ng, 2002), our theory accommodates weaker signal regimes (Bai & Ng, 2023). Provided $\sqrt{n}\sigma_{\mathbf{B}} \gg err + \sqrt{s}\lambda$, the consistency of the estimators is guaranteed. The Stage I error $err$ enters every bound above explicitly, and it enters the range for $\lambda$ in Theorem 3.6 through the term $\sqrt{n}\,err/\sigma_{\mathbf{B}}$. The accuracy of SMART therefore inherits the quality of Stage I

in a way that can be read off the statements. Changing the Stage I estimator changes only *err*, so a better recovery method can be substituted without altering the Stage II objective, its algorithm, or the form of these results.

To demonstrate the versatility of our framework, we analyze data from generalized linear distributions, which encompass diverse distributions including Gaussian, Poisson, and Binary. Adopting the generative mechanism described in Example 2 and postulating a low-rank structure for the parameter matrix $\mathbf{M}$, we obtain the stage I estimate $\tilde{\mathbf{M}}$ via the maximum likelihood estimator proposed by Chen et al. (2020). The resulting theoretical guarantees are established in the following corollaries.

**Corollary 3.4.** *Under the conditions of Theorem 3.2, assume that the observations $x_{ij}$ are independent given the common representation $\boldsymbol{f}_i$, and that the GFM estimator is utilized in the first stage. If $s\lambda \ll n \vee p$, then*

$$\frac{1}{\sqrt{np}} \left\| \mathbf{M}^{(k)*} - \hat{\mathbf{F}} \left( \hat{\mathbf{B}}^{(k)} \right)^\top \right\|_F \lesssim \frac{1}{\sqrt{n} \wedge \sqrt{p}}$$

*with probability tending to one for $k \in \{1, 2\}$.*

**Corollary 3.5.** *Suppose that the conditions of Theorem 3.3 and Corollary 3.4 are satisfied. If we further assume $\sigma_\mathbf{B} \asymp \sqrt{p}$, then the following convergence rates hold with probability tending to one:*

$$\frac{1}{\sqrt{n}} \left\| \hat{\mathbf{F}}\mathbf{O} - \mathbf{F}^* \right\|_F \lesssim \frac{1}{\sqrt{n} \wedge \sqrt{p}}, \qquad \frac{1}{\sqrt{p}} \left\| \hat{\mathbf{B}}^{(k)}\mathbf{O} - \mathbf{B}^{(k)*} \right\|_F \lesssim \frac{1}{\sqrt{n} \wedge \sqrt{p}}, \quad k \in \{1, 2\},$$

*and*

$$\frac{1}{\sqrt{s}} \left\| \hat{\Delta \mathbf{B}}\mathbf{O} - \Delta \mathbf{B}^* \right\|_F \lesssim \frac{\sqrt{n} \vee \sqrt{p}}{\sqrt{ns}}.$$

## 3.4 Support Selection Consistency

The row support of $\Delta \mathbf{B}$ is invariant under the rotational ambiguity of the latent space, so the true support $\mathcal{S}^*$ is well defined without fixing a coordinate system. While Theorem 3.3 controls the estimation error of $\hat{\Delta \mathbf{B}}$ up to rotation, it does not by itself guarantee that the estimated support matches $\mathcal{S}^*$. This subsection closes the gap between rotation invariance of the support and its exact recovery. We write $\beta_{\min} = \min_{j \in \mathcal{S}^*} \|\Delta \boldsymbol{b}_j^*\|_2$ for the minimal signal strength.

The support estimate is read directly from the estimator of Equation (1), with no separate construction. Writing $(\hat{\mathbf{F}}, \hat{\mathbf{B}}^{(1)}, \hat{\Delta \mathbf{B}})$ for the minimizer analyzed in Theorems 3.2 and 3.3, we set $\hat{\mathcal{S}} = \{j : \|\hat{\Delta \boldsymbol{b}}_j\|_2 > 0\}$. This set is well defined because the penalty produces exact zero rows.

The analysis requires control of the Stage I noise after projection onto the latent space. For $k \in \{1, 2\}$ let $\boldsymbol{e}_j^{(k)}$ denote the $j$th column of $\tilde{\mathbf{M}}^{(k)} - \mathbf{M}^{(k)*}$, and set $\boldsymbol{e}_{\Delta, j} = \boldsymbol{e}_j^{(2)} - \boldsymbol{e}_j^{(1)}$.

*Condition* (C3). There exists a sequence $\rho_n > 0$ such that $\max_{k \in \{1,2\}} \max_{1 \le j \le p} \|n^{-1}(\mathbf{F}^*)^\top \boldsymbol{e}_j^{(k)}\|_2 = O_p(\rho_n)$ and $\max_{k \in \{1,2\}} \max_{1 \le j \le p} n^{-1/2}\|\boldsymbol{e}_j^{(k)}\|_2 = O_p(1)$.

By the triangle inequality the same bounds hold for the difference noise, and in particular $\rho_{n,p} := \max_{1 \le j \le p} \|n^{-1}(\mathbf{F}^*)^\top \boldsymbol{e}_{\Delta, j}\|_2 = O_p(\rho_n)$ controls the maximal projection of the noise difference onto the true factor space. The second requirement asks only for stochastic boundedness of the rescaled column norms.

**Theorem 3.6.** *Suppose that the first stage satisfies $\|\tilde{\mathbf{M}}^{(k)} - \mathbf{M}^{(k)*}\|_F \lesssim err$ with probability tending to one for $k \in \{1, 2\}$, where err is a sequence. Under Conditions (C1) to (C3), assume that $r = r^*$ and $\mathcal{S}^* \ne \emptyset$, and let $\lambda$ be a sequence satisfying*

$$\max \left\{ n\rho_n, \ \frac{\sqrt{n}\,err}{\sigma_\mathbf{B}}, \ \frac{ns}{\sigma_\mathbf{B}^2} \right\} \ll \lambda \ll \min \left\{ n\beta_{\min}, \ \frac{n \vee p}{s} \right\}. \tag{2}$$

*Then $\mathbb{P}(\hat{\mathcal{S}} = \mathcal{S}^*) \to 1$ as $n, p \to +\infty$.*

Writing $L_{n,p}$ and $U_{n,p}$ for the two sides of Equation (2), the interval is nonempty if and only if $L_{n,p} = o(U_{n,p})$, in which case $\lambda = \sqrt{L_{n,p}U_{n,p}}$ is admissible; Proposition C.7 in Appendix C makes this precise. The cross validation choice used in the experiments is a practical surrogate.

The window simplifies when the first stage is refined by the entrywise correction of Chen & Li (2024), which upgrades the Frobenius guarantee for Chen et al. (2020) to a maximum norm guarantee.

**Corollary 3.7.** *Under Conditions (C1) and (C2), assume $\sigma_{\mathbf{B}} \asymp \sqrt{p}$, $r = r^*$ and $\mathcal{S}^* \neq \emptyset$. Assume further that the entries are fully observed and independent given the representation, and that the first stage uses the estimator refined by the procedure of Chen & Li (2024), so that under their regularity conditions $\max_{k \in \{1,2\}} \|\tilde{\mathbf{M}}^{(k)} - \mathbf{M}^{(k)*}\|_{\max} = O_p(d_{n,p})$ with $d_{n,p} = \{\log(np)\}^2/\sqrt{n \wedge p} \to 0$. If $\beta_{\min} \gg d_{n,p}$ and $s \ll (n \vee p)/(n\, d_{n,p})$, then the interval $n\, d_{n,p} \ll \lambda \ll \min\{n\beta_{\min}, (n \vee p)/s\}$ is nonempty, and for any $\lambda$ in this range $\mathbb{P}(\hat{\mathcal{S}} = \mathcal{S}^*) \to 1$.*

### 3.5 Determination of Latent Dimensionality

While the preceding analysis presumes a given latent dimensionality $r$, in practice, it is typically unknown. Correctly specifying the latent dimensionality is critical (Bai & Ng, 2002; Chen & Li, 2022); underestimating $r$ incurs signal loss, whereas overestimating it introduces extraneous noise. Following Chen & Li (2022) and Liu et al. (2023), we adopt an information criterion approach to determine $r$. Define

$$\text{IC}(r) = \left\| \tilde{\mathbf{M}}^{(1)} - \mathbf{F}(\mathbf{B}^{(1)})^{\top} \right\|_F^2 + \left\| \tilde{\mathbf{M}}^{(2)} - \mathbf{F}(\mathbf{B}^{(2)})^{\top} \right\|_F^2 + r\, \eta(n,p),$$

and estimate the dimensionality via

$$\hat{r} = \underset{1 \leq r \leq r_{\max}}{\arg\min}\ \text{IC}(r),$$

where $r_{\max}$ is a predetermined upper bound satisfying $r^* \leq r_{\max} < n \wedge p$, and $\eta(n,p)$ is a penalty function. The following theorem establishes the selection consistency of our criterion.

**Theorem 3.8.** *Suppose that the estimation error in the first stage satisfies $\left\| \tilde{\mathbf{M}}^{(k)} - \mathbf{M}^{(k)*} \right\|_F \lesssim err$ with probability tending to one for $k \in \{1, 2\}$. Under Condition (C1), and provided that the penalty function satisfies $(err)^2 + s\lambda \ll \eta(n,p) \ll n\sigma_{\mathbf{B}}^2$, then $\mathbb{P}(\hat{r} = r^*) \to 1$ as $n, p \to +\infty$.*

Existing low rank information criteria charge about $(n + p)r$ leading parameters. SMART additionally estimates the sparse loading differences, so under the $(\mathbf{B}^{(1)}, \Delta\mathbf{B})$ parameterization the leading complexity is about $(n + p + s)r$, and a criterion that ignores the extra term underestimates the model complexity when the shift is nonnegligible. We recommend the penalty functions $\eta_1(n,p) = 2(n + p + \hat{s}_r) \log\{np/(n+p)\}$ and $\eta_2(n,p) = 2(n + p + \hat{s}_r) \log(n \wedge p)$, where $\hat{s}_r$ denotes the number of nonzero rows of the estimated shift matrix obtained under candidate rank $r$, owing to their empirical performance (see Section 5.1 for details), and establish their theoretical validity in the following corollary.

**Corollary 3.9.** *Under the conditions of Theorem 3.8, assume that the observations $x_{ij}$ are independent given the common representation $\boldsymbol{f}_i$, and that the GFM estimator is utilized in the first stage. Furthermore, assume $s\lambda \ll n \vee p$ and $\sigma_{\mathbf{B}} \asymp \sqrt{p}$. If the penalty function is specified as either $\eta_1(n,p)$ or $\eta_2(n,p)$, then $\mathbb{P}(\hat{r} = r^*) \to 1$ as $n, p \to +\infty$.*

## 4 Extensions: Multi-Environment Scenario

While the preceding analysis focused on pairwise environmental shifts, extending SMART to multiple environments facilitates comprehensive structural shift detection. This setting is natural for time series analysis, change point detection, and transfer learning involving several source domains.

Consider data matrices $\mathbf{X}^{(1)}, \ldots, \mathbf{X}^{(K)}$ from $K$ environments. Retaining the two-stage strategy, we perform the signal recovery stage followed by a joint optimization across all $K$ environments. The estimators are obtained by minimizing the cumulative reconstruction error penalized by adjacent structural shifts:

$$\left( \hat{\mathbf{F}}, \{\hat{\mathbf{B}}^{(k)}\}_{k=1}^K \right) = \underset{\mathbf{F}, \mathbf{B}^{(k)}}{\arg\min} \left\{ \sum_{k=1}^K \left\| \tilde{\mathbf{M}}^{(k)} - \mathbf{F}(\mathbf{B}^{(k)})^{\top} \right\|_F^2 + \lambda \sum_{k=2}^K \sum_{j=1}^p \left\| \boldsymbol{b}_j^{(k)} - \boldsymbol{b}_j^{(k-1)} \right\|_2 \right\}. \tag{3}$$

We suppose that the sample representations remain invariant across environments, attributing observed variations to shifts in feature influences. The optimization problem is solved via an alternating iterative algorithm analogous to the pairwise case.

We establish the theoretical guarantees for the multi-environment setting as follows.

**Theorem 4.1.** *Suppose that the estimation error in the first stage satisfies $\left\|\tilde{\mathbf{M}}^{(k)} - \mathbf{M}^{(k)*}\right\|_F \lesssim err$ with probability tending to one for all $k \in \{1,\dots,K\}$. Under Condition (C1) and assuming $r \geq r^*$, the average reconstruction error satisfies*

$$\frac{1}{K}\sum_{k=1}^{K}\left\|\mathbf{M}^{(k)*} - \hat{\mathbf{F}}\left(\hat{\mathbf{B}}^{(k)}\right)^{\top}\right\|_F \lesssim err + \sqrt{s\lambda}.$$

*Furthermore, under Condition (C2) and assuming $r = r^*$, there exists an orthogonal rotation matrix $\mathbf{O}$ such that the following convergence rates hold with probability tending to one:*

$$\frac{1}{\sqrt{n}}\left\|\hat{\mathbf{F}}\mathbf{O} - \mathbf{F}^*\right\|_F \lesssim \frac{err + \sqrt{s\lambda}}{\sqrt{n}\sigma_{\mathbf{B}}}, \qquad \frac{1}{K}\sum_{k=1}^{K}\left(\frac{1}{\sqrt{p}}\left\|\hat{\mathbf{B}}^{(k)}\mathbf{O} - \mathbf{B}^{(k)*}\right\|_F\right) \lesssim \frac{err + \sqrt{s\lambda}}{\sqrt{n}\sigma_{\mathbf{B}}},$$

*and*

$$\frac{1}{K-1}\sum_{k=2}^{K}\left(\frac{1}{\sqrt{s}}\left\|\hat{\Delta\mathbf{B}}^{(k)}\mathbf{O} - \Delta\mathbf{B}^{(k)*}\right\|_F\right) \lesssim \frac{\sqrt{p}}{\sigma_{\mathbf{B}}}\cdot\frac{err + \sqrt{s\lambda}}{\sqrt{ns}},$$

*where $\Delta\mathbf{B}^{(k)*} = \mathbf{B}^{(k)*} - \mathbf{B}^{(k-1)*}$.*

The result mirrors the pairwise theory: the second stage inherits the preliminary recovery error while the fused group penalty localizes which rows change between adjacent environments. Hence the inferential target becomes the union, or sequence, of row supports across the $K-1$ adjacent shifts.

We make this target precise and extend the support theory of Section 3.4. For $k \in \{2,\dots,K\}$ let $\mathcal{S}_k^* = \{j : \|\Delta\boldsymbol{b}_j^{(k)*}\|_2 > 0\}$ be the support of the $k$th adjacent shift. The inferential target is the edge feature set $\mathcal{A}^* = \{(k,j) : 2 \leq k \leq K, \ \|\Delta\boldsymbol{b}_j^{(k)*}\|_2 > 0\}$, which records both the shifted features and the environments where they shift rather than only their union. Write $s_K = |\mathcal{A}^*|$ and $\beta_{\min,K} = \min_{(k,j)\in\mathcal{A}^*}\|\Delta\boldsymbol{b}_j^{(k)*}\|_2$. In Theorem 4.1 the quantity $s$ denotes the union sparsity $|\cup_{k=2}^{K}\mathcal{S}_k^*|$, which satisfies $s \leq s_K \leq (K-1)s$.

For $K \geq 3$ the fused penalty couples the adjacent differences across environments, so the supports are read off an edgewise estimate rather than off the raw fused solution. Write $\hat{\Delta\boldsymbol{b}}_j^{(k)}$ for the resulting shift rows. The estimated edge feature set is $\hat{\mathcal{A}} = \{(k,j) : \|\hat{\Delta\boldsymbol{b}}_j^{(k)}\|_2 > 0\}$.

*Condition* (C3'). For $k \in \{2,\dots,K\}$ let $\boldsymbol{e}_{\Delta,j}^{(k)}$ denote the $j$th column of $(\tilde{\mathbf{M}}^{(k)} - \mathbf{M}^{(k)*}) - (\tilde{\mathbf{M}}^{(k-1)} - \mathbf{M}^{(k-1)*})$. There exists a sequence $\rho_{K,n} > 0$ such that $\max_{2\leq k\leq K}\max_{1\leq j\leq p}\|n^{-1}(\mathbf{F}^*)^{\top}\boldsymbol{e}_{\Delta,j}^{(k)}\|_2 = O_p(\rho_{K,n})$ and $\max_{2\leq k\leq K}\max_{1\leq j\leq p}n^{-1/2}\|\boldsymbol{e}_{\Delta,j}^{(k)}\|_2 = O_p(1)$.

**Theorem 4.2.** *Let $K$ be fixed. Suppose that $\max_{1\leq k\leq K}\|\tilde{\mathbf{M}}^{(k)} - \mathbf{M}^{(k)*}\|_F \lesssim err$ with probability tending to one for a sequence $err$, that Condition (C1), Condition (C2) and Condition (C3') hold with constants uniform over the $K$ environments, and that $r = r^*$. Assume $\mathcal{A}^* \neq \emptyset$ and let $\lambda$ be a sequence satisfying*

$$\max\left\{n\rho_{K,n}, \ \frac{\sqrt{n}\,err}{\sigma_{\mathbf{B}}}, \ \frac{ns_K}{\sigma_{\mathbf{B}}^2}\right\} \ll \lambda \ll \min\left\{n\beta_{\min,K}, \ \frac{n \vee p}{s_K}\right\}.$$

*Then $\mathbb{P}(\hat{\mathcal{A}} = \mathcal{A}^*) \to 1$ as $n,p \to +\infty$.*

For determining the latent dimensionality, we generalize the information criterion approach. Defining the penalized objective for $K$ environments, we estimate the dimensionality via:

$$\hat{r} = \underset{1\leq r\leq r_{\max}}{\arg\min}\left\{\sum_{k=1}^{K}\left\|\tilde{\mathbf{M}}^{(k)} - \hat{\mathbf{F}}_r(\hat{\mathbf{B}}_r^{(k)})^{\top}\right\|_F^2 + r\,\eta_K(n,p)\right\},$$

where $\hat{\mathbf{F}}_r$ and $\hat{\mathbf{B}}_r^{(k)}$ denote the estimators obtained under rank $r$, and $\eta_K(n,p)$ is a penalty function. In practice we take $\eta_K(n,p) = 2(n + p + \hat{s}_{K,r})\,g(n,p)$ with $g(n,p) \in \{\log\{np/(n+p)\}, \log(n \wedge p)\}$, where $\hat{s}_{K,r}$ counts the estimated shifts across all adjacent pairs under candidate rank $r$.. The theoretical consistency of this selection criterion is established below.

**Theorem 4.3.** *Suppose that the conditions established in Theorem 3.8 are satisfied for all $k \in \{1, \ldots, K\}$. Then $\mathbb{P}(\hat{r} = r^*) \to 1$ as $n, p \to +\infty$.*

We also develop a one-step estimator for additive Gaussian noise, which merges the two stages into a single penalized objective and avoids a separate denoising step. Because it applies to a narrower data setting than the two-stage procedure, we present the formulation, its theoretical guarantee, and the corresponding experiments in Appendix B.

# 5 Experiments on Synthetic and Real Data

## 5.1 Experiments on Synthetic Data

### 5.1.1 Experiment Settings

In this section, we evaluate the finite-sample performance through 200 Monte Carlo simulations. The data generation process follows the generalized linear models framework described in Example 2.

Regarding parameter settings, the elements of $\mathbf{F}$ and $\mathbf{B}^{(1)}$ are independently sampled from $U[-\alpha, \alpha]$. The structural difference $\Delta\mathbf{B}$ follows a row-sparse assumption, where a support set $\mathcal{S}^*$ of size $s$ is randomly selected, with non-zero elements sampled from $U[-c, c]$. The simulations cover three typical distributions: Gaussian, Poisson, and Binary.

The experimental grid includes sample sizes and dimensions $(n, p) \in \{50, 100, 300\}^2$, true ranks $r^* \in \{2, 4\}$, sparsity levels $s \in \{5, 10, 20\}$, signal strength $c = 2$, and distribution parameter $\alpha = 2$. To ensure numerical stability, data values are truncated at 2000 for the Poisson distribution. We benchmark the SMART method against Naïve Lasso (N-Lasso) and unpenalized SVD estimator. For all penalized methods, the regularization parameter $\lambda$ is determined via Cross-Validation (CV). Detailed algorithmic implementations for these benchmarks are deferred to Appendix A. Due to space constraints, we focus the discussion on the Poisson distribution as a representative case. Results for Gaussian and Binary distributions, which exhibit similar trends, are provided in Appendix D. We employ two metrics to assess the accuracy of variable selection and subspace recovery:

The comparison is designed as a controlled ablation. N-Lasso uses the same Stage I recovery, initialization, alternating updates, and CV procedure as SMART, but replaces the rowwise group $\ell_{2,1}$ penalty with an entrywise $\ell_1$ penalty on $\Delta\mathbf{B}$. This isolates the effect of enforcing sparsity at the row level. The unpenalized SVD baseline removes the sparsity penalty altogether and serves as a low-rank reconstruction reference for subspace recovery. For variable selection we additionally equip the SVD baseline with a row level selection rule. The rule ranks the rows of its estimated difference $\hat{\Delta\mathbf{B}}$ by their norms and truncates the ranking at the largest gap of the sorted sequence, so the baseline selects a support without any oracle knowledge of the true sparsity level. The exact rule is given in Appendix A, and all methods share the same Stage I input, rank, and row level metrics. Stage I is computed with the true number of factors and is shared by all methods.

$F_1$ Score: Evaluates the attribution results. Let $\hat{\mathcal{S}} = \{j : \|\hat{\Delta}\boldsymbol{b}_j\|_2 > 10^{-5}\}$; the score is defined as:

$$F_1 = 2|\mathcal{S}^* \cap \hat{\mathcal{S}}|/(|\mathcal{S}^*| + |\hat{\mathcal{S}}|).$$

Minimum Canonical Correlation ($\rho_{\min}$): Measures the alignment between true subspace $\mathbf{U}$ and estimate $\hat{\mathbf{U}}$:

$$\rho_{\min}(\mathbf{U}, \hat{\mathbf{U}}) = \sigma_{\min}(\mathbf{U}^\top \hat{\mathbf{U}}).$$

This metric corresponds to the cosine of the largest principal angle, where values closer to 1 indicate superior recovery. In the main text, we focus on the subspace recovery of $\mathbf{B}^{(k)}$. Analysis regarding the common representation matrix $\mathbf{F}$, as well as additional metrics, is provided in Appendix D. We further validated the one-step estimation method on incomplete data in Appendix D.

### 5.1.2 Experimental Results

For variable selection of the structural difference $\Delta\mathbf{B}$, we compare SMART with the Naïve Lasso and with the SVD baseline equipped with the row level selection rule, because the raw SVD estimate has no exactly zero rows and produces a support only through such a rule. The results for Poisson data are summarized in

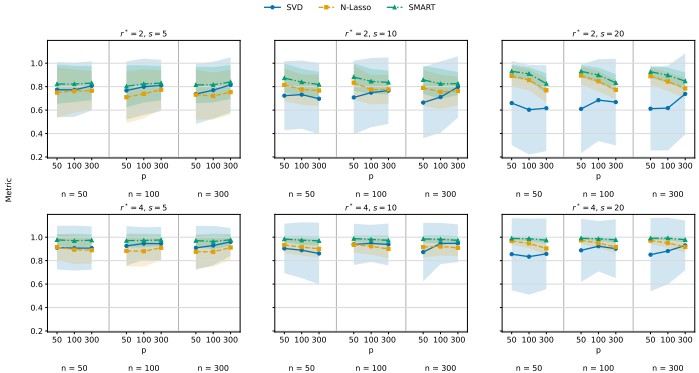

Figure 2: Variable selection performance ($F_1$ score) for Poisson data, with varying $n$, $p$, $r^*$, and $s$ over 200 repetitions.

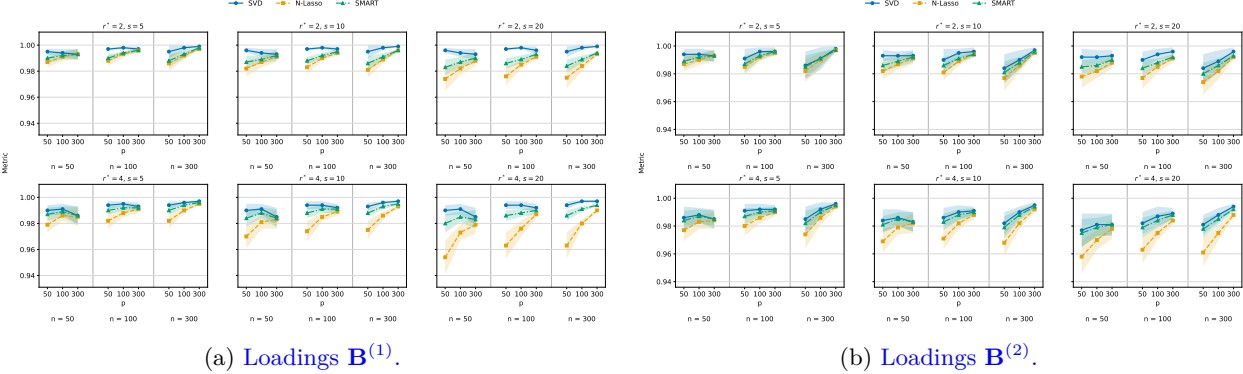

(a) Loadings $\mathbf{B}^{(1)}$.               (b) Loadings $\mathbf{B}^{(2)}$.

Figure 3: Subspace recovery accuracy ($\rho_{\min}$) for Poisson data, with varying $n$, $p$, $r^*$, and $s$ over 200 repetitions.

Figure 2. Compared to the SVD and Naïve Lasso, SMART achieves superior performance with high stability regarding sample size $n$. It remains robust against high dimensionality $p$, and notably, its selection capability strengthens as the true rank $r^*$ and $s$ increase, benefiting from richer signal information.

As shown in Figure 3a and 3b, SMART accurately recovers the latent structure, surpassing Naïve Lasso and aligning closely with unpenalized SVD. This confirms that introducing sparsity does not hinder the estimation of the latent representation space. The method exhibits consistent asymptotic properties, with accuracy improving as $n$ and $p$ increase, thereby successfully balancing structural selection and estimation fidelity.

The rank selection criteria recover the true rank in almost every replication across all three distributions; the setup and full results are reported in Appendix D.1. Two further experiments probe the theory directly. The empirical estimation errors track the theoretical scaling with $n$ and $p$ (Appendix D.6), and direct tests on observed coordinates fail in a setting where the observed marginal distribution does not change, whereas SMART still recovers the changed mechanism support (Appendix D.7).

Three further sets of experiments broaden the empirical evidence. For the multi-environment extension of Section 4, varying $K$ from 2 to 5 keeps the union $F_1$ scores high while the fitting time grows smoothly with $K$ (Appendix D.8). On mixed data combining Gaussian and Poisson coordinates under the same latent loading structure, SMART remains competitive or superior to N-Lasso, supporting the modular Stage I design (Appendix D.9). Comparing one-step and two-stage estimation under varying Gaussian noise, the one-step estimator is faster and slightly more accurate at low noise, whereas the two-stage estimator also covers non-Gaussian and mixed data (Appendix D.10).

## 5.2 Experiments on Real Data

In this section, we demonstrate the practical utility of the SMART framework. We primarily focus on identifying structural shifts in high-dimensional gene expression data to discover potential biomarkers for breast cancer. As an additional real data analysis, we also apply SMART to player performance metrics from the NBA 2023-24 season; the full NBA analysis is reported in Appendix E.3.

Breast cancer is a highly heterogeneous disease driven by complex molecular alterations. Distinguishing between "passenger" mutations and true "driver" structural shifts remains a critical challenge in oncology. By applying SMART to paired Tumor and Para-cancerous tissue samples, we aim to isolate significant mechanistic deviations ($\Delta\mathbf{B}$) that underpin tumorigenesis, thereby identifying robust candidate biomarkers for diagnosis and prognosis.

We utilize paired gene expression data from the TCGA-BRCA dataset. To mitigate sequencing noise, we filter genes by mean expression (10–5000), then compute the coefficient of variation ($\sigma/\mu$) in tumor samples and select the top 300 genes as model input. The model was configured with $r = 6$ and $\lambda = 3$ based on the selection heuristics described in Appendix E. Stage I is run once with a preset $r_{\max}$ and then held fixed, so the preliminary estimate does not depend on the candidate rank.

The model identifies 13 significant genes in $\Delta\mathbf{B}$ (Figure 4). We highlight key findings. Firstly, SMART captures a cohesive cluster of genes, including `CSN1S1`, `CSN3`, and `LALBA`, that are associated with aberrant alveogenesis. Langille et al. (2022) posits that the expression of these markers reflects lineage infidelity, where cells (potentially including basal cells) inappropriately activate the lactation program, a process linked to tumorigenesis. Additionally, the model identifies `CA6`, which is consistent with findings showing its significant differential expression between breast cancer and normal tissues Wu et al. (2020). These results suggest that SMART effectively pinpoints markers of specific pathogenic developmental programs.

The remaining selected genes provide further biological validation of the detected structural shifts, spanning tumor suppression, epigenetic regulation, and metabolic adaptation; we discuss them in Appendix E.2.

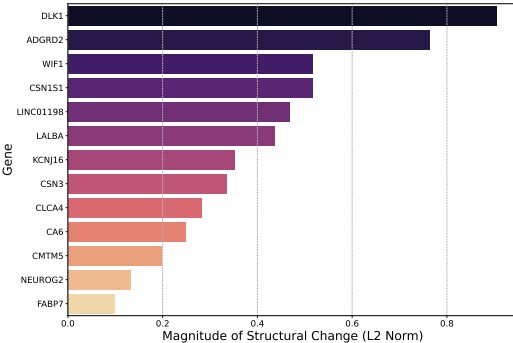

Figure 4: Significant non-zero difference loadings identified in the TCGA dataset and their magnitudes.

## 6    Conclusion

In this paper, we introduced SMART, a modular two-stage framework that bridges the gap between signal recovery and structural attribution. By integrating low-rank reconstruction error minimization with a group-structured sparse penalty, SMART effectively achieves the precise localization of mechanism shifts. This formulation ensures that the identified sparse structural drifts are invariant to rotational ambiguity, thereby explicitly disentangling invariant components from specific mechanism changes and providing interpretable insights into dynamic systems.

Theoretically, we established that the estimators via SMART inherit the convergence rates of the preliminary signal recovery and provided a consistent information criterion for latent dimensionality selection. The framework's versatility is further demonstrated through its extensions to a one-step estimator for additive noise models and multi-environment scenarios. Empirically, the efficacy of SMART is validated on both count data (BRCA Gene Expression) and continuous data (NBA), showcasing its broad applicability across diverse scientific domains.

Several avenues remain for future exploration. First, the current second-stage attribution assumes a linear interaction between latent representations and features in the attribution stage; extending this framework to accommodate non-linear structural disentanglement represents a valuable direction. Second, the performance of the attribution stage relies on the quality of the initial signal recovery. Future research could investigate more robust signal recovery strategies to mitigate potential error propagation.

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

# A   Algorithm and Implementation Details

## A.1   Algorithms

In this section, we provide pseudocode for four algorithms: our proposed method, SMART; two compared methods, sparse mechanism attribution with Naïve Lasso and Unpenalized SVD Projection; and the Constrained Joint Maximum Likelihood Estimation (CJMLE) to solve model for generalized linear data (Example 2) (Chen et al., 2020; Chen & Li, 2022), which serves as the shared algorithm for the Stage I signal recovery.

---

**Algorithm 1** CJMLE: Constrained Joint Maximum Likelihood Estimation Algorithm

---

1:  **Input:** Observed data $\mathbf{X} \circ \mathbf{\Omega} = (x_{ij}\omega_{ij})$, latent dimensionality $r$, constraint constant $C'$.
2:      Initialize $\boldsymbol{f}_i^{[0]}$ and $\boldsymbol{b}_j^{[0]}$ for all $i, j$.
3:  **Alternating Minimization:**
4:  **for** $l = 1, 2, \ldots$ **do**
5:      **Update Row Parameters ($\boldsymbol{f}$):**
6:      **for** $i = 1, \ldots, n$ **do**
7:          Compute $\boldsymbol{f}_i^{[l]}$ by minimizing the negative log-likelihood:

$$\boldsymbol{f}_i^{[l]} \in \underset{\|\boldsymbol{f}\| \leq C'}{\arg\min} - \sum_{j=1}^{p} \omega_{ij} \left[ x_{ij}(\boldsymbol{f}^\top \boldsymbol{b}_j^{(l-1)}) - \psi(\boldsymbol{f}^\top \boldsymbol{b}_j^{(l-1)}) \right].$$

8:      **end for**
9:      **Update Column Parameters ($\boldsymbol{b}$):**
10:     **for** $j = 1, \ldots, p$ **do**
11:         Compute $\boldsymbol{b}_j^{[l]}$ by minimizing the negative log-likelihood:

$$\boldsymbol{b}_j^{[l]} \in \underset{\|\boldsymbol{b}\| \leq C'}{\arg\min} - \sum_{i=1}^{n} \omega_{ij} \left[ x_{ij}((\boldsymbol{f}_i^{[l]})^\top \boldsymbol{b}) - \psi((\boldsymbol{f}_i^{[l]})^\top \boldsymbol{b}) \right].$$

12:     **end for**
13:     **if** converged **then**
14:         **break**
15:     **end if**
16: **end for**
17: Iteratively perform Alternating Minimization until convergence.
18: **Return:** $\hat{\boldsymbol{f}}_i = \boldsymbol{f}_i^{[L]}$ and $\hat{\boldsymbol{b}}_j = \boldsymbol{b}_j^{[L]}$ where $L$ is the last iteration.

---

For variable selection, the SVD baseline ranks the rows of $\hat{\Delta}\mathbf{B}$ by their norms $d_j = \|\hat{\Delta}\boldsymbol{b}_j\|_2$ and truncates the ranking at the largest gap of the sorted sequence $d_{(1)} \geq \cdots \geq d_{(p)}$. The number of selected rows is $\hat{s}_{\text{diff}} = \min \arg\max_{1 \leq k \leq p-1}(d_{(k)} - d_{(k+1)})$, where the minimum breaks ties toward the smaller index, and $\hat{s}_{\text{diff}} = 0$ whenever $d_{(1)} \leq 10^{-5}$, which matches the numerical zero threshold used for the penalized methods. The selected support consists of the $\hat{s}_{\text{diff}}$ variables with the largest row norms, and the remaining rows are set to zero. The rule uses no oracle information about the true sparsity level, so the comparison with SMART and the Naïve Lasso is conducted on equal terms.

## A.2   Implementation Details

To determine the optimal regularization parameter $\lambda$, we implement an entry-wise 3-fold CV strategy specifically designed for matrix-structured data. This process integrates both Stage I and Stage II to ensure that the latent signal recovery does not leak information about the validation set.

Before initiating the pipeline, the indices of the observed data matrices are randomly partitioned into three disjoint sets. In each fold, one set is designated as the validation mask $\mathbf{\Omega}_{\text{val}}$, effectively treating these entries as missing values during the training phase. The remaining entries $\mathbf{\Omega}_{\text{train}}$ are fed into the CJMLE framework

---

**Algorithm 2** SMART: Sparse Mechanism Attribution for Representation Algorithm

---

1: **Input:** Observed data $\mathbf{X}^{(1)} \circ \mathbf{\Omega}^{(1)} = (x_{ij}^{(1)} \omega_{ij}^{(1)}), \mathbf{X}^{(2)} \circ \mathbf{\Omega}^{(2)} = (x_{ij}^{(2)} \omega_{ij}^{(2)})$, regularization parameter $\lambda$, latent dimensionality $r$, constraint constant $C$.

    **Stage I: Signal Recovery**

    $\tilde{\mathbf{M}}^{(1)}, \tilde{\mathbf{M}}^{(2)} \leftarrow \mathbf{X}^{(1)} \circ \mathbf{\Omega}^{(1)}, \mathbf{X}^{(2)} \circ \mathbf{\Omega}^{(2)}$ via CJMLE (Algorithm 1) or other methods.

    **Stage II: Sparse Mechanism Attribution**

2: **Initialization:**

3:     Compute SVD: $\mathbf{U}, \mathbf{\Sigma}, \mathbf{V}^\top = \mathrm{SVD}(\tilde{\mathbf{M}}^{(1)}, \tilde{\mathbf{M}}^{(2)})$.

4:     Initialize $\mathbf{F}$ and $\Delta\mathbf{B}$: $\mathbf{F}^{[0]} \leftarrow \sqrt{n}\mathbf{U}_{:,1:r}, \quad \Delta\mathbf{B}^{[0]} \leftarrow \mathbf{0}$.

5: **Iterative Solver**

6: **for** $t = 1, 2, \ldots$ **do**

7:     **Update $\mathbf{B}^{(1)}$:**

$$\mathbf{B}^{(1)[t]} = \frac{1}{2}\left(\tilde{\mathbf{M}}^{(1)} + \tilde{\mathbf{M}}^{(2)}\right)^\top \mathbf{F}^{[t-1]}\left((\mathbf{F}^{[t-1]})^\top \mathbf{F}^{[t-1]}\right)^{-1} - \frac{1}{2}\Delta\mathbf{B}^{[t-1]}.$$

8:     **Update $\Delta\mathbf{B}$ via group Lasso:**

$$\Delta\boldsymbol{b}_j^{[t]} = \arg\min_{\Delta\boldsymbol{b}_j} \sum_{i=1}^n \left[\left\{\tilde{m}_{ij}^{(1)} - (\boldsymbol{b}_j^{(1)[t]})^\top \boldsymbol{f}_i^{[t-1]}\right\}^2 + \left\{\tilde{m}_{ij}^{(2)} - (\boldsymbol{b}_j^{(1)[t]})^\top \boldsymbol{f}_i^{[t-1]} - \Delta\boldsymbol{b}_j^\top \boldsymbol{f}_i^{[t-1]}\right\}^2\right] + \lambda\|\Delta\boldsymbol{b}_j\|_2.$$

9:     **Update F:**

$$\mathbf{F}^{[t]} = \arg\min_{\mathbf{F}} \left\|\tilde{\mathbf{M}}^{(1)} - \mathbf{F}(\mathbf{B}^{(1)[t]})^\top\right\|_F^2 + \left\|\tilde{\mathbf{M}}^{(2)} - \mathbf{F}(\mathbf{B}^{(1)[t]} + \Delta\mathbf{B}^{[t]})^\top\right\|_F^2.$$

10:     **Orthogonalization and Projection:** Ensure that the estimator lies within the feasible set $\mathcal{F}(C)$.

11: **end for**

12: **Return:** $\hat{\mathbf{F}} = \mathbf{F}^{[t]}, \hat{\mathbf{B}}^{(1)} = \mathbf{B}^{(1)[t]}, \hat{\Delta\mathbf{B}} = \Delta\mathbf{B}^{[t]}$.

---

(Algorithm 1) to recover the training signals $\tilde{\mathbf{M}}_{\mathrm{train}}$. Stage II is then executed for each value in a candidate grid of $\lambda$ using these training signals. The candidate grids are defined as follows:

- **Gaussian:** $\{0.60, 0.61, \ldots, 0.70\}$

- **Poisson:** $\{0.80, 0.81, \ldots, 0.90\}$

- **Binary:** $\{\exp(0), \exp(0.2), \ldots, \exp(2.0)\}$

The optimal parameter is chosen by minimizing the prediction error on the masked validation entries. Once the optimal $\lambda^*$ is selected, the final model is retrained on the full dataset. The complete selection process is detailed in Algorithm 5.

## B  Extensions: One-Step Estimation

For additive noise models (as described in Example 1), we can integrate the two-stage procedure into a one-step estimation algorithm. In this one-step formulation, we substitute the raw observations $\mathbf{X}^{(k)}$ for the stage-one estimates $\tilde{\mathbf{M}}^{(k)}$. Specifically, we directly estimate by

$$\left(\hat{\mathbf{F}}, \hat{\mathbf{B}}^{(1)}, \hat{\Delta\mathbf{B}}\right) = \arg\min_{\mathbf{F}, \mathbf{B}^{(1)}, \Delta\mathbf{B}} \left\{\left\|\mathbf{X}^{(1)} - \mathbf{F}(\mathbf{B}^{(1)})^\top\right\|_F^2 + \left\|\mathbf{X}^{(2)} - \mathbf{F}(\mathbf{B}^{(1)} + \Delta\mathbf{B})^\top\right\|_F^2 + \lambda\|\Delta\mathbf{B}\|_{2,1}\right\}.$$

Minimizing this objective yields estimates for the representation $\mathbf{F}$, the loadings $\mathbf{B}^{(k)}$, and the structural shift $\Delta\mathbf{B}$. We employ an analogous alternating algorithm to solve this problem. Let $\varepsilon_{ij}$ denote the $(i, j)$-th entry of the noise matrix $\mathcal{E}$.

---

**Algorithm 3** Structural Disentanglement with Naïve Lasso Algorithm

---

1: **Input:** Observed data $\mathbf{X}^{(1)} \circ \mathbf{\Omega}^{(1)} = (x_{ij}^{(1)} \omega_{ij}^{(1)}), \mathbf{X}^{(2)} \circ \mathbf{\Omega}^{(2)} = (x_{ij}^{(2)} \omega_{ij}^{(2)})$, regularization parameter $\lambda$, latent dimensionality $r$, constraint constant $C$.
 **Stage I: Signal Recovery**
 $\tilde{\mathbf{M}}^{(1)}, \tilde{\mathbf{M}}^{(2)} \leftarrow \mathbf{X}^{(1)} \circ \mathbf{\Omega}^{(1)}, \mathbf{X}^{(2)} \circ \mathbf{\Omega}^{(2)}$ via CJMLE (Algorithm 1) or other methods.
 **Stage II: Sparse Estimation via Lasso**
2: **Initialization:**
3:    Compute SVD: $\mathbf{U}, \mathbf{\Sigma}, \mathbf{V}^\top = \mathrm{SVD}(\tilde{\mathbf{M}}^{(1)}, \tilde{\mathbf{M}}^{(2)})$
4:    Initialize $\mathbf{F}$ and $\Delta\mathbf{B}$: $\mathbf{F}^{[0]} \leftarrow \sqrt{n}\mathbf{U}_{:,1:r}, \quad \Delta\mathbf{B}^{[0]} \leftarrow \mathbf{0}$
5: **Iterative Solver**
6: **for** $t = 1, 2, \ldots$ **do**
7:    **Update $\mathbf{B}^{(1)}$ via Lasso:**

$$\mathbf{B}^{(1)[t]} = \frac{1}{2}\left(\tilde{\mathbf{M}}^{(1)} + \tilde{\mathbf{M}}^{(2)}\right)^\top \mathbf{F}^{[t-1]}\left((\mathbf{F}^{[t-1]})^\top \mathbf{F}^{[t-1]}\right)^{-1} - \frac{1}{2}\Delta\mathbf{B}^{[t-1]}.$$

8:    **Update $\Delta\mathbf{B}$:**

$$\Delta\boldsymbol{b}_j^{[t]} = \arg\min_{\Delta\boldsymbol{b}_j} \sum_{i=1}^n \left[\left\{\tilde{m}_{ij}^{(1)} - (\boldsymbol{b}_j^{(1)[t]})^\top \boldsymbol{f}_i^{[t-1]}\right\}^2 + \left\{\tilde{m}_{ij}^{(2)} - (\boldsymbol{b}_j^{(1)[t]})^\top \boldsymbol{f}_i^{[t-1]} - \Delta\boldsymbol{b}_j^\top \boldsymbol{f}_i^{[t-1]}\right\}^2\right] + \lambda\|\Delta\boldsymbol{b}_j\|_1.$$

9:    **Update F:**

$$\mathbf{F}^{[t]} = \arg\min_{\mathbf{F}} \left\|\tilde{\mathbf{M}}^{(1)} - \mathbf{F}(\mathbf{B}^{(1)[t]})^\top\right\|_F^2 + \left\|\tilde{\mathbf{M}}^{(2)} - \mathbf{F}(\mathbf{B}^{(1)[t]} + \Delta\mathbf{B}^{[t]})^\top\right\|_F^2.$$

10:    **Orthogonalization and Projection:** Ensure that the estimator lies within the feasible set $\mathcal{F}(C)$.
11: **end for**
12: **Return:** $\hat{\mathbf{F}} = \mathbf{F}^{[t]}, \hat{\mathbf{B}}^{(1)} = \mathbf{B}^{(1)[t]}, \hat{\Delta\mathbf{B}} = \Delta\mathbf{B}^{[t]}$.

---

**Algorithm 4** Unpenalized SVD Algorithm

---

1: **Input:** Observed data $\mathbf{X}^{(1)} \circ \mathbf{\Omega}^{(1)} = (x_{ij}^{(1)} \omega_{ij}^{(1)}), \mathbf{X}^{(2)} \circ \mathbf{\Omega}^{(2)} = (x_{ij}^{(2)} \omega_{ij}^{(2)})$, latent dimensionality $r$.
2: **Signal Recovery:** $\tilde{\mathbf{M}}^{(1)}, \tilde{\mathbf{M}}^{(2)} \leftarrow \mathbf{X}^{(1)} \circ \mathbf{\Omega}^{(1)}, \mathbf{X}^{(2)} \circ \mathbf{\Omega}^{(2)}$ via CJMLE (Algorithm 1) or other methods.
3: **SVD and Factor Extraction:**
4:    Compute SVD: $\mathbf{U}, \mathbf{\Sigma}, \mathbf{V}^\top = \mathrm{SVD}(\tilde{\mathbf{M}}^{(1)}, \tilde{\mathbf{M}}^{(2)})$
5:    Extract Common Representations: $\hat{\mathbf{F}} \leftarrow \sqrt{n}\mathbf{U}_{:,1:r}$.
6: **Projection:**
7:    Compute Pseudo-inverse: $\mathbf{F}^\dagger \leftarrow (\hat{\mathbf{F}}^\top \hat{\mathbf{F}})^{-1}\hat{\mathbf{F}}^\top$.
8:    Estimate Loadings for each environment:

$$\hat{\mathbf{B}}^{(1)} = (\mathbf{F}^\dagger \tilde{\mathbf{M}}^{(1)})^\top, \quad \hat{\mathbf{B}}^{(2)} = (\mathbf{F}^\dagger \tilde{\mathbf{M}}^{(2)})^\top.$$

9: **Compute Structural Difference:**
$$\hat{\Delta\mathbf{B}} = \hat{\mathbf{B}}^{(2)} - \hat{\mathbf{B}}^{(1)}.$$

10: **Return:** $\hat{\mathbf{F}}, \hat{\mathbf{B}}^{(1)}, \hat{\Delta\mathbf{B}}$.

---

To establish the theoretical guarantees for this one-step estimator, we introduce an additional regularity condition.

*Condition* (C4). (C3.1) $\mathbb{E}\,|\varepsilon_{ij}|^\xi \leq C_1$ for $\xi > 14$.

(C3.2) $\frac{1}{n}\sum_{i_1=1}^n \sum_{i_2=1}^n \left\{p^{-1}\sum_{j=1}^p \mathbb{E}\,(\varepsilon_{i_1j}\varepsilon_{i_2j})\right\}^2 \leq C_2$.

(C3.3) $\mathbb{E}\left(p^{-\frac{1}{2}}\sum_{j=1}^p \left[\varepsilon_{i_1j}\varepsilon_{i_2j} - \mathbb{E}\,(\varepsilon_{i_1j}\varepsilon_{i_2j})\right]\right)^2 \leq C_3$ for any $(i_1, i_2)$.

---

**Algorithm 5** Entry-wise CV for Hyperparameter Tuning Algorithm

---

1: **Input:** Observed data $\mathbf{X}^{(1)}, \mathbf{X}^{(2)}$, candidate grid $\boldsymbol{\Lambda}_{\text{grid}}$, latent dimensionality $r$.
2:   Partition observed entries of $\boldsymbol{\Omega}$ into three disjoint sets: $\mathcal{I}_1, \mathcal{I}_2, \mathcal{I}_3$.
3: **for** $\lambda \in \boldsymbol{\Lambda}_{\text{grid}}$ **do**
4:     Initialize validation error: $E_\lambda \leftarrow 0$.
5:   **for** $k = 1$ to $3$ **do**
6:     **Masking:**
7:       Define Validation Mask: $\boldsymbol{\Omega}_{\text{val}} \leftarrow \mathcal{I}_k$.
8:       Define Training Mask: $\boldsymbol{\Omega}_{\text{train}} \leftarrow \boldsymbol{\Omega} \setminus \mathcal{I}_k$.
9:       $\hat{\mathbf{F}}, \hat{\mathbf{B}}^{(1)}, \hat{\Delta}\mathbf{B} \leftarrow \text{SMART}(\mathbf{X}^{(1)} \circ \boldsymbol{\Omega}_{\text{train}}^{(1)}, \mathbf{X}^{(2)} \circ \boldsymbol{\Omega}_{\text{train}}^{(2)}, \lambda, r)$ (or Structural Disentanglement with Naïve Lasso).
10:     Predict and evaluate on validation set:
11:       $\hat{\mathbf{X}}_{\text{val}}^{(1)}, \hat{\mathbf{X}}_{\text{val}}^{(2)} \leftarrow \hat{\mathbf{F}}(\hat{\mathbf{B}}^{(1)})^\top, \hat{\mathbf{F}}(\hat{\mathbf{B}}^{(1)} + \hat{\Delta}\mathbf{B})^\top$.
12:       $\epsilon_{k,\lambda} \leftarrow \text{Loss}(\mathbf{X}^{(1)}[\boldsymbol{\Omega}_{\text{val}}], \mathbf{X}^{(2)}[\boldsymbol{\Omega}_{\text{val}}], \hat{\mathbf{X}}_{\text{val}}^{(1)}[\boldsymbol{\Omega}_{\text{val}}], \hat{\mathbf{X}}_{\text{val}}^{(2)}[\boldsymbol{\Omega}_{\text{val}}])$.
13:       $E_\lambda \leftarrow E_\lambda + \epsilon_{k,\lambda}$.
14:   **end for**
15: **end for**
16: **Selection and Final Fit:**
17:   Select optimal parameter: $\hat{\lambda} \leftarrow \arg\min_{\lambda \in \boldsymbol{\Lambda}_{\text{grid}}}(E_\lambda/3)$.
18:   Refit on full data:
19:   $\hat{\mathbf{F}}, \hat{\mathbf{B}}^{(1)}, \hat{\Delta}\mathbf{B} \leftarrow \text{SMART}(\mathbf{X}^{(1)}, \mathbf{X}^{(2)}, \hat{\lambda}, r)$ (or Structural Disentanglement with Naïve Lasso).
20: **Return:** optimal $\hat{\lambda}$ and final estimators $\hat{\mathbf{F}}, \hat{\mathbf{B}}^{(1)}, \hat{\Delta}\mathbf{B}$.

---

$C_1, C_2$, and $C_3$ are constants. Condition (C4) relaxes the independence assumption on the error terms $\varepsilon_{ij}$, thereby accommodating dependence structures among the observations $x_{ij}$. This flexibility is particularly useful when analyzing time series data.

Then we establish the convergence rates of the estimation error as follows.

**Theorem B.1.** *Under Conditions (C1) and (C4), and assume $r \geq r^*$, then*

$$\left\| \hat{\mathbf{M}}^{(k)} - \mathbf{M}^{(k)*} \right\|_F \lesssim n^{\frac{1}{2}} p^{\frac{1}{4}} \vee n^{\frac{1}{4}} p^{\frac{1}{2}} + \sqrt{s\lambda}$$

*with probability tending to one for $k \in \{1, 2\}$. Further assume Condition (C2) and $r = r^*$, then there exists an orthogonal rotation matrix $\mathbf{O}$ such that the following convergence rates hold with probability tending to one:*

$$\frac{1}{\sqrt{n}} \left\| \hat{\mathbf{F}}\mathbf{O} - \mathbf{F}^* \right\|_F \lesssim \frac{n^{\frac{1}{2}} p^{\frac{1}{4}} \vee n^{\frac{1}{4}} p^{\frac{1}{2}} + \sqrt{s\lambda}}{\sqrt{n}\sigma_{\mathbf{B}}},$$

$$\frac{1}{\sqrt{p}} \left\| \hat{\mathbf{B}}^{(k)}\mathbf{O} - \mathbf{B}^{(k)*} \right\|_F \lesssim \frac{n^{\frac{1}{2}} p^{\frac{1}{4}} \vee n^{\frac{1}{4}} p^{\frac{1}{2}} + \sqrt{s\lambda}}{\sqrt{n}\sigma_{\mathbf{B}}},$$

*and*

$$\frac{1}{\sqrt{s}} \left\| \hat{\Delta}\mathbf{B}\mathbf{O} - \Delta\mathbf{B}^* \right\|_F \lesssim \sqrt{\frac{p}{\sigma_{\mathbf{B}}^2}} \cdot \frac{n^{\frac{1}{2}} p^{\frac{1}{4}} \vee n^{\frac{1}{4}} p^{\frac{1}{2}} + \sqrt{s\lambda}}{\sqrt{ns}}.$$

Regarding the determination of the latent dimensionality, the information criterion approach proposed in the preceding section remains applicable.

Extending the one-step formulation to Poisson or binary likelihoods is feasible in principle, by replacing the squared loss with the corresponding negative log likelihood while keeping the row sparse group penalty. Our one-step theory, however, rests on additive noise and squared loss, and coupling a nonquadratic likelihood with the structural penalty would require redesigning both the optimization and the analysis. The two-stage design already decouples distribution specific recovery from attribution, so we leave the likelihood based one-step estimator to future work.

## C   Proof of Theoretical Guarantee

**Proof of Theorem 3.2:** By definition of the Maximum Likelihood Estimator (MLE), we obtain the following inequality

$$\left\|\tilde{\mathbf{M}}^{(1)} - \hat{\mathbf{F}}\left(\hat{\mathbf{B}}^{(1)}\right)^{\top}\right\|_{F}^{2} + \left\|\tilde{\mathbf{M}}^{(2)} - \hat{\mathbf{F}}\left(\hat{\mathbf{B}}^{(1)} + \Delta\hat{\mathbf{B}}\right)^{\top}\right\|_{F}^{2} + \lambda\sum_{j=1}^{p}\left\|\hat{\Delta\boldsymbol{b}}_{j}\right\|_{2}$$

$$\leq \left\|\tilde{\mathbf{M}}^{(1)} - \mathbf{F}^{*}\left(\mathbf{B}^{(1)*}\right)^{\top}\right\|_{F}^{2} + \left\|\tilde{\mathbf{M}}^{(2)} - \mathbf{F}^{*}\left(\mathbf{B}^{(1)*} + \Delta\mathbf{B}^{*}\right)^{\top}\right\|_{F}^{2} + \lambda\sum_{j=1}^{p}\left\|\Delta\boldsymbol{b}_{j}^{*}\right\|_{2}.$$

Rearranging the terms yields

$$0 \geq \left\|\tilde{\mathbf{M}}^{(1)} - \hat{\mathbf{F}}\left(\hat{\mathbf{B}}^{(1)}\right)^{\top}\right\|_{F}^{2} - \left\|\tilde{\mathbf{M}}^{(1)} - \mathbf{F}^{*}\left(\mathbf{B}^{(1)*}\right)^{\top}\right\|_{F}^{2}$$

$$+ \left\|\tilde{\mathbf{M}}^{(2)} - \hat{\mathbf{F}}\left(\hat{\mathbf{B}}^{(1)} + \Delta\hat{\mathbf{B}}\right)^{\top}\right\|_{F}^{2} - \left\|\tilde{\mathbf{M}}^{(2)} - \mathbf{F}^{*}\left(\mathbf{B}^{(1)^{*}} + \Delta\mathbf{B}^{*}\right)^{\top}\right\|_{F}^{2}$$

$$+ \lambda\sum_{j=1}^{p}\left(\left\|\hat{\Delta\boldsymbol{b}}_{j}\right\|_{2} - \left\|\Delta\boldsymbol{b}_{j}^{*}\right\|_{2}\right)$$

$$:=\mathbf{I} + \mathbf{II} + \mathbf{III}.$$

To bound the first term, **I**, we observe that

$$\left\|\tilde{\mathbf{M}}^{(1)} - \hat{\mathbf{F}}\left(\hat{\mathbf{B}}^{(1)}\right)^{\top}\right\|_{F}^{2} = \left\|\tilde{\mathbf{M}}^{(1)} - \mathbf{M}^{(1)*} + \mathbf{M}^{(1)*} - \hat{\mathbf{F}}\left(\hat{\mathbf{B}}^{(1)}\right)^{\top}\right\|_{F}^{2}$$

$$\geq \left\|\tilde{\mathbf{M}}^{(1)} - \mathbf{M}^{(1)*}\right\|_{F}^{2} + \left\|\mathbf{M}^{(1)*} - \hat{\mathbf{F}}\left(\hat{\mathbf{B}}^{(1)}\right)^{\top}\right\|_{F}^{2}$$

$$- 2\left\|\tilde{\mathbf{M}}^{(1)} - \mathbf{M}^{(1)*}\right\|_{F} \cdot \left\|\mathbf{M}^{(1)*} - \hat{\mathbf{F}}\left(\hat{\mathbf{B}}^{(1)}\right)^{\top}\right\|_{F}.$$

Consequently,

$$\mathbf{I} \geq \left\|\mathbf{M}^{(1)*} - \hat{\mathbf{F}}\left(\hat{\mathbf{B}}^{(1)}\right)^{\top}\right\|_{F}^{2} - 2\left\|\hat{\mathbf{M}}^{(1)} - \mathbf{M}^{(1)*}\right\|_{F} \cdot \left\|\mathbf{M}^{(1)*} - \hat{\mathbf{F}}\left(\hat{\mathbf{B}}^{(1)}\right)^{\top}\right\|_{F}. \tag{4}$$

Similarly, for the second term, **II**, we decompose it as

$$\left\|\tilde{\mathbf{M}}^{(2)} - \hat{\mathbf{F}}\left(\hat{\mathbf{B}}^{(1)} + \Delta\hat{\mathbf{B}}\right)^{\top}\right\|_{F}^{2} = \left\|\tilde{\mathbf{M}}^{(2)} - \mathbf{M}^{(2)*} + \mathbf{M}^{(2)*} - \hat{\mathbf{F}}\left(\hat{\mathbf{B}}^{(1)} + \Delta\hat{\mathbf{B}}\right)^{\top}\right\|_{F}^{2}$$

$$\geq \left\|\tilde{\mathbf{M}}^{(2)} - \mathbf{M}^{(2)*}\right\|_{F}^{2} + \left\|\mathbf{M}^{(2)*} - \hat{\mathbf{F}}\left(\hat{\mathbf{B}}^{(1)} + \Delta\hat{\mathbf{B}}\right)^{\top}\right\|_{F}^{2}$$

$$- 2\left\|\tilde{\mathbf{M}}^{(2)} - \mathbf{M}^{(2)*}\right\|_{F} \cdot \left\|\mathbf{M}^{(2)*} - \hat{\mathbf{F}}\left(\hat{\mathbf{B}}^{(1)} + \Delta\hat{\mathbf{B}}\right)^{\top}\right\|_{F}.$$

This leads to the bound

$$\mathbf{II} \geq \left\|\mathbf{M}^{(2)*} - \hat{\mathbf{F}}\left(\hat{\mathbf{B}}^{(1)} + \Delta\hat{\mathbf{B}}\right)^{\top}\right\|_{F}^{2} - 2\left\|\tilde{\mathbf{M}}^{(2)} - \mathbf{M}^{(2)*}\right\|_{F} \cdot \left\|\mathbf{M}^{(2)*} - \hat{\mathbf{F}}\left(\hat{\mathbf{B}}^{(1)} + \Delta\hat{\mathbf{B}}\right)^{\top}\right\|_{F}.$$

Recall that $\mathcal{S}^* = \left\{ j : \left\| \Delta \boldsymbol{b}_j^* \right\|_2 \neq 0 \right\}$, and $s = |\mathcal{S}^*|$. Then for term **III**, we derive the following bound

$$
\begin{aligned}
\mathbf{III} &= \lambda \sum_{j=1}^{p} \left( \left\| \Delta \hat{\boldsymbol{b}}_j \right\|_2 - \left\| \Delta \boldsymbol{b}_j^* \right\|_2 \right) \\
&= \lambda \left\{ \sum_{j \in \mathcal{S}^*} \left( \left\| \Delta \hat{\boldsymbol{b}}_j \right\|_2 - \left\| \Delta \boldsymbol{b}_j^* \right\|_2 \right) + \sum_{j \notin \mathcal{S}^*} \left\| \Delta \hat{\boldsymbol{b}}_j \right\|_2 \right\} \\
&\geq \lambda \sum_{j \in \mathcal{S}^*} \left( \left\| \Delta \hat{\boldsymbol{b}}_j \right\|_2 - \left\| \Delta \boldsymbol{b}_j^* \right\|_2 \right) \\
&\geq -\lambda \sum_{j \in \mathcal{S}^*} \left\| \Delta \hat{\boldsymbol{b}}_j - \Delta \boldsymbol{b}_j^* \right\|_2 \\
&\geq -\lambda \sqrt{s} \left\| \Delta \hat{\mathbf{B}}_{\mathcal{S}^*} - \Delta \mathbf{B}^* \right\|_F,
\end{aligned}
\tag{5}
$$

where $\Delta \hat{\mathbf{B}}_{\mathcal{S}^*}$ represents a matrix in which only the rows from the set $\mathcal{S}^*$ in the original $\Delta \hat{\mathbf{B}}$ matrix are non-zero, with all other rows set to zero.

Thus we can bound this term by relying on the boundedness of the parameters. Specifically, by assuming that $\|\boldsymbol{b}_j^{(1)*}\|_2 \leq C$ and $\|\boldsymbol{b}_j^{(2)*}\|_2 \leq C$ in Condition (C1), it follows that $\|\Delta \boldsymbol{b}_j^*\|_2 = \|\boldsymbol{b}_j^{(2)*} - \boldsymbol{b}_j^{(1)*}\|_2 \leq 2C$. By the same logic, the boundedness of the estimates implies $\|\Delta \hat{\boldsymbol{b}}_j\| \leq \|\hat{\boldsymbol{b}}_j^{(2)} - \hat{\boldsymbol{b}}_j^{(1)}\| \leq \|\hat{\boldsymbol{b}}_j^{(2)}\| + \|\hat{\boldsymbol{b}}_j^{(1)}\| \leq 2C$.

Thus, we have

$$
\mathbf{III} \geq -2Cs\lambda.
$$

Finally, combining the results for **I**, **II**, and **III**, we conclude that

$$
\begin{aligned}
0 \geq \mathbf{I} + \mathbf{II} + \mathbf{III} \geq{}& \left\| \mathbf{M}^{(1)*} - \hat{\mathbf{F}} \left( \hat{B}^{(1)} \right)^\top \right\|_F^2 + \left\| \mathbf{M}^{(2)*} - \hat{\mathbf{F}} \left( \hat{\mathbf{B}}^{(1)} + \Delta \hat{\mathbf{B}} \right)^\top \right\|_F^2 \\
&- 2 \left\| \tilde{\mathbf{M}}^{(1)} - \mathbf{M}^{(1)*} \right\|_F \cdot \left\| \mathbf{M}^{(1)*} - \hat{\mathbf{F}} \left( \hat{\mathbf{B}}^{(1)} \right)^\top \right\|_F \\
&- 2 \left\| \tilde{\mathbf{M}}^{(2)} - \mathbf{M}^{(2)*} \right\|_F \cdot \left\| \mathbf{M}^{(2)*} - \hat{\mathbf{F}} \left( \hat{\mathbf{B}}^{(1)} + \Delta \hat{\mathbf{B}} \right)^\top \right\|_F \\
&- 2Cs\lambda.
\end{aligned}
$$

Rearranging the terms from the above inequality and applying the Cauchy-Schwarz inequality yields the following upper bound:

$$
\begin{aligned}
& \left\| \mathbf{M}^{(1)*} - \hat{\mathbf{F}} \left( \hat{\mathbf{B}}^{(1)} \right)^\top \right\|_F^2 + \left\| \mathbf{M}^{(2)*} - \hat{\mathbf{F}} \left( \hat{\mathbf{B}}^{(1)} + \Delta \hat{\mathbf{B}} \right)^\top \right\|_F^2 \\
\leq{}& 2 \left\| \tilde{\mathbf{M}}^{(1)} - \mathbf{M}^{(1)*} \right\|_F \cdot \left\| \mathbf{M}^{(1)*} - \hat{\mathbf{F}} \left( \hat{\mathbf{B}}^{(1)} \right)^\top \right\|_F \\
& + 2 \left\| \tilde{\mathbf{M}}^{(2)} - \mathbf{M}^{(2)*} \right\|_F \cdot \left\| \mathbf{M}^{(2)*} - \hat{\mathbf{F}} \left( \hat{\mathbf{B}}^{(1)} + \Delta \hat{\mathbf{B}} \right)^\top \right\|_F + 2Cs\lambda \\
\leq{}& 2 \left( \left\| \tilde{\mathbf{M}}^{(1)} - \mathbf{M}^{(1)*} \right\|_F^2 + \left\| \tilde{\mathbf{M}}^{(2)} - \mathbf{M}^{(2)*} \right\|_F^2 \right)^{\frac{1}{2}} \\
& \cdot \left( \left\| \mathbf{M}^{(1)*} - \hat{\mathbf{F}} \left( \hat{\mathbf{B}}^{(1)} \right)^\top \right\|_F^2 + \left\| \mathbf{M}^{(2)*} - \hat{\mathbf{F}} \left( \hat{\mathbf{B}}^{(1)} + \Delta \hat{\mathbf{B}} \right)^\top \right\|_F^2 \right)^{\frac{1}{2}} + 2Cs\lambda.
\end{aligned}
$$

Based on this algebraic structure, it implies that either

$$
\begin{aligned}
& \left\| \mathbf{M}^{(1)*} - \hat{\mathbf{F}} \left( \hat{\mathbf{B}}^{(1)} \right)^\top \right\|_F^2 + \left\| \mathbf{M}^{(2)*} - \hat{\mathbf{F}} \left( \hat{\mathbf{B}}^{(1)} + \Delta \hat{\mathbf{B}} \right)^\top \right\|_F^2 \\
\leq{}& 16 \left( \left\| \tilde{\mathbf{M}}^{(1)} - \mathbf{M}^{(1)*} \right\|_F^2 + \left\| \tilde{\mathbf{M}}^{(2)} - \mathbf{M}^{(2)*} \right\|_F^2 \right)
\end{aligned}
\tag{6}
$$

or

$$\left\| \mathbf{M}^{(1)*} - \hat{\mathbf{F}} \left( \hat{\mathbf{B}}^{(1)} \right)^{\top} \right\|_F^2 + \left\| \mathbf{M}^{(2)*} - \hat{\mathbf{F}} \left( \hat{\mathbf{B}}^{(1)} + \hat{\Delta \mathbf{B}} \right)^{\top} \right\|_F^2 \leq 4Cs\lambda. \tag{7}$$

Furthermore, utilizing the elementary inequality $(a + b)^2 \leq 2(a^2 + b^2)$, we observe that

$$\frac{1}{2} \left( \left\| \mathbf{M}^{(1)*} - \hat{\mathbf{F}} \left( \hat{\mathbf{B}}^{(1)} \right)^{\top} \right\|_F + \left\| \mathbf{M}^{(2)*} - \hat{\mathbf{F}} \left( \hat{\mathbf{B}}^{(1)} + \hat{\Delta \mathbf{B}} \right)^{\top} \right\|_F \right)^2$$
$$\leq \left\| \mathbf{M}^{(1)*} - \hat{\mathbf{F}} \left( \hat{\mathbf{B}}^{(1)} \right)^{\top} \right\|_F^2 + \left\| \mathbf{M}^{(2)*} - \hat{\mathbf{F}} \left( \hat{\mathbf{B}}^{(1)} + \hat{\Delta \mathbf{B}} \right)^{\top} \right\|_F^2. \tag{8}$$

Combining Equations (6) and (7) with Equation (8) leads to the following bounds on the sum of the Frobenius norms

$$\left\| \mathbf{M}^{(1)*} - \hat{\mathbf{F}} \left( \hat{\mathbf{B}}^{(1)} \right)^{\top} \right\|_F + \left\| \mathbf{M}^{(2)*} - \hat{\mathbf{F}} \left( \hat{\mathbf{B}}^{(1)} + \hat{\Delta \mathbf{B}} \right)^{\top} \right\|_F$$
$$\leq 4\sqrt{2} \left( \left\| \tilde{\mathbf{M}}^{(1)} - \mathbf{M}^{(1)*} \right\|_F^2 + \left\| \tilde{\mathbf{M}}^{(2)} - \mathbf{M}^{(2)*} \right\|_F^2 \right)^{\frac{1}{2}}$$

or

$$\left\| \mathbf{M}^{(1)*} - \hat{\mathbf{F}} \left( \hat{\mathbf{B}}^{(1)} \right)^{\top} \right\|_F + \left\| \mathbf{M}^{(2)*} - \hat{\mathbf{F}} \left( \hat{\mathbf{B}}^{(1)} + \hat{\Delta \mathbf{B}} \right)^{\top} \right\|_F \leq 2\sqrt{2Cs\lambda}.$$

We now incorporate the error from the first stage. Assuming that

$$\left\| \tilde{\mathbf{M}}^{(1)} - \mathbf{M}^{(1)*} \right\|_F + \left\| \tilde{\mathbf{M}}^{(2)} - \mathbf{M}^{(2)*} \right\|_F \lesssim err$$

with probability tending to one, it follows that

$$\left\| \mathbf{M}^{(1)*} - \hat{\mathbf{F}} \left( \hat{\mathbf{B}}^{(1)} \right)^{\top} \right\|_F + \left\| \mathbf{M}^{(2)*} - \hat{\mathbf{F}} \left( \hat{\mathbf{B}}^{(1)} + \hat{\Delta \mathbf{B}} \right)^{\top} \right\|_F \lesssim err + \sqrt{s\lambda}$$

with probability tending to one. Consequently, we obtain the individual consistency results

$$\left\| \mathbf{M}^{(1)*} - \hat{\mathbf{F}} \left( \hat{\mathbf{B}}^{(1)} \right)^{\top} \right\|_F \lesssim err + \sqrt{s\lambda}$$
$$\left\| \mathbf{M}^{(2)*} - \hat{\mathbf{F}} \left( \hat{\mathbf{B}}^{(2)} \right)^{\top} \right\|_F \lesssim err + \sqrt{s\lambda}. \tag{9}$$

with probability tending to one. $\square$

**Proof of Theorem 3.3:** Assuming the rank is correctly specified (i.e., $r = r^*$), we invoke Theorem 3 of Yu et al. (2015). This ensures the existence of an orthogonal rotation matrix $\mathbf{O}$ such that the estimation error of the factor matrix satisfies

$$\frac{1}{\sqrt{n}} \left\| \hat{\mathbf{F}} \mathbf{O} - \mathbf{F}^* \right\|_F \lesssim \frac{\sigma_1 \left( \mathbf{M}^{(1)*} \right) \left\| \mathbf{M}^{(1)*} - \hat{\mathbf{F}} \hat{\mathbf{B}}^{(1)} \right\|_F}{\sigma_{r^*}^2 \left( \mathbf{M}^{(1)*} \right)}$$
$$\lesssim \frac{err + \sqrt{s\lambda}}{\sqrt{n}\sigma_{\mathbf{B}}}.$$

The validity of this bound is contingent on the Condition (C2) regarding the singular values and signal strength.

By decomposing the error term and applying the triangle inequality,

$$\left\| \mathbf{M}^{(1)*} - \hat{\mathbf{F}} \left( \hat{\mathbf{B}}^{(1)} \right)^{\top} \right\|_F = \left\| \mathbf{F}^* \left( \mathbf{B}^{(1)*} \right)^{\top} - \hat{\mathbf{F}} \left( \hat{\mathbf{B}}^{(1)} \right)^{\top} \right\|_F$$
$$= \left\| \left( \mathbf{F}^* - \hat{\mathbf{F}} \mathbf{O} \right) \left( \mathbf{B}^{(1)*} \right)^{\top} + \hat{\mathbf{F}} \mathbf{O} \left( \mathbf{B}^{(1)*} \right)^{\top} - \hat{\mathbf{F}} \left( \hat{\mathbf{B}}^{(1)} \right)^{\top} \right\|_F$$
$$\geq \left\| \hat{\mathbf{F}} \left( \mathbf{B}^{(1)*} \mathbf{O}^{\top} - \hat{\mathbf{B}}^{(1)} \right)^{\top} \right\|_F - \left\| \left( \mathbf{F}^* - \hat{\mathbf{F}} \mathbf{O} \right) \left( \mathbf{B}^{(1)*} \right)^{\top} \right\|_F.$$

Rearranging these terms provides the following inequality

$$\left\| \hat{\mathbf{F}} \left( \mathbf{B}^{(1)*} \mathbf{O}^\top - \hat{\mathbf{B}}^{(1)} \right)^\top \right\|_F \le \left\| \mathbf{M}^{(1)*} - \hat{\mathbf{F}} \left( \hat{\mathbf{B}}^{(1)} \right)^\top \right\|_F + \left\| \left( \mathbf{F}^* - \hat{\mathbf{F}} \mathbf{O} \right) \left( \mathbf{B}^{(1)*} \right)^\top \right\|_F . \tag{10}$$

Furthermore, under the identification constraints $\hat{\mathbf{F}}^\top \hat{\mathbf{F}} = n \mathbf{I}_r$, the term on the left-hand side of Equation (10) can be simplified as

$$\left\| \hat{\mathbf{F}} \left( \mathbf{B}^{(1)*} \mathbf{O}^\top - \hat{\mathbf{B}}^{(1)} \right)^\top \right\|_F = \mathrm{tr} \left\{ \left( \mathbf{B}^{(1)*} \mathbf{O}^\top - \hat{\mathbf{B}}^{(1)} \right)^\top \hat{\mathbf{F}}^\top \hat{\mathbf{F}} \left( \mathbf{B}^{(1)*} \mathbf{O}^\top - \hat{\mathbf{B}}^{(1)} \right)^\top \right\}$$

$$= n \left\| \hat{\mathbf{B}}^{(1)} \mathbf{O} - \mathbf{B}^{(1)*} \right\|_F^2 .$$

Additionally, the second term on the right-hand side of Equation (10) is bounded by

$$\left\| \left( \mathbf{F}^* - \hat{\mathbf{F}} \mathbf{O} \right) \left( \mathbf{B}^{(1)*} \right)^\top \right\|_F \le \left\| \mathbf{F}^* - \hat{\mathbf{F}} \mathbf{O} \right\|_F \left\| \mathbf{B}^{(1)*} \right\|_F \lesssim \sqrt{p} \left\| \mathbf{F}^* - \hat{\mathbf{F}} \mathbf{O} \right\|_F .$$

Substituting these results back into the inequality and normalizing by $\sqrt{p}$, we derive the convergence rate for the loading matrix $\hat{\mathbf{B}}^{(1)}$

$$\frac{1}{\sqrt{p}} \left\| \hat{\mathbf{B}}^{(1)} \mathbf{O} - \mathbf{B}^{(1)*} \right\|_F \lesssim \frac{1}{\sqrt{n}} \left\| \hat{\mathbf{F}} \mathbf{O} - \mathbf{F}^* \right\|_F + \frac{1}{\sqrt{np}} \left\| \mathbf{M}^{(1)*} - \hat{\mathbf{F}} \left( \hat{\mathbf{B}}^{(1)} \right)^\top \right\|_F$$

$$\lesssim \frac{err + \sqrt{s\lambda}}{\sqrt{n} \sigma_\mathbf{B}} + \frac{err + \sqrt{s\lambda}}{\sqrt{np}}$$

$$\lesssim \frac{err + \sqrt{s\lambda}}{\sqrt{n} \sigma_\mathbf{B}} .$$

By an analogous argument, the same bound holds for $\hat{\mathbf{B}}^{(2)}$

$$\frac{1}{\sqrt{p}} \left\| \hat{\mathbf{B}}^{(2)} \mathbf{O} - \mathbf{B}^{(2)*} \right\|_F \lesssim \frac{err + \sqrt{s\lambda}}{\sqrt{n} \sigma_\mathbf{B}} .$$

Consequently, the error bound for the difference $\hat{\Delta \mathbf{B}} = \hat{\mathbf{B}}^{(2)} - \hat{\mathbf{B}}^{(1)}$ is given by

$$\frac{1}{\sqrt{p}} \left\| \hat{\Delta \mathbf{B}} \mathbf{O} - \Delta \mathbf{B}^* \right\|_F \lesssim \frac{err + \sqrt{s\lambda}}{\sqrt{n} \sigma_\mathbf{B}} .$$

Finally, adjusting the normalization for the sparsity level $s$, we conclude

$$\frac{1}{\sqrt{s}} \left\| \hat{\Delta \mathbf{B}} \mathbf{O} - \Delta \mathbf{B}^* \right\|_F \lesssim \sqrt{\frac{p}{\sigma_\mathbf{B}^2}} \cdot \frac{err + \sqrt{s\lambda}}{\sqrt{ns}} .$$

Thus the proof is finished. $\qquad \square$

**Proof of Corollary 3.4:** By invoking Theorem 1 in Chen & Li (2022), we have

$$\left\| \tilde{\mathbf{M}}^{(1)} - \mathbf{M}^{(1)*} \right\|_F + \left\| \tilde{\mathbf{M}}^{(2)} - \mathbf{M}^{(2)*} \right\|_F \lesssim \sqrt{n} \vee \sqrt{p}$$

with probability tending to one. This, along with $s\lambda \ll n \vee p$, confirms that the requisite error bounds for $\hat{\mathbf{M}}^{(1)}$ and $\hat{\mathbf{M}}^{(2)}$ holds. $\qquad \square$

**Proof of Corollary 3.5:** Based on Theorem 3.3, under the conditions that $s\lambda \ll n \vee p$ and $\sigma_\mathbf{B} \asymp \sqrt{p}$, we observe that

$$\frac{s\lambda}{n \sigma_\mathbf{B}^2} \asymp \frac{s\lambda}{np} \ll \frac{1}{n \wedge p} .$$

Consequently, the final term in Equation (9) is dominated by the estimation error, and the proof is complete. $\square$

We next prove the support recovery results. Throughout this part, $\hat{\mathbf{F}}$ denotes the factor component of the global minimizer of Equation (1) over $\mathcal{F}(C)$ at the same tuning parameter $\lambda$, so that $n^{-1}\hat{\mathbf{F}}^\top\hat{\mathbf{F}} = \mathbf{I}_r$ and Theorem 3.3 applies to it. We write $\tilde{\boldsymbol{m}}_j^{(k)}$ and $\boldsymbol{m}_j^{(k)*} = \mathbf{F}^*\boldsymbol{b}_j^{(k)*}$ for the $j$-th columns of $\tilde{\mathbf{M}}^{(k)}$ and $\mathbf{M}^{(k)*} = \mathbf{F}^*(\mathbf{B}^{(k)*})^\top$, respectively, and recall the notation of Theorem 3.6: the difference noise $\boldsymbol{e}_{\Delta,j} = (\tilde{\boldsymbol{m}}_j^{(2)} - \boldsymbol{m}_j^{(2)*}) - (\tilde{\boldsymbol{m}}_j^{(1)} - \boldsymbol{m}_j^{(1)*})$, the primitive quantities $\rho_{n,p} = \max_{1\le j\le p}\|n^{-1}(\mathbf{F}^*)^\top\boldsymbol{e}_{\Delta,j}\|_2$ and $\kappa_{n,p} = \max_{1\le j\le p} n^{-1/2}\|\boldsymbol{e}_{\Delta,j}\|_2$, both of which are $O_p(\rho_n)$ and $O_p(1)$ respectively under Condition (C3) by the triangle inequality, the screening statistics $\hat{\boldsymbol{z}}_j = n^{-1}\hat{\mathbf{F}}^\top(\tilde{\boldsymbol{m}}_j^{(2)} - \tilde{\boldsymbol{m}}_j^{(1)})$, the group soft-thresholding operator $\mathcal{T}_\tau(\boldsymbol{z}) = (1 - \tau/\|\boldsymbol{z}\|_2)_+\boldsymbol{z}$, $\mathcal{T}_\tau(\boldsymbol{0}) = \boldsymbol{0}$, where $(x)_+ = \max(x,0)$, the shift rows $\hat{\Delta}\boldsymbol{b}_j$ of the estimator of Equation (1) and the induced support $\hat{\mathcal{S}} = \{j : \|\hat{\Delta}\boldsymbol{b}_j\|_2 > 0\}$, the threshold level $\tau_n = \lambda/n$, the minimal signal $\beta_{\min} = \min_{j\in\mathcal{S}^*}\|\Delta\boldsymbol{b}_j^*\|_2$, and $s = |\mathcal{S}^*|$. The sequences $\lambda$, $err$, and $\rho_n$ are fixed.

**Lemma C.1.** *Let $\mathbf{F} \in \mathbb{R}^{n\times r}$ satisfy $n^{-1}\mathbf{F}^\top\mathbf{F} = \mathbf{I}_r$, let $\boldsymbol{y}_1, \boldsymbol{y}_2 \in \mathbb{R}^n$, and let $\lambda > 0$. Consider*

$$L(\boldsymbol{b},\boldsymbol{d}) = \|\boldsymbol{y}_1 - \mathbf{F}\boldsymbol{b}\|_2^2 + \|\boldsymbol{y}_2 - \mathbf{F}(\boldsymbol{b}+\boldsymbol{d})\|_2^2 + \lambda\|\boldsymbol{d}\|_2, \qquad \boldsymbol{b},\boldsymbol{d}\in\mathbb{R}^r. \tag{11}$$

*Then $L$ has a unique minimizer $(\hat{\boldsymbol{b}},\hat{\boldsymbol{d}})$, given by*

$$\hat{\boldsymbol{d}} = \mathcal{T}_{\lambda/n}(\boldsymbol{z}), \qquad \hat{\boldsymbol{b}} = \frac{1}{2}\left(\boldsymbol{u}_1 + \boldsymbol{u}_2 - \hat{\boldsymbol{d}}\right),$$

*where $\boldsymbol{u}_k = n^{-1}\mathbf{F}^\top\boldsymbol{y}_k$ for $k \in \{1,2\}$ and $\boldsymbol{z} = \boldsymbol{u}_2 - \boldsymbol{u}_1$. In particular, $\hat{\boldsymbol{d}} = \boldsymbol{0}$ if and only if $\|\boldsymbol{z}\|_2 \le \lambda/n$.*

**Proof of Lemma C.1:** Let $\Pi_{\mathbf{F}} = n^{-1}\mathbf{F}\mathbf{F}^\top$ denote the orthogonal projection onto the column space of $\mathbf{F}$. For any $\boldsymbol{y} \in \mathbb{R}^n$ and $\boldsymbol{v} \in \mathbb{R}^r$, using $\mathbf{F}^\top\mathbf{F} = n\mathbf{I}_r$,

$$\|\boldsymbol{y} - \mathbf{F}\boldsymbol{v}\|_2^2 = \|\boldsymbol{y} - \Pi_{\mathbf{F}}\boldsymbol{y}\|_2^2 + \left\|\mathbf{F}\left(n^{-1}\mathbf{F}^\top\boldsymbol{y} - \boldsymbol{v}\right)\right\|_2^2 = \|\boldsymbol{y} - \Pi_{\mathbf{F}}\boldsymbol{y}\|_2^2 + n\left\|n^{-1}\mathbf{F}^\top\boldsymbol{y} - \boldsymbol{v}\right\|_2^2.$$

Applying this identity to both quadratic terms of $L$ yields

$$L(\boldsymbol{b},\boldsymbol{d}) = R + n\|\boldsymbol{u}_1 - \boldsymbol{b}\|_2^2 + n\|\boldsymbol{u}_2 - \boldsymbol{b} - \boldsymbol{d}\|_2^2 + \lambda\|\boldsymbol{d}\|_2, \tag{12}$$

where $R$ collects terms free of $(\boldsymbol{b},\boldsymbol{d})$. The quadratic part of Equation (12) is strictly convex in $(\boldsymbol{b},\boldsymbol{d})$, and adding the convex penalty $\lambda\|\boldsymbol{d}\|_2$ preserves strict convexity, so the minimizer of $L$ is unique. Minimizing Equation (12) over $\boldsymbol{b}$ for fixed $\boldsymbol{d}$ gives $\boldsymbol{b}(\boldsymbol{d}) = (\boldsymbol{u}_1 + \boldsymbol{u}_2 - \boldsymbol{d})/2$, whence $\boldsymbol{u}_1 - \boldsymbol{b}(\boldsymbol{d}) = (\boldsymbol{d} - \boldsymbol{z})/2$ and $\boldsymbol{u}_2 - \boldsymbol{b}(\boldsymbol{d}) - \boldsymbol{d} = (\boldsymbol{z} - \boldsymbol{d})/2$. Substituting back, the profiled objective becomes

$$\frac{n}{2}\|\boldsymbol{d} - \boldsymbol{z}\|_2^2 + \lambda\|\boldsymbol{d}\|_2.$$

The subgradient optimality condition reads $\boldsymbol{0} \in n(\hat{\boldsymbol{d}} - \boldsymbol{z}) + \lambda\,\partial\|\hat{\boldsymbol{d}}\|_2$. If $\|\boldsymbol{z}\|_2 \le \lambda/n$, then $\hat{\boldsymbol{d}} = \boldsymbol{0}$ satisfies this condition, because $\|n\boldsymbol{z}\|_2 \le \lambda$ and $\partial\|\boldsymbol{0}\|_2 = \{\boldsymbol{g} : \|\boldsymbol{g}\|_2 \le 1\}$. If $\|\boldsymbol{z}\|_2 > \lambda/n$, then $\hat{\boldsymbol{d}} \ne \boldsymbol{0}$ (otherwise the condition fails), and $n(\hat{\boldsymbol{d}} - \boldsymbol{z}) + \lambda\hat{\boldsymbol{d}}/\|\hat{\boldsymbol{d}}\|_2 = \boldsymbol{0}$ forces $\hat{\boldsymbol{d}}$ to be proportional to $\boldsymbol{z}$ with $\|\hat{\boldsymbol{d}}\|_2 = \|\boldsymbol{z}\|_2 - \lambda/n$, that is, $\hat{\boldsymbol{d}} = \{1 - \lambda/(n\|\boldsymbol{z}\|_2)\}\boldsymbol{z}$. In both cases $\hat{\boldsymbol{d}} = \mathcal{T}_{\lambda/n}(\boldsymbol{z})$, and uniqueness completes the proof. $\square$

Because the squared Frobenius norm decomposes over columns and the penalty $\|\Delta\mathbf{B}\|_{2,1}$ decomposes over rows, for fixed $\mathbf{F} = \hat{\mathbf{F}}$ the Stage II objective Equation (1) satisfies

$$\mathcal{Q}\left(\hat{\mathbf{F}}, \mathbf{B}^{(1)}, \Delta\mathbf{B}; \tilde{\mathbf{M}}^{(1)}, \tilde{\mathbf{M}}^{(2)}\right) = \sum_{j=1}^p \left\{\left\|\tilde{\boldsymbol{m}}_j^{(1)} - \hat{\mathbf{F}}\boldsymbol{b}_j^{(1)}\right\|_2^2 + \left\|\tilde{\boldsymbol{m}}_j^{(2)} - \hat{\mathbf{F}}\left(\boldsymbol{b}_j^{(1)} + \Delta\boldsymbol{b}_j\right)\right\|_2^2 + \lambda\|\Delta\boldsymbol{b}_j\|_2\right\}.$$

Each summand depends only on $(\boldsymbol{b}_j^{(1)}, \Delta\boldsymbol{b}_j)$ and has the form Equation (11) with $\mathbf{F} = \hat{\mathbf{F}}$, $\boldsymbol{y}_1 = \tilde{\boldsymbol{m}}_j^{(1)}$, and $\boldsymbol{y}_2 = \tilde{\boldsymbol{m}}_j^{(2)}$. Lemma C.1 therefore shows that the thresholded differences $\hat{\Delta}\boldsymbol{b}_j = \mathcal{T}_{\lambda/n}(\hat{\boldsymbol{z}}_j)$ constitute the difference component of the unique minimizer of the Stage II loading objective given $\hat{\mathbf{F}}$ whenever the norm constraints of $\mathcal{F}(C)$ are not binding. The proof of Theorem 3.6 verifies that the constraints are slack with probability tending to one, so this characterization applies to the shift component of the estimator of Equation (1) and

$$\hat{\mathcal{S}} = \left\{j : \|\hat{\boldsymbol{z}}_j\|_2 > \lambda/n\right\}.$$

**Lemma C.2.** *For any orthogonal $\mathbf{O} \in \mathbb{R}^{r \times r}$, any $\boldsymbol{z} \in \mathbb{R}^r$, and any $\tau > 0$, it holds that $\|\mathbf{O}^\top \boldsymbol{z}\|_2 = \|\boldsymbol{z}\|_2$ and $\mathcal{T}_\tau(\mathbf{O}^\top \boldsymbol{z}) = \mathbf{O}^\top \mathcal{T}_\tau(\boldsymbol{z})$. Consequently, $\hat{\mathcal{S}} = \{j : \|\hat{\boldsymbol{z}}_j\|_2 > \lambda/n\}$ is invariant under orthogonal rotations of $\hat{\mathbf{F}}$.*

**Proof of Lemma C.2:** Orthogonal matrices preserve the Euclidean norm, and $\mathcal{T}_\tau$ multiplies its argument by the scalar factor $(1 - \tau/\|\cdot\|_2)_+$, which depends on the argument only through its norm. Replacing $\hat{\mathbf{F}}$ by $\hat{\mathbf{F}}\mathbf{O}$ replaces $\hat{\boldsymbol{z}}_j$ by $\mathbf{O}^\top \hat{\boldsymbol{z}}_j$, leaving each norm $\|\hat{\boldsymbol{z}}_j\|_2$, and hence each membership decision, unchanged. □

**Lemma C.3.** *Let $\mathbf{O} \in \mathbb{R}^{r \times r}$ be any orthogonal matrix and set $\bar{\mathbf{F}} = \hat{\mathbf{F}}\mathbf{O}$, so that $\bar{\mathbf{F}}^\top \bar{\mathbf{F}} = n\mathbf{I}_r$ and $\|\bar{\mathbf{F}}\|_2 = \sqrt{n}$. Define*

$$\mathbf{A}_n = n^{-1}\bar{\mathbf{F}}^\top \mathbf{F}^*, \qquad \boldsymbol{w}_j = n^{-1}\bar{\mathbf{F}}^\top \boldsymbol{e}_{\Delta,j}, \qquad \delta_{F,n} = n^{-1/2}\left\|\hat{\mathbf{F}}\mathbf{O} - \mathbf{F}^*\right\|_F, \qquad \zeta_{n,p} = \max_{1 \le j \le p} \|\boldsymbol{w}_j\|_2.$$

*Then, for every $j$,*

$$\mathbf{O}^\top \hat{\boldsymbol{z}}_j = \mathbf{A}_n \Delta \boldsymbol{b}_j^* + \boldsymbol{w}_j, \tag{13}$$

*and, deterministically,*

$$\zeta_{n,p} \le \rho_{n,p} + \delta_{F,n}\, \kappa_{n,p}. \tag{14}$$

**Proof of Lemma C.3:** Since $\mathbf{M}^{(k)*} = \mathbf{F}^*(\mathbf{B}^{(k)*})^\top$, the $j$-th column of $\mathbf{M}^{(k)*}$ equals $\mathbf{F}^* \boldsymbol{b}_j^{(k)*}$, and hence

$$\tilde{\boldsymbol{m}}_j^{(2)} - \tilde{\boldsymbol{m}}_j^{(1)} = \mathbf{F}^* \Delta \boldsymbol{b}_j^* + \boldsymbol{e}_{\Delta,j}.$$

Therefore $\mathbf{O}^\top \hat{\boldsymbol{z}}_j = n^{-1}\mathbf{O}^\top \hat{\mathbf{F}}^\top (\mathbf{F}^* \Delta \boldsymbol{b}_j^* + \boldsymbol{e}_{\Delta,j}) = \mathbf{A}_n \Delta \boldsymbol{b}_j^* + \boldsymbol{w}_j$, which is Equation (13). For Equation (14), decompose $\boldsymbol{w}_j = n^{-1}(\mathbf{F}^*)^\top \boldsymbol{e}_{\Delta,j} + n^{-1}(\bar{\mathbf{F}} - \mathbf{F}^*)^\top \boldsymbol{e}_{\Delta,j}$. The first term is bounded by $\rho_{n,p}$ uniformly in $j$. For the second term,

$$n^{-1}\left\|(\bar{\mathbf{F}} - \mathbf{F}^*)^\top \boldsymbol{e}_{\Delta,j}\right\|_2 \le n^{-1}\left\|\bar{\mathbf{F}} - \mathbf{F}^*\right\|_2 \|\boldsymbol{e}_{\Delta,j}\|_2 \le n^{-1}\left\|\bar{\mathbf{F}} - \mathbf{F}^*\right\|_F \|\boldsymbol{e}_{\Delta,j}\|_2$$
$$= \delta_{F,n} \cdot n^{-1/2}\|\boldsymbol{e}_{\Delta,j}\|_2 \le \delta_{F,n}\, \kappa_{n,p}.$$

Since these bounds hold simultaneously for every $j$, taking the maximum over $j$ yields Equation (14); in particular, no union bound over $j$ is incurred at this step. □

**Lemma C.4.** *With the notation of Lemma C.3, $\|\mathbf{A}_n - \mathbf{I}_r\|_2 \le \delta_{F,n}$, and consequently $\left\|\mathbf{A}_n \Delta \boldsymbol{b}_j^*\right\|_2 \ge (1 - \delta_{F,n})\left\|\Delta \boldsymbol{b}_j^*\right\|_2$ for every $j$.*

**Proof of Lemma C.4:** Using $\bar{\mathbf{F}}^\top \bar{\mathbf{F}} = n\mathbf{I}_r$, we may write $\mathbf{A}_n - \mathbf{I}_r = n^{-1}\bar{\mathbf{F}}^\top (\mathbf{F}^* - \bar{\mathbf{F}})$, so that

$$\|\mathbf{A}_n - \mathbf{I}_r\|_2 \le n^{-1}\left\|\bar{\mathbf{F}}\right\|_2 \left\|\mathbf{F}^* - \bar{\mathbf{F}}\right\|_2 \le n^{-1}\sqrt{n}\left\|\mathbf{F}^* - \bar{\mathbf{F}}\right\|_F = \delta_{F,n}.$$

The second claim follows from the triangle inequality: $\|\mathbf{A}_n \Delta \boldsymbol{b}_j^*\|_2 \ge \|\Delta \boldsymbol{b}_j^*\|_2 - \|(\mathbf{A}_n - \mathbf{I}_r)\Delta \boldsymbol{b}_j^*\|_2 \ge (1 - \delta_{F,n})\|\Delta \boldsymbol{b}_j^*\|_2$. □

**Proposition C.5.** *Let $0 \le c_F < 1$ and $q \ge 0$. On any realization for which $\delta_{F,n} \le c_F$ and $\zeta_{n,p} \le q$, and provided that*

$$q \le \frac{\lambda}{n} < (1 - c_F)\beta_{\min} - q,$$

*it holds that $\hat{\mathcal{S}} = \mathcal{S}^*$.*

**Proof of Proposition C.5:** By Lemma C.2, $\|\hat{\boldsymbol{z}}_j\|_2 = \|\mathbf{O}^\top \hat{\boldsymbol{z}}_j\|_2$ for every $j$, so membership in $\hat{\mathcal{S}}$ may be checked after rotation. First let $j \notin \mathcal{S}^*$, so that $\Delta \boldsymbol{b}_j^* = \mathbf{0}$. By Equation (13),

$$\|\hat{\boldsymbol{z}}_j\|_2 = \|\boldsymbol{w}_j\|_2 \le \zeta_{n,p} \le q \le \frac{\lambda}{n},$$

and Lemma C.1 gives $\hat{\Delta}\boldsymbol{b}_j = \mathcal{T}_{\lambda/n}(\hat{\boldsymbol{z}}_j) = \mathbf{0}$; note that the boundary case $\|\hat{\boldsymbol{z}}_j\|_2 = \lambda/n$ also yields the zero vector. Hence $j \notin \hat{\mathcal{S}}$. Next let $j \in \mathcal{S}^*$. By Equation (13), Lemma C.4, and the definition of $\beta_{\min}$,

$$\|\hat{\boldsymbol{z}}_j\|_2 \ge \left\|\mathbf{A}_n \Delta \boldsymbol{b}_j^*\right\|_2 - \|\boldsymbol{w}_j\|_2 \ge (1 - c_F)\left\|\Delta \boldsymbol{b}_j^*\right\|_2 - q \ge (1 - c_F)\beta_{\min} - q > \frac{\lambda}{n},$$

so $\hat{\Delta}\boldsymbol{b}_j \ne \mathbf{0}$ and $j \in \hat{\mathcal{S}}$. Combining the two directions yields $\hat{\mathcal{S}} = \mathcal{S}^*$. □

**Lemma C.6.** *Under the conditions of Theorem 3.6, let* $\mathbf{O}$ *be a minimizer of* $\|\hat{\mathbf{F}}\mathbf{Q} - \mathbf{F}^*\|_F$ *over orthogonal* $\mathbf{Q} \in \mathbb{R}^{r \times r}$; *a minimizer exists by compactness of the orthogonal group and continuity of the objective, and can be chosen measurably because the associated argmin correspondence is closed valued and measurable. Then, with* $\tau_n = \lambda/n$: *(i)* $\delta_{F,n} = o_p(\tau_n)$; *(ii)* $\rho_{n,p} = o_p(\tau_n)$; *(iii)* $\zeta_{n,p} = o_p(\tau_n)$; *(iv)* $\tau_n = o(\beta_{\min})$ *and* $\tau_n = o(1)$.

**Proof of Lemma C.6:** Under the conditions of Theorem 3.6, the hypotheses of Theorem 3.3 are met at the same $\lambda$. Hence there exist a constant $C_0 < \infty$ and events $\mathcal{A}_{n,p}$ with $\mathbb{P}(\mathcal{A}_{n,p}) \to 1$ on which some orthogonal matrix $\mathbf{O}'$ satisfies $n^{-1/2}\|\hat{\mathbf{F}}\mathbf{O}' - \mathbf{F}^*\|_F \leq C_0\, a_{F,n}$, where

$$a_{F,n} = \frac{err + \sqrt{s\lambda}}{\sqrt{n}\sigma_{\mathbf{B}}}.$$

By the minimizing property of $\mathbf{O}$, on $\mathcal{A}_{n,p}$ we also have $\delta_{F,n} \leq n^{-1/2}\|\hat{\mathbf{F}}\mathbf{O}' - \mathbf{F}^*\|_F \leq C_0\, a_{F,n}$. We verify that $a_{F,n} = o(\tau_n)$, using that $\lambda$ and $err$ are fixed. For the first component of $a_{F,n}$,

$$\frac{err/(\sqrt{n}\sigma_{\mathbf{B}})}{\lambda/n} = \frac{\sqrt{n}\, err}{\sigma_{\mathbf{B}}\lambda} \to 0, \qquad \text{since } \sqrt{n}\, err/\sigma_{\mathbf{B}} \ll \lambda.$$

For the second component,

$$\frac{\sqrt{s\lambda}/(\sqrt{n}\sigma_{\mathbf{B}})}{\lambda/n} = \frac{\sqrt{n}\sqrt{s\lambda}}{\sigma_{\mathbf{B}}\lambda} = \left(\frac{ns}{\sigma_{\mathbf{B}}^2\lambda}\right)^{1/2} \to 0, \qquad \text{since } ns/\sigma_{\mathbf{B}}^2 \ll \lambda.$$

Hence $a_{F,n} = o(\tau_n)$, and for any $\epsilon > 0$,

$$\mathbb{P}(\delta_{F,n} > \epsilon\tau_n) \leq \mathbb{P}(\mathcal{A}_{n,p}^c) + \mathbf{1}\{C_0\, a_{F,n} > \epsilon\tau_n\} \to 0,$$

which proves (i). For (ii), $\rho_{n,p} = O_p(\rho_n)$ by assumption and $\rho_n/\tau_n = n\rho_n/\lambda \to 0$ since $n\rho_n \ll \lambda$. For (iii), combine Equation (14) with (i), (ii), and $\kappa_{n,p} = O_p(1)$:

$$\zeta_{n,p} \leq \rho_{n,p} + \delta_{F,n}\,\kappa_{n,p} = o_p(\tau_n) + o_p(\tau_n)\,O_p(1) = o_p(\tau_n).$$

For (iv), $\tau_n/\beta_{\min} = \lambda/(n\beta_{\min}) \to 0$ since $\lambda \ll n\beta_{\min}$; moreover, Condition (C1) implies $\|\Delta\boldsymbol{b}_j^*\|_2 \leq \|\boldsymbol{b}_j^{(2)*}\|_2 + \|\boldsymbol{b}_j^{(1)*}\|_2 \leq 2C$ for every $j$, so $\beta_{\min} \leq 2C$ and $\tau_n = o(1)$. $\qquad\square$

**Proof of Theorem 3.6:** Let $\mathbf{O}$ be the orthogonal matrix of Lemma C.6 and define the event

$$\mathcal{G}_{n,p} = \left\{\delta_{F,n} \leq \frac{1}{2}\right\} \cap \left\{\zeta_{n,p} \leq \frac{\tau_n}{2}\right\} \cap \left\{\tau_n \leq \frac{\beta_{\min}}{4}\right\}.$$

The third event is deterministic, because $\lambda$ is a sequence and $\beta_{\min}$ is a population quantity; by Lemma C.6(iv), it holds for all $(n,p)$ sufficiently large. By Lemma C.6(i) and (iv), $\delta_{F,n} = o_p(\tau_n) = o_p(1)$, and by Lemma C.6(iii), $\mathbb{P}(\zeta_{n,p} > \tau_n/2) \to 0$. A union bound over the three events gives

$$\mathbb{P}(\mathcal{G}_{n,p}^c) \leq \mathbb{P}\left(\delta_{F,n} > \frac{1}{2}\right) + \mathbb{P}\left(\zeta_{n,p} > \frac{\tau_n}{2}\right) + \mathbf{1}\left\{\tau_n > \frac{\beta_{\min}}{4}\right\} \to 0.$$

On $\mathcal{G}_{n,p}$, apply Proposition C.5 with $c_F = 1/2$ and $q = \tau_n/2$. Both requirements are met: $q = \tau_n/2 \leq \tau_n = \lambda/n$, and, using $\beta_{\min} \geq 4\tau_n$,

$$(1 - c_F)\beta_{\min} - q = \frac{\beta_{\min}}{2} - \frac{\tau_n}{2} \geq 2\tau_n - \frac{\tau_n}{2} = \frac{3\tau_n}{2} > \tau_n = \frac{\lambda}{n}.$$

Hence $\{j : \|\hat{\boldsymbol{z}}_j\|_2 > \tau_n\} = \mathcal{S}^*$ on $\mathcal{G}_{n,p}$.

It remains to connect this set to the support $\hat{\mathcal{S}}$ of the shift component of the minimizer of Equation (1). By Lemma C.1, the unconstrained minimizer of each feature subproblem satisfies $\hat{\boldsymbol{b}}_j = \{(1+c_j)\boldsymbol{u}_j^{(1)} + (1-c_j)\boldsymbol{u}_j^{(2)}\}/2$ and $\hat{\boldsymbol{b}}_j + \hat{\boldsymbol{d}}_j = \{(1-c_j)\boldsymbol{u}_j^{(1)} + (1+c_j)\boldsymbol{u}_j^{(2)}\}/2$, where $\boldsymbol{u}_j^{(k)} = n^{-1}\hat{\mathbf{F}}^\top \tilde{\boldsymbol{m}}_j^{(k)}$ and $c_j = (1 - \tau_n/\|\hat{\boldsymbol{z}}_j\|_2)_+ \in [0, 1)$, so both loading vectors lie in the convex hull of $\{\boldsymbol{u}_j^{(1)}, \boldsymbol{u}_j^{(2)}\}$. Writing $\rho_{n,p}^e$ and $\kappa_{n,p}^e$ for the environmentwise

maxima in Condition (C3) and arguing exactly as in Lemmas C.3 and C.4 with $\boldsymbol{e}_{\Delta,j}$ replaced by $\boldsymbol{e}_j^{(k)}$, we obtain $\max_{k,j} \|\mathbf{O}^\top \boldsymbol{u}_j^{(k)} - \boldsymbol{b}_j^{(k)*}\|_2 \leq \delta_{F,n} C + \rho_{n,p}^e + \delta_{F,n} \kappa_{n,p}^e = o_p(1)$, using $\delta_{F,n} = o_p(1)$, $\rho_{n,p}^e = O_p(\rho_n)$ with $\rho_n = o(1)$ from $n\rho_n \ll \lambda \ll n\beta_{\min}$ and $\beta_{\min} \leq 2C$, and $\kappa_{n,p}^e = O_p(1)$. Hence $\max_{k,j} \|\boldsymbol{u}_j^{(k)}\|_2 \leq C + o_p(1)$ by Condition (C1). Since the radius of $\mathcal{F}(C)$ is a sufficiently large constant, it exceeds this bound with probability tending to one, the norm constraints on the loadings are slack, and the loading component of the minimizer of Equation (1) coincides with the unconstrained solution, so that $\hat{\mathcal{S}} = \{j : \|\hat{\boldsymbol{z}}_j\|_2 > \tau_n\}$. Intersecting this event with $\mathcal{G}_{n,p}$ gives $\mathbb{P}(\hat{\mathcal{S}} = \mathcal{S}^*) \to 1$ as $n, p \to +\infty$. $\square$

**Proposition C.7.** *Let $L_{n,p} = \max\{n\rho_n, \sqrt{n}\, err/\sigma_{\mathbf{B}}, ns/\sigma_{\mathbf{B}}^2\}$ and $U_{n,p} = \min\{n\beta_{\min}, (n \vee p)/s\}$ denote the two sides of Equation (2), and assume $\rho_n > 0$. A sequence $\lambda$ satisfying Equation (2) exists if and only if $L_{n,p} = o(U_{n,p})$, in which case $\lambda = \sqrt{L_{n,p} U_{n,p}}$ is one such sequence. Moreover $L_{n,p} = o(U_{n,p})$ holds if and only if the six conditions*

$$\frac{\rho_n}{\beta_{\min}} \to 0, \quad \frac{ns\rho_n}{n \vee p} \to 0, \quad \frac{err}{\sqrt{n}\sigma_{\mathbf{B}}\beta_{\min}} \to 0, \quad \frac{s\sqrt{n}\, err}{\sigma_{\mathbf{B}}(n \vee p)} \to 0, \quad \frac{s}{\sigma_{\mathbf{B}}^2 \beta_{\min}} \to 0, \quad \frac{ns^2}{\sigma_{\mathbf{B}}^2(n \vee p)} \to 0$$

*are satisfied.*

**Proof of Proposition C.7:** Write

$$L_{n,p} = \max\left\{ n\rho_n, \frac{\sqrt{n}\, err}{\sigma_{\mathbf{B}}}, \frac{ns}{\sigma_{\mathbf{B}}^2} \right\}, \qquad U_{n,p} = \min\left\{ n\beta_{\min}, \frac{n \vee p}{s} \right\},$$

and note that $L_{n,p} > 0$ whenever $\rho_n > 0$. If a sequence $\lambda$ with $L_{n,p} \ll \lambda \ll U_{n,p}$ exists, then

$$\frac{L_{n,p}}{U_{n,p}} = \frac{L_{n,p}}{\lambda} \cdot \frac{\lambda}{U_{n,p}} \to 0,$$

so $L_{n,p} = o(U_{n,p})$. Conversely, if $L_{n,p} = o(U_{n,p})$, the choice $\lambda = \sqrt{L_{n,p} U_{n,p}}$ satisfies $\lambda/L_{n,p} = (U_{n,p}/L_{n,p})^{1/2} \to \infty$ and $\lambda/U_{n,p} = (L_{n,p}/U_{n,p})^{1/2} \to 0$, so the window is nonempty; this choice is an existence device rather than a tuning recommendation. Finally, since the maximum of finitely many nonnegative sequences is $o(\cdot)$ of the minimum of finitely many positive sequences if and only if every pairwise ratio vanishes, $L_{n,p} = o(U_{n,p})$ is equivalent to the six conditions

$$\frac{\rho_n}{\beta_{\min}} \to 0, \quad \frac{ns\rho_n}{n \vee p} \to 0, \quad \frac{err}{\sqrt{n}\sigma_{\mathbf{B}}\beta_{\min}} \to 0, \quad \frac{s\sqrt{n}\, err}{\sigma_{\mathbf{B}}(n \vee p)} \to 0, \quad \frac{s}{\sigma_{\mathbf{B}}^2 \beta_{\min}} \to 0, \quad \frac{ns^2}{\sigma_{\mathbf{B}}^2(n \vee p)} \to 0.$$

$\square$

**Proof of Corollary 3.7:** Let $d_{n,p} = \{\log(np)\}^2/\sqrt{n \wedge p}$ and write $\|\mathbf{A}\|_{\max} = \max_{i,j} |a_{ij}|$ for the entrywise maximum norm of a matrix $\mathbf{A} = (a_{ij})$.

*Step 1 (Stage I entrywise rate).* Write the thin singular value decomposition $\mathbf{M}^{(k)*} = \mathbf{U}^{(k)} \mathbf{\Sigma}^{(k)} (\mathbf{V}^{(k)})^\top$. Since $n^{-1}(\mathbf{F}^*)^\top \mathbf{F}^* = \mathbf{I}_r$, the matrix $n^{-1/2}\mathbf{F}^*$ has orthonormal columns spanning the column space of $\mathbf{M}^{(k)*}$, so $\mathbf{U}^{(k)} = n^{-1/2}\mathbf{F}^*\mathbf{Q}_k$ for some orthogonal $\mathbf{Q}_k$, and Condition (C1) yields the row-norm bound $\max_i \|(\mathbf{U}^{(k)})_{i,\cdot}\|_2 \leq Cn^{-1/2}$. Because $(\mathbf{M}^{(k)*})^\top \mathbf{M}^{(k)*} = n\mathbf{B}^{(k)*}(\mathbf{B}^{(k)*})^\top$, we have $\sigma_i(\mathbf{M}^{(k)*}) = \sqrt{n}\,\sigma_i(\mathbf{B}^{(k)*}) \asymp \sqrt{np}$ for $i \leq r^*$ under Condition (C2) with $\sigma_{\mathbf{B}} \asymp \sqrt{p}$, and $\mathbf{V}^{(k)} = (\mathbf{M}^{(k)*})^\top \mathbf{U}^{(k)} (\mathbf{\Sigma}^{(k)})^{-1} = \sqrt{n}\,\mathbf{B}^{(k)*}\mathbf{Q}_k(\mathbf{\Sigma}^{(k)})^{-1}$ satisfies $\max_j \|(\mathbf{V}^{(k)})_{j,\cdot}\|_2 \leq C/\sigma_{r^*}(\mathbf{B}^{(k)*}) \lesssim p^{-1/2}$; moreover $\max_{i,j} |m_{ij}^{(k)*}| \leq C^2$. Hence the structural requirements of Theorem 5 of Chen & Li (2024) hold with no missingness and fixed rank $r = r^*$. Under their generalized latent factor model regularity conditions (a common, twice differentiable link with known dispersion, together with their moment conditions) and the side condition $\{\log(np)\}^4 = o(n \wedge p)$, applying their entrywise refinement (their refinement algorithm) to the CJMLE initial estimator of Chen et al. (2020), whose scaled Frobenius error satisfies $(np)^{-1/2}\|\tilde{\mathbf{M}}_{\text{init}}^{(k)} - \mathbf{M}^{(k)*}\|_F \lesssim (n \wedge p)^{-1/2}$ with probability tending to one by Theorem 1 of Chen & Li (2022), delivers Stage I estimates $\tilde{\mathbf{M}}^{(k)}$ obeying

$$\max_{k \in \{1,2\}} \left\|\tilde{\mathbf{M}}^{(k)} - \mathbf{M}^{(k)*}\right\|_{\max} \lesssim d_{n,p} \tag{15}$$

with probability tending to one. The corollary presumes that Stage I returns this refined estimator.

*Step 2 (primitive conditions).* On the event in Equation (15), every column satisfies $\|\tilde{\boldsymbol{m}}_j^{(k)} - \boldsymbol{m}_j^{(k)*}\|_2 \leq \sqrt{n}\,\|\tilde{\mathbf{M}}^{(k)} - \mathbf{M}^{(k)*}\|_{\max}$, whence

$$\kappa_{n,p} = \max_j n^{-1/2}\,\|\boldsymbol{e}_{\Delta,j}\|_2 \leq 2\max_k \left\|\tilde{\mathbf{M}}^{(k)} - \mathbf{M}^{(k)*}\right\|_{\max} \lesssim d_{n,p} = o(1),$$

and, using $\|\mathbf{F}^*\|_2 = \sqrt{n}$,

$$\rho_{n,p} = \max_j \left\|n^{-1}(\mathbf{F}^*)^\top \boldsymbol{e}_{\Delta,j}\right\|_2 \leq n^{-1}\,\|\mathbf{F}^*\|_2 \max_j \|\boldsymbol{e}_{\Delta,j}\|_2 \leq 2\max_k \left\|\tilde{\mathbf{M}}^{(k)} - \mathbf{M}^{(k)*}\right\|_{\max} \lesssim d_{n,p}.$$

Thus the primitive conditions of Theorem 3.6 hold with $\rho_n \asymp d_{n,p}$ and, in fact, $\kappa_{n,p} = O_p(d_{n,p}) = o_p(1)$; the assumed lower bound $nd_{n,p} \ll \lambda$ is precisely $n\rho_n \ll \lambda$, so that $\rho_{n,p} = o_p(\tau_n)$.

*Step 3 (factor error).* By Equation (15), $\|\tilde{\mathbf{M}}^{(k)} - \mathbf{M}^{(k)*}\|_F \leq \sqrt{np}\,\|\tilde{\mathbf{M}}^{(k)} - \mathbf{M}^{(k)*}\|_{\max} \lesssim \sqrt{np}\,d_{n,p}$ with probability tending to one, so Theorem 3.3 applies with $err = \sqrt{np}\,d_{n,p}$, which is fixed. With $\mathbf{O}$ chosen as in Lemma C.6 and $\sigma_{\mathbf{B}} \asymp \sqrt{p}$,

$$\frac{err}{\sqrt{n}\sigma_{\mathbf{B}}} \asymp \frac{\sqrt{np}\,d_{n,p}}{\sqrt{np}} = d_{n,p} \to 0, \qquad \frac{\sqrt{s\lambda}}{\sqrt{n}\sigma_{\mathbf{B}}} \asymp \left(\frac{s\lambda}{np}\right)^{1/2} \ll \left(\frac{n \vee p}{np}\right)^{1/2} = \left(\frac{1}{n \wedge p}\right)^{1/2} \to 0,$$

where the second display uses $\lambda \ll (n \vee p)/s$. Hence $\delta_{F,n} = o_p(1)$.

*Step 4 (leakage and assembly).* By Equation (14),

$$\zeta_{n,p} \leq \rho_{n,p} + \delta_{F,n}\,\kappa_{n,p} = O_p(d_{n,p}) + o_p(1)\,O_p(d_{n,p}) = O_p(d_{n,p}) = o_p(\tau_n),$$

since $\tau_n = \lambda/n \gg d_{n,p}$ by assumption. This is where the entrywise refinement pays off: because $\kappa_{n,p} = O_p(d_{n,p}) = o_p(1)$, the leakage term $\delta_{F,n}\kappa_{n,p}$ requires only $\delta_{F,n} = o_p(1)$ rather than $\delta_{F,n} = o_p(\tau_n)$, and the two lower bounds $\sqrt{n}\,err/\sigma_{\mathbf{B}} \ll \lambda$ and $ns/\sigma_{\mathbf{B}}^2 \ll \lambda$ of Theorem 3.6 are no longer needed. Moreover, $\tau_n = o(\beta_{\min})$ from $\lambda \ll n\beta_{\min}$, and $\tau_n = o(1)$ since $\beta_{\min} \leq 2C$ by Condition (C1). Defining $\mathcal{G}_{n,p}$ as in the proof of Theorem 3.6, we again obtain $\mathbb{P}(\mathcal{G}_{n,p}) \to 1$, and Proposition C.5 with $c_F = 1/2$ and $q = \tau_n/2$ yields $\{j : \|\hat{\boldsymbol{z}}_j\|_2 > \tau_n\} = \mathcal{S}^*$ on $\mathcal{G}_{n,p}$. The constraint slackness step at the end of the proof of Theorem 3.6 applies verbatim, because the entrywise bound controls the environmentwise quantities in Condition (C3), and it transfers the conclusion to $\hat{\mathcal{S}}$. Finally, by the argument of Proposition C.7 with $L_{n,p} = nd_{n,p}$ and $U_{n,p} = \min\{n\beta_{\min}, (n\vee p)/s\}$, the simplified window is nonempty if and only if $\beta_{\min} \gg d_{n,p}$ and $ns\,d_{n,p} \ll n\vee p$. $\square$

**Proof of Theorem 3.8:** We begin by decomposing the difference in the information criterion values

$$\begin{aligned}
&\left\{\mathcal{L}\left(\mathbf{F}^*, \mathbf{B}^{(1)*}; \tilde{\mathbf{M}}^{(1)}\right) + \mathcal{L}\left(\mathbf{F}^*, \mathbf{B}^{(2)*}; \tilde{\mathbf{M}}^{(2)}\right) + r^*\eta(n,p)\right\} \\
&- \left\{\mathcal{L}\left(\hat{\mathbf{F}}_r, \hat{\mathbf{B}}_r^{(1)}; \tilde{\mathbf{M}}^{(1)}\right) + \mathcal{L}\left(\hat{\mathbf{F}}_r, \hat{\mathbf{B}}_r^{(2)}; \tilde{\mathbf{M}}^{(2)}\right) + r\eta(n,p)\right\} \\
&= \left\{\mathcal{L}\left(\mathbf{F}^*, \mathbf{B}^{(1)*}; \tilde{\mathbf{M}}^{(1)}\right) - \mathcal{L}\left(\hat{\mathbf{F}}_r, \hat{\mathbf{B}}_r^{(1)}; \tilde{\mathbf{M}}^{(1)}\right)\right\} + \left\{\mathcal{L}\left(\mathbf{F}^*, \mathbf{B}^{(2)*}; \tilde{\mathbf{M}}^{(2)}\right) - \mathcal{L}\left(\hat{\mathbf{F}}_r, \hat{\mathbf{B}}_r^{(2)}; \tilde{\mathbf{M}}^{(2)}\right)\right\} \\
&\quad + \left\{r^*\eta(n,p) - r\eta(n,p)\right\} \\
&= \mathbf{I} + \mathbf{II} + \mathbf{III}.
\end{aligned}$$

We analyze the proof in two scenarios.

**Case 1:** overestimation ($r \geq r^*$).

First, consider the term **I**. By applying Theorem 3.2, we bound the estimation error as follows

$$\begin{aligned}
\left\|\tilde{\mathbf{M}}^{(1)} - \hat{\mathbf{F}}_r\left(\hat{\mathbf{B}}_r^{(1)}\right)^\top\right\|_F &\leq \left\|\tilde{\mathbf{M}}^{(1)} - \mathbf{M}^{(1)*}\right\|_F + \left\|\mathbf{M}^{(1)*} - \hat{\mathbf{F}}_r\left(\hat{\mathbf{B}}_r^{(1)}\right)^\top\right\|_F \\
&\lesssim err + \sqrt{s\lambda}.
\end{aligned}$$

Consequently, the magnitude of **I** satisfies

$$\begin{aligned}
|\mathbf{I}| &\leq \left\|\tilde{\mathbf{M}}^{(1)} - \mathbf{F}^*\left(\mathbf{B}^{(1)*}\right)^\top\right\|_F^2 + \left\|\tilde{\mathbf{M}}^{(1)} - \hat{\mathbf{F}}_r\left(\hat{\mathbf{B}}_r^{(1)}\right)^\top\right\|_F^2 \\
&\lesssim (err)^2 + s\lambda
\end{aligned}$$

with probability tending to one.

Similarly, for the second term **II**, we have

$$|\mathbf{II}| \leq \left\| \tilde{\mathbf{M}}^{(1)} - \mathbf{F}^* \left( \mathbf{B}^{(1)*} \right)^\top \right\|_F^2 + \left\| \tilde{\mathbf{M}}^{(1)} - \hat{\mathbf{F}}_r \left( \hat{\mathbf{B}}_r^{(1)} \right)^\top \right\|_F^2$$
$$\lesssim (err)^2 + s\lambda.$$

For simplicity, we denote

$$\mathrm{IC}(*) = \mathcal{L}\left( \mathbf{F}^*, \mathbf{B}^{(1)*}; \tilde{\mathbf{M}}^{(1)} \right) + \mathcal{L}\left( \mathbf{F}^*, \mathbf{B}^{(2)*}; \tilde{\mathbf{M}}^{(2)} \right) + r^* \eta(n, p).$$

Thus, for any $r \geq r^*$, the difference in the criteria can be expressed as

$$\begin{aligned} \mathrm{IC}\left( r^* \right) - \mathrm{IC}(r) &= \left\{ \mathrm{IC}\left( r^* \right) - \mathrm{IC}(*) \right\} - \left\{ \mathrm{IC}(r) - \mathrm{IC}(*) \right\} \\ &= O_p\left( (err)^2 + s\lambda \right) + \left( r^* - r \right) \eta(n, p). \end{aligned} \tag{16}$$

Imposing the condition $\eta(n, p) \gg (err)^2 + s\lambda$, we observe that the penalty difference dominates the estimation error in Equation (16). Since $r > r^*$, the term $(r^* - r)\eta(n, p)$ is negative and dominant. Therefore,

$$\mathbb{P}\left\{ \mathrm{IC}\left( r^* \right) - \mathrm{IC}(r) < 0 \right\} \to 1$$

as $n, p \to +\infty$.

**Case 2:** Underestimation $(1 \leq r < r^*)$.

We begin by decomposing the first term, **I**. Using the definition of the Frobenius norm and expanding the squared terms, we obtain

$$\begin{aligned} \mathbf{I} &= \left\| \tilde{\mathbf{M}}^{(1)} - \mathbf{F}^* \left( \mathbf{B}^{(1)*} \right)^\top \right\|_F^2 - \left\| \tilde{\mathbf{M}}^{(1)} - \hat{\mathbf{F}}_r \left( \hat{\mathbf{B}}_r^{(1)} \right)^\top \right\|_F^2 \\ &= \left\| \tilde{\mathbf{M}}^{(1)} - \mathbf{M}^{(1)*} \right\|_F^2 - \left\| \tilde{\mathbf{M}}^{(1)} - \mathbf{M}^{(1)*} + \mathbf{M}^{(1)*} - \hat{\mathbf{M}}_r^{(1)} \right\|_F^2 \\ &\leq 2 \left\| \tilde{\mathbf{M}}^{(1)} - \mathbf{M}^{(1)*} \right\|_F \cdot \left\| \hat{\mathbf{M}}_r^{(1)} - \mathbf{M}^{(1)*} \right\|_F - \left\| \hat{\mathbf{M}}_r^{(1)} - \mathbf{M}^{(1)*} \right\|_F^2. \end{aligned}$$

Note that this expression represents a quadratic function with respect to $\left\| \hat{\mathbf{M}}_r^{(1)} - \mathbf{M}^{(1)*} \right\|_F$.

Recall that for any rank-$r$ approximation where $r < r^*$, the error is lower-bounded by the singular values of the true matrix. Specifically,

$$\left\| \hat{\mathbf{M}}_r^{(1)} - \mathbf{M}^{(1)*} \right\|_F \geq \sigma_{r^*} \left( \mathbf{M}^{(1)*} \right).$$

To bound **I**, let us define the auxiliary function $\mu(a) = 2a \left\| \tilde{\mathbf{M}}^{(1)} - \mathbf{M}_r^{(1)*} \right\|_F - a^2$. The inequality for **I** can then be expressed as

$$\mathbf{I} \leq \sup_{a \geq \sigma_{r^*} \left( \mathbf{M}^{(1)*} \right)} \mu(a).$$

If the singular value is sufficiently large such that

$$\sigma_{r^*} \left( \mathbf{M}^{(1)*} \right) \geq 4 \left\| \tilde{\mathbf{M}}^{(1)} - \mathbf{M}^{(1)*} \right\|_F,$$

then the supremum of $\mu(a)$ over the domain $a \geq \sigma_{r^*} \left( \mathbf{M}^{(1)*} \right)$ is attained at the boundary $a = \sigma_{r^*} \left( \mathbf{M}^{(1)*} \right)$.

Substituting this value back into $\mu(a)$, we derive the upper bound

$$\begin{aligned} \mathbf{I} \leq \mu\left( \sigma_{r^*} \left( \mathbf{M}^{(1)*} \right) \right) &= -\sigma_{r^*}^2 \left( \mathbf{M}^{(1)*} \right) + 2\sigma_{r^*} \left( \mathbf{M}^{(1)*} \right) \left\| \tilde{\mathbf{M}}^{(1)} - \mathbf{M}^{(1)*} \right\|_F \\ &\leq -\frac{1}{2} \sigma_{r^*}^2 \left( \mathbf{M}^{(1)*} \right). \end{aligned}$$

By an analogous argument, the second term **II** satisfies

$$\mathbf{II} \leq -\frac{1}{2}\sigma_{r^*}^2\left(\mathbf{M}^{(2)*}\right),$$

provided that the condition

$$\sigma_{r^*}\left(\mathbf{M}^{(2)*}\right) \geq 4\left\|\tilde{\mathbf{M}}^{(2)} - \mathbf{M}^{(2)*}\right\|_F$$

is met.

By the conditions, $\sigma_{r^*}\left(\mathbf{M}^{(1)*}\right) \asymp \sqrt{n}\sigma_{\mathbf{B}}$, $\left\|\tilde{\mathbf{M}}^{(1)} - \mathbf{M}^{(1)*}\right\|_F \lesssim err$, and $n\sigma_{\mathbf{B}}^2 \gg (err)^2 + s\lambda$, the required inequalities

$$\sigma_{r^*}\left(\mathbf{M}^{(1)*}\right) \geq 4\left\|\tilde{\mathbf{M}}^{(1)} - \mathbf{M}^{(1)*}\right\|_F$$

and

$$\sigma_{r^*}\left(\mathbf{M}^{(2)*}\right) \geq 4\left\|\tilde{\mathbf{M}}^{(2)} - \mathbf{M}^{(2)*}\right\|_F$$

hold with probability tending to one.

Combining these results, we obtain the following difference in the information criterion

$$\begin{aligned}
\mathrm{IC}\left(r^*\right) - \mathrm{IC}(r) &= \{\mathrm{IC}\left(r^*\right) - \mathrm{IC}(*)\} - \{\mathrm{IC}(r) - \mathrm{IC}(*)\} \\
&\leq -\frac{1}{2}\sigma_{r^*}^2\left(\mathbf{M}^{(1)*}\right) - \frac{1}{2}\sigma_{r^*}^2\left(\mathbf{M}^{(2)*}\right) + O_p\left((err)^2 + s\lambda\right) \\
&\quad + \left(r^* - r\right)\eta(n,p) \\
&\lesssim -n\sigma_{\mathbf{B}}^2 + O_p\left((err)^2 + s\lambda\right) + \left(r^* - r\right)\eta(n,p).
\end{aligned}$$

Given the condition $(err)^2 + s\lambda \ll \eta(n,p) \ll n\sigma_{\mathbf{B}}^2$, the term $-n\sigma_{\mathbf{B}}^2$ dominates the terms. Thus, we conclude that

$$\mathbb{P}\left\{\mathrm{IC}\left(r^*\right) - \mathrm{IC}(r) < 0\right\} \to 1$$

as $n, p \to +\infty$.

The above two cases lead to

$$\mathbb{P}\left(\hat{r} = r^*\right) \to 1$$

as $n, p \to +\infty$. $\qquad\square$

**Proof of Corollary 3.9:** We first establish the asymptotic dominance of the penalty term $\eta(n,p)$. Specifically, it holds that

$$\eta_1(n,p) = 2(n + p + \hat{s}_r)\log\left(\frac{np}{n+p}\right) \gg (n \vee p) \gtrsim err,$$

Furthermore, we verify that the penalty vanishes relative to the signal strength,

$$\frac{\eta_1(n,p)}{n\sigma_{\mathbf{B}}^2} = \frac{2(n+p+\hat{s})}{np}\log\left(\frac{np}{n+p}\right) \to 0.$$

The proof for the alternative penalty specification, given by

$$\eta_2(n,p) = 2(n + p + \hat{s})\log\left(n \wedge p\right)$$

follows an analogous argument and is therefore omitted. $\qquad\square$

**Proof of Theorem 4.1:** The proof of this theorem parallels that of Theorem 3.2. Therefore, we focus primarily on the necessary modifications for the multi-environment setting.

By the definition of the MLE, we have the following inequality

$$0 \geq \sum_{k=1}^{K} \left\{ \left\| \tilde{\mathbf{M}}^{(k)} - \hat{\mathbf{F}} \left( \hat{\mathbf{B}}^{(k)} \right)^{\top} \right\|_F^2 - \left\| \tilde{\mathbf{M}}^{(k)} - \mathbf{F}^* \left( \mathbf{B}^{(k*)} \right)^{\top} \right\|_F^2 \right\}$$

$$+ \lambda \sum_{k=2}^{K} \sum_{j=1}^{p} \left\{ \left\| \hat{\boldsymbol{b}}_j^{(k)} - \hat{\boldsymbol{b}}_j^{(k-1)} \right\|_2 - \left\| \boldsymbol{b}_j^{(k)*} - \boldsymbol{b}_j^{(k-1)*} \right\|_2 \right\}$$

$$:= \sum_{k=1}^{K} \mathbf{I}_k + \sum_{k=2}^{K} \mathbf{II}_k.$$

For the first term $\mathbf{I}_k$, applying an argument analogous to Equation (4) yields the lower bound

$$\mathbf{I}_k \geq \left\| \mathbf{M}^{(k)*} - \hat{\mathbf{F}} \left( \hat{\mathbf{B}}^{(k)} \right)^{\top} \right\|_F^2 - 2 \left\| \tilde{\mathbf{M}}^{(k)} - \mathbf{M}^{(k)*} \right\|_F \cdot \left\| \mathbf{M}^{(k)*} - \hat{\mathbf{F}} \left( \hat{\mathbf{B}}^{(k)} \right)^{\top} \right\|_F.$$

Similarly, summing the penalty terms leads to a bound analogous to Equation (5),

$$\sum_{k=2}^{K} \mathbf{II}_k \geq -2C(K-1)s\lambda.$$

Aggregating these bounds, we obtain

$$\sum_{k=1}^{K} \left\| \mathbf{M}^{(k)*} - \hat{\mathbf{F}} \left( \hat{\mathbf{B}}^{(k)} \right)^{\top} \right\|_F^2$$

$$\leq 2 \sum_{k=1}^{K} \left\| \tilde{\mathbf{M}}^{(k)} - \mathbf{M}^{(k)*} \right\|_F \cdot \left\| \mathbf{M}^{(k)*} - \hat{\mathbf{F}} \left( \hat{\mathbf{B}}^{(k)} \right)^{\top} \right\|_F + 2C(K-1)s\lambda$$

$$\leq 2 \left( \sum_{k=1}^{K} \left\| \tilde{\mathbf{M}}^{(k)} - \mathbf{M}^{(k)*} \right\|_F^2 \right)^{\frac{1}{2}} \cdot \left( \sum_{k=1}^{K} \left\| \mathbf{M}^{(k)*} - \hat{\mathbf{F}} \left( \hat{\mathbf{B}}^{(k)} \right)^{\top} \right\|_F^2 \right)^{\frac{1}{2}} + 2C(K-1)s\lambda.$$

Thus, either

$$\sum_{k=1}^{K} \left\| \mathbf{M}^{(k)*} - \hat{\mathbf{F}} \left( \hat{\mathbf{B}}^{(k)} \right)^{\top} \right\|_F^2 \leq 16 \sum_{k=1}^{K} \left\| \tilde{\mathbf{M}}^{(k)} - \mathbf{M}^{(k)*} \right\|_F^2$$

or

$$\sum_{k=1}^{K} \left\| \mathbf{M}^{(k)*} - \hat{\mathbf{F}} \left( \hat{\mathbf{B}}^{(k)} \right)^{\top} \right\|_F^2 \leq 4C(K-1)s\lambda$$

holds.

Applying the Cauchy-Schwarz inequality allows us to relate the sum of squares to the square of the sum

$$K \sum_{k=1}^{K} \left\| \mathbf{M}^{(k)*} - \hat{\mathbf{F}} \left( \hat{\mathbf{B}}^{(k)} \right)^{\top} \right\|_F^2 \geq \left\{ \sum_{k=1}^{K} \left\| \mathbf{M}^{(k)*} - \hat{\mathbf{F}} \left( \hat{\mathbf{B}}^{(k)} \right)^{\top} \right\|_F \right\}^2.$$

Dividing by $K$ and substituting the bounds derived above, we find

$$\frac{1}{K} \left\{ \sum_{k=1}^{K} \left\| \mathbf{M}^{(k)*} - \hat{\mathbf{F}} \left( \hat{\mathbf{B}}^{(k)} \right)^{\top} \right\|_F \right\}^2 \leq 16 \sum_{k=1}^{K} \left\| \mathbf{M}^{(k)*} - \hat{\mathbf{F}} \left( \hat{\mathbf{B}}^{(k)} \right)^{\top} \right\|_F^2 \lesssim K(err)^2$$

and consequently,

$$\frac{1}{K} \sum_{k=1}^{K} \left\| \mathbf{M}^{(k)*} - \hat{\mathbf{F}} \left( \hat{\mathbf{B}}^{(k)} \right)^{\top} \right\|_F \lesssim err,$$

or, in the case where the $s\lambda$ dominates,

$$\frac{1}{K}\sum_{k=1}^{K}\left\|\mathbf{M}^{(k)*}-\hat{\mathbf{F}}\left(\hat{\mathbf{B}}^{(k)}\right)^{\top}\right\|_{F}\lesssim\sqrt{\frac{K-1}{K}s\lambda}\lesssim\sqrt{s\lambda}.$$

Therefore, the average estimation error satisfies

$$\frac{1}{K}\sum_{k=1}^{K}\left\|\mathbf{M}^{(k)*}-\hat{\mathbf{F}}\left(\hat{\mathbf{B}}^{(k)}\right)^{\top}\right\|_{F}\lesssim err+\sqrt{s\lambda}.$$

with probability tending to one.

Next, we derive the error bounds for $\hat{\mathbf{F}}$ and $\hat{\mathbf{B}}^{(k)}$. Let us define the following stacked matrices

$$\hat{\bar{\mathbf{M}}}=\left(\hat{\mathbf{M}}^{(1)},\hat{\mathbf{M}}^{(2)},\ldots,\hat{\mathbf{M}}^{(K)}\right),$$

$$\bar{\mathbf{B}}^{*}=\left(\left(\mathbf{B}^{(1)*}\right)^{\top},\ldots,\left(\mathbf{B}^{(K)*}\right)^{\top}\right)^{\top},$$

and

$$\bar{\mathbf{M}}^{*}=\left(\mathbf{M}^{(1)*},\mathbf{M}^{(2)*},\ldots,\mathbf{M}^{(K)*}\right)=\mathbf{F}^{*}\left(\bar{\mathbf{B}}^{*}\right)^{\top}.$$

Under the Condition (C2), the singular values of the stacked signal matrix scale as

$$\sigma^{2}\left(\bar{\mathbf{M}}^{*}\right)\asymp Kn\sigma_{\mathbf{B}}^{2}.$$

We know that the total reconstruction error for the stacked matrix is bounded by

$$\left\|\bar{\mathbf{M}}-\bar{\mathbf{M}}^{*}\right\|_{F}^{2}=\sum_{k=1}^{K}\left\|\hat{\mathbf{M}}^{(k)}-\mathbf{M}^{(k)*}\right\|_{F}^{2}\lesssim K\left\{(err)^{2}+s\lambda\right\}$$

Applying the perturbation bound for the factor matrix $\hat{\mathbf{F}}$, we obtain

$$\frac{1}{n}\left\|\hat{\mathbf{F}}\mathbf{O}-\mathbf{F}^{*}\right\|_{F}^{2}\lesssim\frac{K\left\{(err)^{2}+s\lambda\right\}}{Kn\sigma_{\mathbf{B}}^{2}},$$

which simplifies to

$$\frac{1}{\sqrt{n}}\left\|\hat{\mathbf{F}}\mathbf{O}-\mathbf{F}^{*}\right\|_{F}\lesssim\frac{err+\sqrt{s\lambda}}{\sqrt{n}\sigma_{\mathbf{B}}}.$$

We now turn to the analysis of the $\hat{\mathbf{B}}^{(k)}$. This process follows the logic of Theorem 3.3, except that

$$\left\|\left(\mathbf{F}^{*}-\hat{\mathbf{F}}\mathbf{O}\right)\left(\bar{\mathbf{B}}^{*}\right)^{\top}\right\|_{F}\leq\left\|\mathbf{F}^{*}-\hat{\mathbf{F}}\mathbf{O}\right\|_{F}\left\|\bar{\mathbf{B}}^{*}\right\|_{F}\lesssim\sqrt{Kp}\left\|\mathbf{F}^{*}-\hat{\mathbf{F}}\mathbf{O}\right\|_{F}.$$

Substituting this into the error decomposition yields

$$\frac{1}{Kp}\left\|\hat{\mathbf{B}}\mathbf{O}-\bar{\mathbf{B}}^{*}\right\|_{F}^{2}\leq\frac{1}{n}\left\|\hat{\mathbf{F}}\mathbf{O}-\mathbf{F}^{*}\right\|_{F}^{2}+\frac{1}{Knp}\left\|\hat{\bar{\mathbf{M}}}-\bar{\mathbf{M}}^{*}\right\|_{F}^{2}$$

$$\lesssim\frac{(err)^{2}+s\lambda}{n\sigma_{\mathbf{B}}^{2}}+\frac{1}{Knp}\sum_{k=1}^{K}\left\|\mathbf{M}^{(k)*}-\hat{\mathbf{M}}^{(k)*}\right\|_{F}^{2}$$

$$\lesssim\frac{(err)^{2}+s\lambda}{n\sigma_{\mathbf{B}}^{2}}.$$

This implies a bound on the average squared error of $\hat{\mathbf{B}}^{(k)}$,

$$\frac{1}{p}\frac{1}{K}\sum_{k=1}^{K}\left\|\hat{\mathbf{B}}^{(k)}\mathbf{O}-\mathbf{B}^{(k)*}\right\|_{F}^{2}\lesssim\frac{(err)^{2}+s\lambda}{n\sigma_{\mathbf{B}}^{2}}.$$

Using the inequality relating the sum of squares to the square of the sum

$$K \sum_{k=1}^{K} \left\| \hat{\mathbf{B}}^{(k)} \mathbf{O} - \mathbf{B}^{(k)*} \right\|_F^2 \geq \left( \sum_{k=1}^{K} \left\| \hat{\mathbf{B}}^{(k)} \mathbf{O} - \mathbf{B}^{(k)*} \right\| \right)^2,$$

we deduce that the average estimation error for $\hat{\mathbf{B}}^{(k)}$ satisfies,

$$\frac{1}{K} \sum_{k=1}^{K} \left( \frac{1}{\sqrt{p}} \left\| \hat{\mathbf{B}}^{(k)} \mathbf{O} - \mathbf{B}^{(k)*} \right\|_F \right) \lesssim \frac{err + \sqrt{s}\lambda}{\sqrt{n}\sigma_{\mathbf{B}}}.$$

Finally, we analyze the structural changes. By the triangle inequality, the error of the differences is bounded by the sum of the individual estimation errors

$$\sum_{k=2}^{K} \left\| \Delta \hat{\mathbf{B}}^{(k)} - \Delta \mathbf{B}^{(k)*} \right\|_F$$

$$= \sum_{k=2}^{K} \left\| \left( \hat{\mathbf{B}}^{(k)} - \hat{\mathbf{B}}^{(k-1)} \right) \mathbf{O} - \left( \mathbf{B}^{(k)*} - \mathbf{B}^{(k-1)*} \right) \right\|_F$$

$$\leq \sum_{k=2}^{K} \left\{ \left\| \hat{\mathbf{B}}^{(k)} \mathbf{O} - \mathbf{B}^{(k)*} \right\|_F + \left\| \hat{\mathbf{B}}^{(k-1)} \mathbf{O} - \mathbf{B}^{(k-1)*} \right\|_F \right\}$$

$$< 2 \sum_{k=1}^{K} \left\| \hat{\mathbf{B}}^{(k)} \mathbf{O} - \mathbf{B}^{(k)*} \right\|_F.$$

Combining this with the average bound derived in Equation (5),

$$\frac{1}{K} \sum_{k=1}^{K} \left( \frac{1}{\sqrt{p}} \left\| \hat{\mathbf{B}}^{(k)} \mathbf{O} - \mathbf{B}^{(k)*} \right\|_F \right) \lesssim \frac{err + \sqrt{s}\lambda}{\sqrt{n}\sigma_{\mathbf{B}}},$$

we conclude that

$$\frac{1}{K-1} \sum_{k=2}^{K} \left\| \Delta \hat{\mathbf{B}}^{(k)} - \Delta \mathbf{B}^{(k)*} \right\|_F \lesssim \sqrt{\frac{p}{n\sigma_{\mathbf{B}}^2}} \left( err + \sqrt{s}\lambda \right).$$

Adjusting for the sparsity level $s$, we arrive at the final rate

$$\frac{1}{K-1} \sum_{k=2}^{K} \left( \frac{1}{\sqrt{s}} \left\| \Delta \hat{\mathbf{B}}^{(k)} \mathbf{O} - \Delta \mathbf{B}^{(k)*} \right\|_F \right) \lesssim \frac{\sqrt{p}}{\sigma_{\mathbf{B}}} \cdot \frac{err + \sqrt{s}\lambda}{\sqrt{ns}}.$$

Then the proof is finished. $\qquad\qquad\qquad\qquad\qquad\qquad\qquad\qquad\qquad\qquad\qquad\qquad\qquad\square$

**Proof of Theorem 4.2:** Recall the notation of Theorem 4.2: for $k \in \{2, \ldots, K\}$ and $j \in \{1, \ldots, p\}$, the edge noise $\boldsymbol{e}_{\Delta,j}^{(k)} = (\tilde{\boldsymbol{m}}_j^{(k)} - \boldsymbol{m}_j^{(k)*}) - (\tilde{\boldsymbol{m}}_j^{(k-1)} - \boldsymbol{m}_j^{(k-1)*})$, the primitive quantities

$$\rho_{K,n,p} = \max_{2 \leq k \leq K} \max_{1 \leq j \leq p} \left\| n^{-1} (\mathbf{F}^*)^\top \boldsymbol{e}_{\Delta,j}^{(k)} \right\|_2, \qquad \kappa_{K,n,p} = \max_{2 \leq k \leq K} \max_{1 \leq j \leq p} n^{-1/2} \left\| \boldsymbol{e}_{\Delta,j}^{(k)} \right\|_2,$$

with the envelopes $\rho_{K,n,p} = O_p(\rho_{K,n})$ and $\kappa_{K,n,p} = O_p(1)$ assumed in the theorem, the screening statistics $\hat{\boldsymbol{z}}_j^{(k)} = n^{-1} \hat{\mathbf{F}}^\top (\tilde{\boldsymbol{m}}_j^{(k)} - \tilde{\boldsymbol{m}}_j^{(k-1)})$, the refitted edge differences $\hat{\Delta}\boldsymbol{b}_j^{(k)} = \mathcal{T}_{\lambda/n}(\hat{\boldsymbol{z}}_j^{(k)})$, and the estimated edge-feature support $\hat{\mathcal{A}} = \{(k,j) : \|\hat{\Delta}\boldsymbol{b}_j^{(k)}\|_2 > 0\}$, where $\hat{\mathbf{F}}$ is the factor component of the joint multi-environment minimizer of Section 4, normalized so that $n^{-1}\hat{\mathbf{F}}^\top \hat{\mathbf{F}} = \mathbf{I}_r$. Throughout, $K$ is fixed, $\tau_n = \lambda/n$, and $\lambda$, $err$, and $\rho_{K,n}$ are fixed.

*Step 1.* For each adjacent pair $(k-1, k)$ and each $j$, Lemma C.1 applied with $\mathbf{F} = \hat{\mathbf{F}}$, $\boldsymbol{y}_1 = \tilde{\boldsymbol{m}}_j^{(k-1)}$, and $\boldsymbol{y}_2 = \tilde{\boldsymbol{m}}_j^{(k)}$ shows that $\hat{\Delta}\boldsymbol{b}_j^{(k)} = \mathcal{T}_{\lambda/n}(\hat{\boldsymbol{z}}_j^{(k)})$ is the difference component of the unique minimizer of the pairwise objective

$$\left\| \tilde{\boldsymbol{m}}_j^{(k-1)} - \hat{\mathbf{F}}\boldsymbol{b} \right\|_2^2 + \left\| \tilde{\boldsymbol{m}}_j^{(k)} - \hat{\mathbf{F}}(\boldsymbol{b} + \boldsymbol{d}) \right\|_2^2 + \lambda \|\boldsymbol{d}\|_2,$$

and that $(k, j) \in \hat{\mathcal{A}}$ if and only if $\|\hat{\boldsymbol{z}}_j^{(k)}\|_2 > \lambda/n$.

*Step 2.* Since $\mathbf{M}^{(k)*} = \mathbf{F}^*(\mathbf{B}^{(k)*})^\top$ for every $k$,

$$\tilde{\boldsymbol{m}}_j^{(k)} - \tilde{\boldsymbol{m}}_j^{(k-1)} = \mathbf{F}^* \Delta \boldsymbol{b}_j^{(k)*} + \boldsymbol{e}_{\Delta,j}^{(k)}.$$

With $\bar{\mathbf{F}} = \hat{\mathbf{F}}\mathbf{O}$ for the orthogonal matrix $\mathbf{O}$ chosen in Step 3, and $\mathbf{A}_n = n^{-1}\bar{\mathbf{F}}^\top \mathbf{F}^*$, exactly as in Equation (13),

$$\mathbf{O}^\top \hat{\boldsymbol{z}}_j^{(k)} = \mathbf{A}_n \Delta \boldsymbol{b}_j^{(k)*} + \boldsymbol{w}_j^{(k)}, \qquad \boldsymbol{w}_j^{(k)} = n^{-1}\bar{\mathbf{F}}^\top \boldsymbol{e}_{\Delta,j}^{(k)}.$$

The argument of Lemma C.3, applied verbatim to each pair $(k, j)$, gives the deterministic bound

$$\zeta_{K,n,p} := \max_{2 \le k \le K} \max_{1 \le j \le p} \left\| \boldsymbol{w}_j^{(k)} \right\|_2 \le \rho_{K,n,p} + \delta_{F,n}\, \kappa_{K,n,p}, \qquad \delta_{F,n} = n^{-1/2} \left\| \hat{\mathbf{F}}\mathbf{O} - \mathbf{F}^* \right\|_F.$$

We emphasize that this inequality holds simultaneously for all $(K-1)p$ candidate pairs at no additional stochastic cost: the bound is deterministic given the maxima $\rho_{K,n,p}$ and $\kappa_{K,n,p}$, whose stochastic control is assumed in the conditions of the theorem, and the Stage I event $\{\max_{1 \le k \le K} \|\tilde{\mathbf{M}}^{(k)} - \mathbf{M}^{(k)*}\|_F \lesssim err\}$ is a finite intersection of $K$ events each of probability tending to one, hence itself of probability tending to one for fixed $K$.

*Step 3.* By Theorem 4.1 with $r = r^*$ at the same $\lambda$, with probability tending to one there exists an orthogonal matrix $\mathbf{O}'$ such that

$$n^{-1/2} \left\| \hat{\mathbf{F}}\mathbf{O}' - \mathbf{F}^* \right\|_F \lesssim \frac{err + \sqrt{s}\lambda}{\sqrt{n}\sigma_{\mathbf{B}}} \le \frac{err + \sqrt{s_K}\lambda}{\sqrt{n}\sigma_{\mathbf{B}}},$$

where the second inequality uses that the sparsity parameter $s$ appearing in Theorem 4.1 does not exceed the edge count $s_K = |\mathcal{A}^*|$ and that the bound is increasing in the sparsity parameter. Choosing $\mathbf{O}$ as a measurable minimizer of $\|\hat{\mathbf{F}}\mathbf{Q} - \mathbf{F}^*\|_F$ over orthogonal $\mathbf{Q}$, as in Lemma C.6, $\delta_{F,n}$ satisfies the same bound with probability tending to one. The two ratio computations in the proof of Lemma C.6, with $s$ replaced by $s_K$, then yield $\delta_{F,n} = o_p(\tau_n)$ from the lower bounds $\sqrt{n}\, err/\sigma_{\mathbf{B}} \ll \lambda$ and $ns_K/\sigma_{\mathbf{B}}^2 \ll \lambda$. Moreover, $\rho_{K,n,p} = O_p(\rho_{K,n}) = o_p(\tau_n)$ from $n\rho_{K,n} \ll \lambda$; hence, by Step 2 and $\kappa_{K,n,p} = O_p(1)$, $\zeta_{K,n,p} = o_p(\tau_n)$. Finally, $\tau_n = o(\beta_{\min,K})$ from $\lambda \ll n\beta_{\min,K}$, and, since Condition (C1) holds with a uniform constant $C$ across environments, $\beta_{\min,K} = \min_{(k,j) \in \mathcal{A}^*} \|\Delta \boldsymbol{b}_j^{(k)*}\|_2 \le 2C$, so $\tau_n = o(1)$.

*Step 4.* Define

$$\mathcal{G}_{n,p} = \left\{ \delta_{F,n} \le \frac{1}{2} \right\} \cap \left\{ \zeta_{K,n,p} \le \frac{\tau_n}{2} \right\} \cap \left\{ \tau_n \le \frac{\beta_{\min,K}}{4} \right\}.$$

As in the proof of Theorem 3.6,

$$\mathbb{P}\left( \mathcal{G}_{n,p}^c \right) \le \mathbb{P}\left( \delta_{F,n} > \frac{1}{2} \right) + \mathbb{P}\left( \zeta_{K,n,p} > \frac{\tau_n}{2} \right) + \mathbf{1}\left\{ \tau_n > \frac{\beta_{\min,K}}{4} \right\} \to 0.$$

On $\mathcal{G}_{n,p}$, all $(K-1)p$ membership decisions are resolved correctly at once by the argument of Proposition C.5 with $c_F = 1/2$ and $q = \tau_n/2$, applied edge by edge. Indeed, by Lemma C.2, $\|\hat{\boldsymbol{z}}_j^{(k)}\|_2 = \|\mathbf{O}^\top \hat{\boldsymbol{z}}_j^{(k)}\|_2$. If $(k, j) \notin \mathcal{A}^*$, then $\Delta \boldsymbol{b}_j^{(k)*} = \mathbf{0}$ and

$$\left\| \hat{\boldsymbol{z}}_j^{(k)} \right\|_2 = \left\| \boldsymbol{w}_j^{(k)} \right\|_2 \le \zeta_{K,n,p} \le \frac{\tau_n}{2} \le \frac{\lambda}{n},$$

so $\hat{\Delta}\boldsymbol{b}_j^{(k)} = \mathbf{0}$ and $(k, j) \notin \hat{\mathcal{A}}$. If $(k, j) \in \mathcal{A}^*$, then, by Lemma C.4 and $\beta_{\min,K} \ge 4\tau_n$,

$$\left\| \hat{\boldsymbol{z}}_j^{(k)} \right\|_2 \ge (1 - \delta_{F,n}) \left\| \Delta \boldsymbol{b}_j^{(k)*} \right\|_2 - \zeta_{K,n,p} \ge \frac{\beta_{\min,K}}{2} - \frac{\tau_n}{2} \ge \frac{3\tau_n}{2} > \frac{\lambda}{n},$$

so $(k, j) \in \hat{\mathcal{A}}$. Hence $\hat{\mathcal{A}} = \mathcal{A}^*$ on $\mathcal{G}_{n,p}$, and $\mathbb{P}(\hat{\mathcal{A}} = \mathcal{A}^*) \ge \mathbb{P}(\mathcal{G}_{n,p}) \to 1$ as $n, p \to +\infty$. $\quad\square$

$\square$

**Proof of Theorem 4.3:** The proof is similar to Theorem 3.8 and thus omitted.

**Proof of Theorem B.1:** By the properties of the MLE, we obtain the following inequality

$$0 \geq \left\| \mathbf{X}^{(1)} - \hat{\mathbf{M}}^{(1)} \right\|_F^2 - \left\| \mathbf{X}^{(1)} - \mathbf{F}^* \mathbf{B}^{(1)*} \right\|_F^2$$
$$+ \left\| \mathbf{X}^{(2)} - \hat{\mathbf{M}}^{(2)} \right\|_F^2 - \left\| \mathbf{X}^{(2)} - \mathbf{F}^* \mathbf{B}^{(2)*} \right\|_F^2$$
$$+ \lambda \sum_{j=1}^{p} \left( \left\| \hat{\Delta \boldsymbol{b}}_j \right\|_2 - \left\| \Delta \boldsymbol{b}_j^* \right\|_2 \right)$$
$$:= \mathbf{I} + \mathbf{II} + \mathbf{III}.$$

Let $\langle \mathbf{A}, \mathbf{B} \rangle$ denote the matrix inner product $\mathrm{tr}(\mathbf{A}^\top \mathbf{B})$. The first term $\mathbf{I}$ can be expanded as

$$\mathbf{I} = \sum_{i=1}^{n} \sum_{j=1}^{p} \left\{ \left( x_{ij}^{(1)} - \hat{m}_{ij}^{(1)} \right)^2 - \left( x_{ij}^{(1)} - m_{ij}^{(1)*} \right)^2 \right\}$$
$$= \sum_{i=1}^{n} \sum_{j=1}^{p} \left\{ \left( \hat{m}_{ij}^{(1)} - m_{ij}^{(1)*} \right)^2 - 2 \left( x_{ij}^{(1)} - m_{ij}^{(1)*} \right) \left( \hat{m}_{ij}^{(1)} - m_{ij}^{(1)*} \right) \right\}$$
$$= \left\| \hat{\mathbf{M}}^{(1)} - \mathbf{M}^{(1)*} \right\|_F^2 - 2 \left\langle \mathbf{X}^{(1)} - \mathbf{M}^{(1)*}, \hat{\mathbf{M}}^{(1)} - \mathbf{M}^{(1)*} \right\rangle.$$

To bound the inner product, we utilize the matrix inequality $|\langle \mathbf{A}, \mathbf{B} \rangle| \leq \|\mathbf{A}\|_2 \|\mathbf{B}\|_* \leq \sqrt{\mathrm{rank}(\mathbf{B})} \|\mathbf{A}\|_2 \|\mathbf{B}\|_F$. Applying this to our context, where the rank of the difference matrix is at most $r + r^*$, we have

$$\left| \left\langle \mathbf{X}^{(1)} - \mathbf{M}^{(1)*}, \hat{\mathbf{M}}^{(1)} - \mathbf{M}^{(1)*} \right\rangle \right|$$
$$\leq \sqrt{r + r^*} \left\| \mathbf{X}^{(1)} - \mathbf{M}^{(1)*} \right\|_2 \left\| \hat{\mathbf{M}}^{(1)} - \mathbf{M}^{(1)*} \right\|_F.$$

Substituting this back into the expansion of $\mathbf{I}$ yields

$$\mathbf{I} \geq \left\| \hat{\mathbf{M}}^{(1)} - \mathbf{M}^{(1)*} \right\|_F^2 - 2\sqrt{r + r^*} \left\| \mathbf{X}^{(1)} - \mathbf{M}^{(1)*} \right\|_2 \cdot \left\| \hat{\mathbf{M}}^{(1)} - \mathbf{M}^{(1)*} \right\|_F.$$

Similarly, for the second term $\mathbf{II}$, we obtain

$$\mathbf{II} \geq \left\| \hat{\mathbf{M}}^{(2)} - \mathbf{M}^{(2)*} \right\|_F^2 - 2\sqrt{r + r^*} \left\| \mathbf{X}^{(2)} - \mathbf{M}^{(2)*} \right\|_2 \cdot \left\| \hat{\mathbf{M}}^{(2)} - \mathbf{M}^{(2)*} \right\|_F.$$

The analysis of the penalty term $\mathbf{III}$ follows the same logic as in Theorem 3.2, leading to the bound

$$\mathbf{III} \geq -2Cs\lambda.$$

Combining these results, the initial inequality becomes

$$0 \geq \mathbf{I} + \mathbf{II} + \mathbf{III} \geq \left\| \hat{\mathbf{M}}^{(1)} - \mathbf{M}^{(1)*} \right\|_F^2 + \left\| \hat{\mathbf{M}}^{(2)} - \mathbf{M}^{(2)*} \right\|_F^2$$
$$- 2\sqrt{r + r^*} \left\| \mathbf{X}^{(1)} - \mathbf{M}^{(1)*} \right\|_2 \cdot \left\| \hat{\mathbf{M}}^{(1)} - \mathbf{M}^{(1)*} \right\|_F$$
$$- 2\sqrt{r + r^*} \left\| \mathbf{X}^{(2)} - \mathbf{M}^{(2)*} \right\|_2 \cdot \left\| \hat{\mathbf{M}}^{(2)} - \mathbf{M}^{(2)*} \right\|_F$$
$$- 2Cs\lambda.$$

Rearranging the terms and applying the Cauchy-Schwarz inequality, we derive

$$\left\| \hat{\mathbf{M}}^{(1)} - \mathbf{M}^{(1)*} \right\|_F^2 + \left\| \hat{\mathbf{M}}^{(2)} - \mathbf{M}^{(2)*} \right\|_F^2$$
$$\leq 2\sqrt{r + r^*} \left( \left\| \mathbf{X}^{(1)} - \mathbf{M}^{(1)*} \right\|_2 \cdot \left\| \hat{\mathbf{M}}^{(1)} - \mathbf{M}^{(1)*} \right\|_F + \left\| \mathbf{X}^{(2)} - \mathbf{M}^{(2)*} \right\|_2 \cdot \left\| \hat{\mathbf{M}}^{(2)} - \mathbf{M}^{(2)*} \right\|_F \right)$$
$$+ 2Cs\lambda$$
$$\leq 2\sqrt{r + r^*} \left( \left\| \hat{\mathbf{M}}^{(1)} - \mathbf{M}^{(1)*} \right\|_F^2 + \left\| \hat{\mathbf{M}}^{(2)} - \mathbf{M}^{(2)*} \right\|_F^2 \right)^{\frac{1}{2}} \cdot \left( \left\| \mathbf{X}^{(1)} - \mathbf{M}^{(1)*} \right\|_2^2 + \left\| \mathbf{X}^{(2)} - \mathbf{M}^{(2)*} \right\|_2^2 \right)^{\frac{1}{2}}$$
$$+ 2Cs\lambda.$$

We next bound the spectral norms of the noise matrices, $\left\|\mathbf{X}^{(1)} - \mathbf{M}^{(1)*}\right\|_2$ and $\left\|\mathbf{X}^{(2)} - \mathbf{M}^{(2)*}\right\|_2$, by invoking Lemma 1 in Wang (2022). Assumption 2 in Wang (2022) is satisfied under our linear setting, and Assumption 3 is verified by our Condition (C4). Consequently, we have

$$\left\|\mathbf{X}^{(1)} - \mathbf{M}^{(1)*}\right\|_2 = O_p\left(n^{\frac{1}{2}}p^{\frac{1}{4}} \vee n^{\frac{1}{4}}p^{\frac{1}{2}}\right)$$

and

$$\left\|\mathbf{X}^{(2)} - \mathbf{M}^{(2)*}\right\|_2 = O_p\left(n^{\frac{1}{2}}p^{\frac{1}{4}} \vee n^{\frac{1}{4}}p^{\frac{1}{2}}\right).$$

Following the convergence analysis framework established in Theorem 3.2, these bounds imply that

$$\left\|\hat{\mathbf{M}}^{(1)} - \mathbf{M}^{(1)*}\right\|_F \lesssim n^{\frac{1}{2}}p^{\frac{1}{4}} \vee n^{\frac{1}{4}}p^{\frac{1}{2}} + \sqrt{s\lambda}$$

and

$$\left\|\hat{\mathbf{M}}^{(2)} - \mathbf{M}^{(2)*}\right\|_F \lesssim n^{\frac{1}{2}}p^{\frac{1}{4}} \vee n^{\frac{1}{4}}p^{\frac{1}{2}} + \sqrt{s\lambda}.$$

with probability tending to one.

Finally, analogously to Theorem 3.3, we deduce the rates for the factor and loading matrices,

$$\frac{1}{\sqrt{n}}\left\|\hat{\mathbf{F}}\mathbf{O} - \mathbf{F}^*\right\|_F \lesssim \frac{n^{\frac{1}{2}}p^{\frac{1}{4}} \vee n^{\frac{1}{4}}p^{\frac{1}{2}} + \sqrt{s\lambda}}{\sqrt{n}\sigma_{\mathbf{B}}},$$

$$\frac{1}{\sqrt{p}}\left\|\hat{\mathbf{B}}^{(k)}\mathbf{O} - \mathbf{B}^*\right\|_F \lesssim \frac{n^{\frac{1}{2}}p^{\frac{1}{4}} \vee n^{\frac{1}{4}}p^{\frac{1}{2}} + \sqrt{s\lambda}}{\sqrt{n}\sigma_{\mathbf{B}}},$$

and

$$\frac{1}{\sqrt{s}}\left\|\hat{\Delta\mathbf{B}}\mathbf{O} - \Delta\mathbf{B}^*\right\|_F \lesssim \sqrt{\frac{p}{\sigma_{\mathbf{B}}^2}} \cdot \frac{n^{\frac{1}{2}}p^{\frac{1}{4}} \vee n^{\frac{1}{4}}p^{\frac{1}{2}} + \sqrt{s\lambda}}{\sqrt{ns}}.$$

This concludes the proof. $\square$

# D   Additional Results on Synthetic Data

We also examine the reliability of the proposed rank selection criteria. The true rank is fixed at $r^* = 2$, with $s = 10$, $c = 2$, and $r_{\max} = 5$, and we use $\eta_1(n,p)$ and $\eta_2(n,p)$ as penalty terms. Table 1 reports the frequency with which the criteria ($\mathrm{IC}_1$ and $\mathrm{IC}_2$, corresponding to $\eta_1$ and $\eta_2$) correctly identify the true rank. For both Gaussian and Poisson data, they achieve an almost 100% success rate regardless of the sample size $n$ or the dimension $p$. Even under the more challenging Binary distribution, the criteria maintain high precision.

Table 1: Frequency of correctly identifying the true rank with $\mathrm{IC}_1$ and $\mathrm{IC}_2$, with varying $n$ and $p$ for Gaussian, Poisson, and binary data over 200 repetitions.

| $n$ | $p$ | Binary | | Gaussian | | Poisson | |
|---|---|---|---|---|---|---|---|
| | | $\mathrm{IC}_1$ | $\mathrm{IC}_2$ | $\mathrm{IC}_1$ | $\mathrm{IC}_2$ | $\mathrm{IC}_1$ | $\mathrm{IC}_2$ |
| 50 | 50 | 99.5% | 100.0% | 100.0% | 100.0% | 100.0% | 100.0% |
| | 100 | 100.0% | 100.0% | 100.0% | 100.0% | 100.0% | 100.0% |
| | 300 | 100.0% | 100.0% | 100.0% | 100.0% | 100.0% | 100.0% |
| 100 | 50 | 100.0% | 100.0% | 100.0% | 100.0% | 100.0% | 100.0% |
| | 100 | 100.0% | 100.0% | 100.0% | 100.0% | 100.0% | 100.0% |
| | 300 | 100.0% | 100.0% | 100.0% | 100.0% | 100.0% | 100.0% |
| 300 | 50 | 100.0% | 100.0% | 100.0% | 100.0% | 100.0% | 100.0% |
| | 100 | 100.0% | 100.0% | 100.0% | 100.0% | 100.0% | 100.0% |
| | 300 | 100.0% | 100.0% | 100.0% | 100.0% | 100.0% | 100.0% |

## D.1   Additional Results for Poisson Data

To provide a comprehensive dissection of the SMART method's internal behavior regarding variable selection mechanisms, parameter estimation, and model fit, we examine four sets of auxiliary metrics in this supplementary section, complementing the primary indicators presented in the main text.

Specifically, we report Precision and Recall for the difference loading $\Delta\mathbf{B}$ to investigate the model's specific tendencies in sparse recovery. While the $F_1$ score used in the main text offers a harmonic mean of these

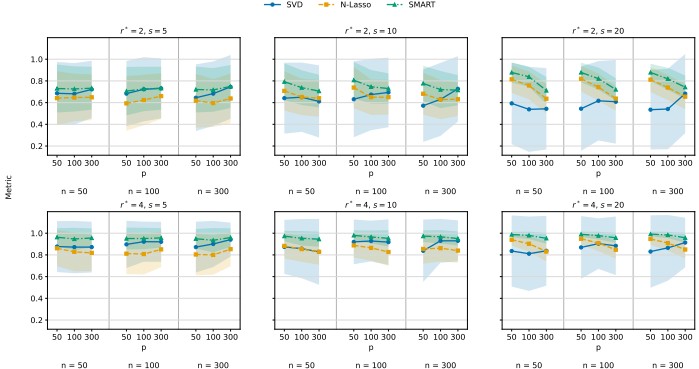

Figure 5: Variable selection recall for Poisson data, with varying $n$, $p$, $r^*$, and $s$ over 200 repetitions.

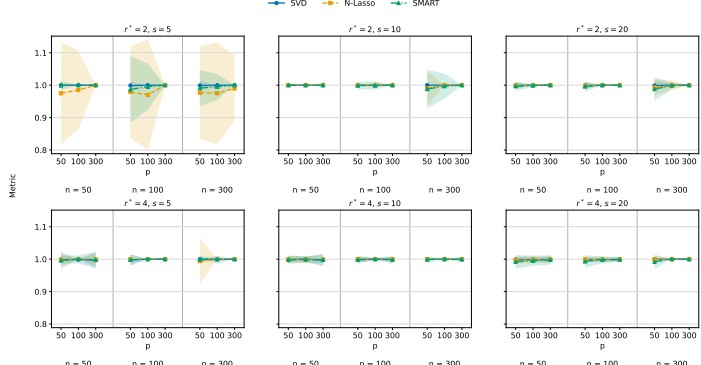

Figure 6: Variable selection precision for Poisson data, with varying $n$, $p$, $r^*$, and $s$ over 200 repetitions.

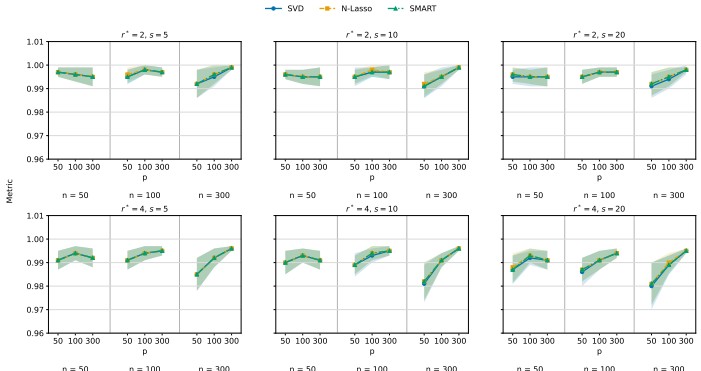

Figure 7: Subspace Recovery Accuracy ($\rho_{\min}$) of $\mathbf{F}$ for Poisson data, with varying $n$, $p$, $r^*$, and $s$ over 200 repetitions.

Table 2: Summary of the additional simulation studies. Detailed results are reported in the following subsections.

| Experiment | Purpose | Main observation |
|---|---|---|
| Multi-environment simulation | Vary $K$ and assess the extension in Section 4. | Union $F_1$ remains high for $K = 2, \ldots, 5$; fitting time increases smoothly with $K$. |
| Mixed data simulation | Combine Gaussian and Poisson coordinates in the same observation matrix. | SMART remains competitive or better than N-Lasso, especially for row support recovery. |
| One-step versus two-stage | Vary Gaussian noise and compare accuracy with computational cost. | One-step is faster and useful at low noise; two-stage is broader because it handles non-Gaussian and mixed data. |

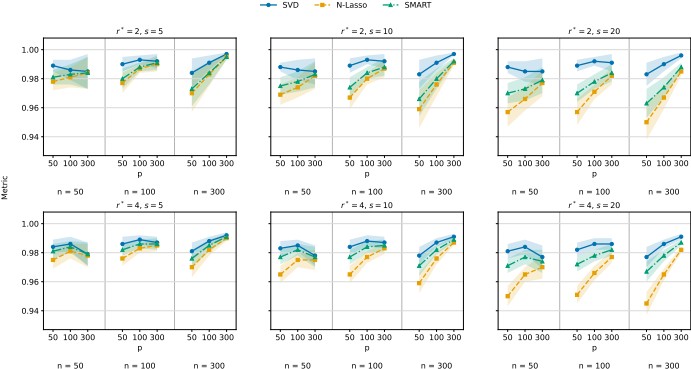

Figure 8: Estimation accuracy of latent parameters ($R^2$) for control group on Poisson data, with varying $n$, $p$, $r^*$, and $s$ over 200 repetitions.

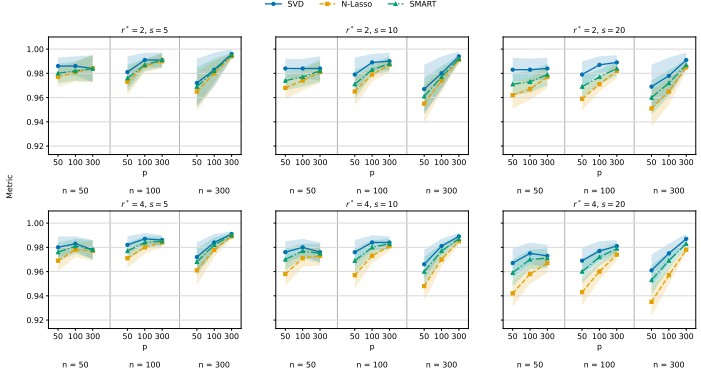

Figure 9: Estimation accuracy of latent parameters ($R^2$) for treatment group on Poisson data, with varying $n$, $p$, $r^*$, and $s$ over 200 repetitions.

aspects, independent analysis allows for finer granular insight. Based on the estimated support set $\hat{\mathcal{S}} = \{j : \|\hat{\Delta}\boldsymbol{b}_j\|_2 > 10^{-5}\}$ and the true support set $\mathcal{S}^*$, Precision is defined as $|\mathcal{S}^* \cap \hat{\mathcal{S}}|/(|\hat{\mathcal{S}}| + \epsilon)$, measuring the proportion of identified difference variables that correspond to true non-zero signals. High precision indicates a low False Positive Rate. Conversely, Recall is defined as $|\mathcal{S}^* \cap \hat{\mathcal{S}}|/(|\mathcal{S}^*| + \epsilon)$, where $\epsilon$ is a small constant to ensure numerical stability, quantifying the proportion of actual difference variables successfully captured by the model. The specific results for these metrics are detailed in Figure 5 (Recall) and Figure 6 (Precision).

Additionally, the subspace recovery results for the shared representation matrix $\mathbf{F}$, which were omitted from the main text, are provided in Figure 7.

Finally, we evaluate the model's parameter estimation capability via $R^2$. This metric directly assesses the model's ability to reconstruct the true parameter matrix $\mathbf{M}^*$. Since the true parameters $\mathbf{M}^{(k)*} = \mathbf{F}(\mathbf{B}^{(k)*})^\top$ are known in the simulation, we calculate the denoised coefficient of determination:

$$R^2(\mathbf{M}^*, \hat{\mathbf{M}}) = 1 - \frac{\sum_{i,j}(m_{ij}^* - \hat{m}_{ij})^2}{\sum_{i,j}(m_{ij}^* - \bar{m}^*)^2},$$

where $\bar{m}^* = (np)^{-1} \sum_{i=1}^{n} \sum_{j=1}^{p} m_{ij}^*$. For Poisson and Gaussian distributions, $\mathbf{M}$ corresponds to the linear predictor, while for the Binary distribution, it corresponds to the Logit probability. An $R^2$ value approaching 1 indicates that the model has effectively stripped away observational noise to reveal the underlying data generation mechanism. The performance is summarized in Figure 8 (Environment 1) and Figure 9 (Environment 2).

## D.2   Results for Gaussian Data

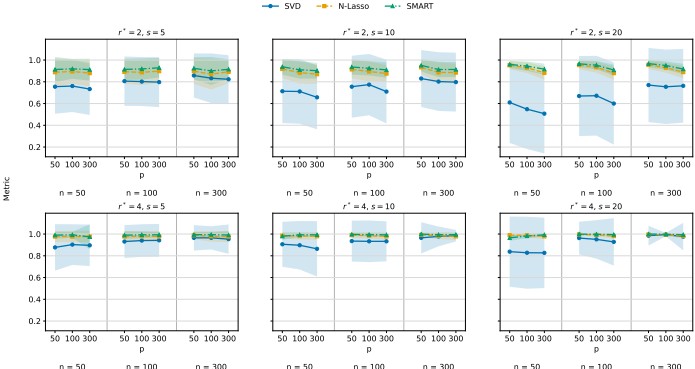

Figure 10: Variable selection performance ($F_1$ score) for Gaussian data, with varying $n$, $p$, $r^*$, and $s$ over 200 repetitions.

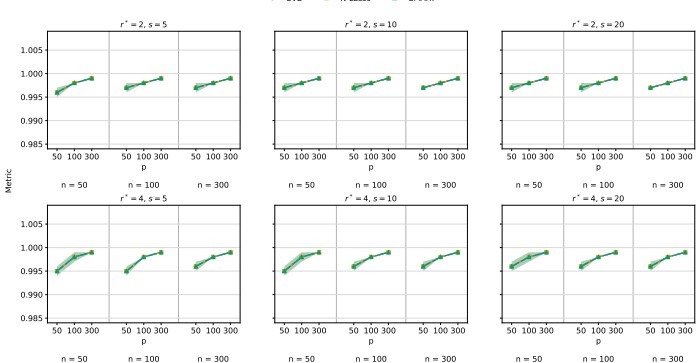

Figure 11: Subspace recovery accuracy ($\rho_{\min}$) of $\mathbf{F}$ for Gaussian data, with varying $n$, $p$, $r^*$, and $s$ over 200 repetitions.

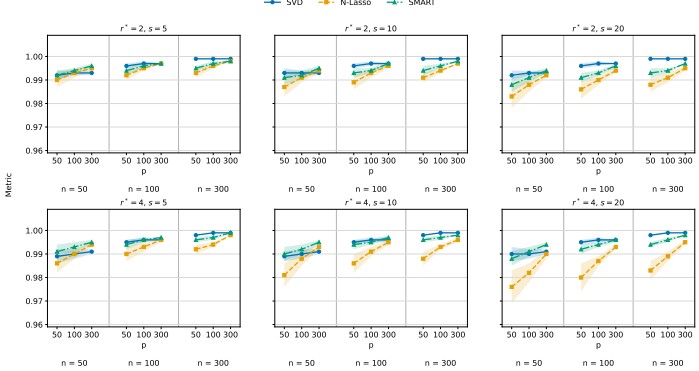

Figure 12: Subspace recovery accuracy ($\rho_{\min}$) of $\mathbf{B}^{(1)}$ for Gaussian data, with varying $n$, $p$, $r^*$, and $s$ over 200 repetitions.

For the Gaussian distribution experiments, the evaluation metrics are consistent with those in the main text, with the addition of the Minimum Canonical Correlation for $\mathbf{F}$. The variable selection performance $F_1$ Score is reported in Figure 10. Subspace recovery accuracies for $\mathbf{F}$, $\mathbf{B}^{(1)}$ and $\mathbf{B}^{(2)}$ are detailed in Figures 11, 12, and 13.

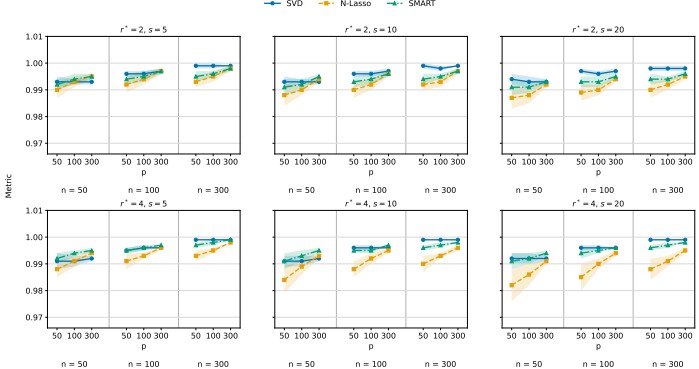

Figure 13: Subspace recovery accuracy ($\rho_{\min}$) of $\mathbf{B}^{(2)}$ for Gaussian data, with varying $n$, $p$, $r^*$, and $s$ over 200 repetitions.

### D.3    Results for Binary Data

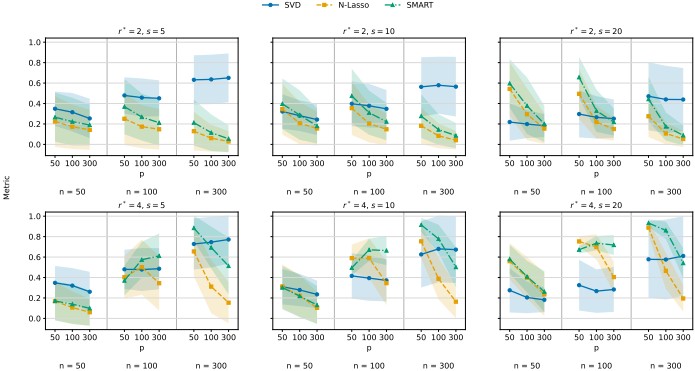

Figure 14: Variable selection performance ($F_1$ score) for Binary data, with varying $n$, $p$, $r^*$, and $s$ over 200 repetitions.

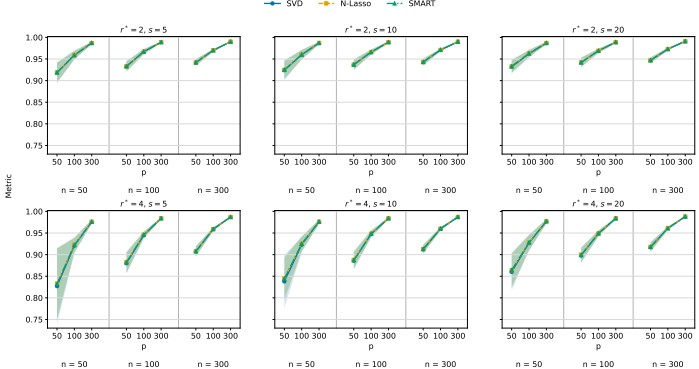

Figure 15: Subspace recovery accuracy ($\rho_{\min}$) of $\mathbf{F}$ for Binary data, with varying $n$, $p$, $r^*$, and $s$ over 200 repetitions.

For the Binary distribution experiments, the evaluation metrics are identical to those used in the Gaussian setting. The variable selection metrics ($F_1$ Score) are presented in Figure 14. Subspace recovery performance is shown in Figures 15, 16 and 17.

### D.4    Results for Poisson Data under Missingness

In this section, we evaluate the applicability and robustness of the SMART framework under incomplete count data scenarios. We assume a Missing Completely At Random (MCAR) mechanism. For the latent matrices $\mathbf{X}_{\text{full}}^{(k)}$ (where $k \in \{1, 2\}$), we introduce a binary mask matrix $\mathbf{\Omega}^{(k)} \in \{0, 1\}^{n \times p}$, where elements are

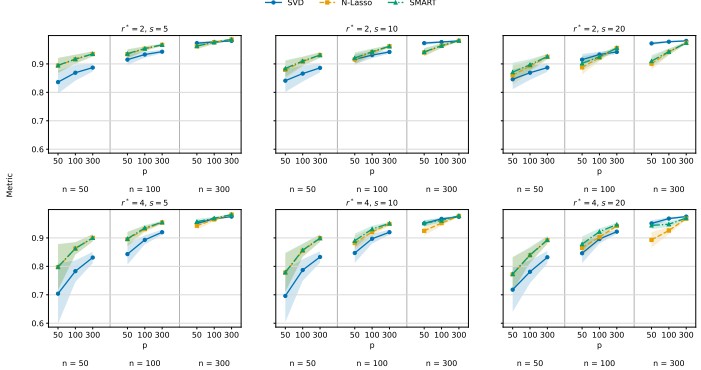

Figure 16: Subspace recovery accuracy ($\rho_{\min}$) of $\mathbf{B}^{(1)}$ for Binary data, with varying $n$, $p$, $r^*$, and $s$ over 200 repetitions.

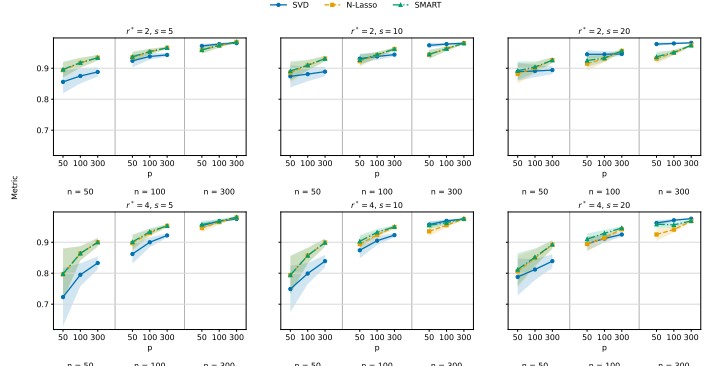

Figure 17: Subspace recovery accuracy ($\rho_{\min}$) of $\mathbf{B}^{(2)}$ for Binary data, with varying $n$, $p$, $r^*$, and $s$ over 200 repetitions.

Table 3: Variable selection performance ($F_1$ Score) for Poisson data under missingness, with varying $n$, $p$, $r^*$, and $\rho$ over 200 repetitions.

| | | | $\rho = 0.1$ | | | | $\rho = 0.2$ | | | |
| | | | N-Lasso | | SMART | | N-Lasso | | SMART | |
| $r^*$ | $n$ | $p$ | mean | std | mean | std | mean | std | mean | std |
|---|---|---|---|---|---|---|---|---|---|---|
| 2 | 50 | 50 | 0.925 | 0.086 | 0.951 | 0.063 | 0.885 | 0.113 | 0.922 | 0.086 |
| | | 100 | 0.899 | 0.082 | 0.929 | 0.068 | 0.835 | 0.116 | 0.888 | 0.087 |
| | | 300 | 0.842 | 0.115 | 0.884 | 0.095 | 0.761 | 0.147 | 0.827 | 0.123 |
| | 100 | 50 | 0.936 | 0.064 | 0.957 | 0.052 | 0.897 | 0.091 | 0.934 | 0.072 |
| | | 100 | 0.886 | 0.100 | 0.919 | 0.082 | 0.852 | 0.118 | 0.894 | 0.105 |
| | | 300 | 0.845 | 0.101 | 0.891 | 0.083 | 0.777 | 0.123 | 0.842 | 0.103 |
| | 300 | 50 | 0.928 | 0.074 | 0.957 | 0.049 | 0.891 | 0.097 | 0.931 | 0.069 |
| | | 100 | 0.909 | 0.089 | 0.938 | 0.070 | 0.847 | 0.120 | 0.905 | 0.097 |
| | | 300 | 0.850 | 0.098 | 0.896 | 0.078 | 0.782 | 0.110 | 0.851 | 0.087 |
| 4 | 50 | 50 | 0.959 | 0.052 | 0.995 | 0.016 | 0.931 | 0.062 | 0.987 | 0.027 |
| | | 100 | 0.933 | 0.083 | 0.982 | 0.038 | 0.882 | 0.086 | 0.966 | 0.045 |
| | | 300 | 0.896 | 0.078 | 0.972 | 0.040 | 0.820 | 0.109 | 0.945 | 0.054 |
| | 100 | 50 | 0.968 | 0.051 | 0.995 | 0.015 | 0.925 | 0.072 | 0.987 | 0.033 |
| | | 100 | 0.931 | 0.072 | 0.986 | 0.028 | 0.876 | 0.093 | 0.974 | 0.036 |
| | | 300 | 0.895 | 0.081 | 0.972 | 0.038 | 0.822 | 0.111 | 0.950 | 0.053 |
| | 300 | 50 | 0.964 | 0.049 | 0.996 | 0.014 | 0.908 | 0.089 | 0.989 | 0.023 |
| | | 100 | 0.932 | 0.073 | 0.986 | 0.026 | 0.876 | 0.095 | 0.978 | 0.035 |
| | | 300 | 0.884 | 0.088 | 0.974 | 0.042 | 0.815 | 0.120 | 0.947 | 0.054 |

Table 4: Subspace recovery ($\rho_{\min}$) of **F** for Poisson data under missingness, with varying $n$, $p$, $r^*$, and $\rho$ over 200 repetitions.

| | | | $\rho = 0.1$ | | | | | | $\rho = 0.2$ | | | | | |
| | | | SVD | | N-Lasso | | SMART | | SVD | | N-Lasso | | SMART | |
| $r^*$ | $n$ | $p$ | mean | std | mean | std | mean | std | mean | std | mean | std | mean | std |
|---|---|---|---|---|---|---|---|---|---|---|---|---|---|---|
| 2 | 50 | 50 | 0.987 | 0.004 | 0.987 | 0.004 | 0.987 | 0.004 | 0.987 | 0.004 | 0.987 | 0.004 | 0.987 | 0.004 |
| | | 100 | 0.992 | 0.002 | 0.992 | 0.002 | 0.992 | 0.002 | 0.991 | 0.002 | 0.991 | 0.002 | 0.991 | 0.002 |
| | | 300 | 0.995 | 0.001 | 0.995 | 0.001 | 0.995 | 0.001 | 0.994 | 0.001 | 0.994 | 0.001 | 0.994 | 0.001 |
| | 100 | 50 | 0.988 | 0.003 | 0.988 | 0.003 | 0.988 | 0.003 | 0.987 | 0.003 | 0.987 | 0.003 | 0.987 | 0.003 |
| | | 100 | 0.992 | 0.002 | 0.992 | 0.002 | 0.992 | 0.002 | 0.991 | 0.002 | 0.991 | 0.002 | 0.991 | 0.002 |
| | | 300 | 0.995 | 0.001 | 0.995 | 0.001 | 0.995 | 0.001 | 0.995 | 0.001 | 0.995 | 0.001 | 0.995 | 0.001 |
| | 300 | 50 | 0.988 | 0.003 | 0.988 | 0.003 | 0.988 | 0.003 | 0.987 | 0.003 | 0.987 | 0.003 | 0.987 | 0.003 |
| | | 100 | 0.992 | 0.002 | 0.992 | 0.002 | 0.992 | 0.002 | 0.992 | 0.002 | 0.992 | 0.002 | 0.992 | 0.002 |
| | | 300 | 0.995 | 0.001 | 0.995 | 0.001 | 0.995 | 0.001 | 0.995 | 0.001 | 0.995 | 0.001 | 0.995 | 0.001 |
| 4 | 50 | 50 | 0.973 | 0.010 | 0.973 | 0.010 | 0.973 | 0.010 | 0.970 | 0.012 | 0.970 | 0.011 | 0.970 | 0.012 |
| | | 100 | 0.987 | 0.004 | 0.987 | 0.003 | 0.987 | 0.003 | 0.985 | 0.004 | 0.985 | 0.004 | 0.985 | 0.004 |
| | | 300 | 0.994 | 0.001 | 0.994 | 0.001 | 0.994 | 0.001 | 0.993 | 0.001 | 0.993 | 0.001 | 0.993 | 0.001 |
| | 100 | 50 | 0.973 | 0.008 | 0.974 | 0.008 | 0.974 | 0.008 | 0.972 | 0.008 | 0.973 | 0.007 | 0.973 | 0.008 |
| | | 100 | 0.988 | 0.003 | 0.988 | 0.003 | 0.988 | 0.003 | 0.987 | 0.003 | 0.987 | 0.003 | 0.987 | 0.003 |
| | | 300 | 0.994 | 0.001 | 0.994 | 0.001 | 0.994 | 0.001 | 0.994 | 0.001 | 0.994 | 0.001 | 0.994 | 0.001 |
| | 300 | 50 | 0.975 | 0.007 | 0.976 | 0.007 | 0.976 | 0.007 | 0.974 | 0.007 | 0.974 | 0.007 | 0.974 | 0.007 |
| | | 100 | 0.988 | 0.003 | 0.988 | 0.003 | 0.988 | 0.003 | 0.987 | 0.003 | 0.987 | 0.003 | 0.987 | 0.003 |
| | | 300 | 0.994 | 0.001 | 0.994 | 0.001 | 0.994 | 0.001 | 0.994 | 0.001 | 0.994 | 0.001 | 0.994 | 0.001 |

Table 5: Subspace recovery ($\rho_{\min}$) of $\mathbf{B}^{(1)}$ for Poisson data under missingness, with varying $n$, $p$, $r^*$, and $\rho$ over 200 repetitions.

| | | | $\rho = 0.1$ | | | | | | $\rho = 0.2$ | | | | | |
| | | | SVD | | N-Lasso | | SMART | | SVD | | N-Lasso | | SMART | |
| $r^*$ | $n$ | $p$ | mean | std | mean | std | mean | std | mean | std | mean | std | mean | std |
|---|---|---|---|---|---|---|---|---|---|---|---|---|---|---|
| 2 | 50 | 50 | 0.984 | 0.004 | 0.967 | 0.010 | 0.976 | 0.007 | 0.982 | 0.005 | 0.960 | 0.012 | 0.971 | 0.008 |
| | | 100 | 0.985 | 0.003 | 0.976 | 0.005 | 0.981 | 0.004 | 0.983 | 0.003 | 0.971 | 0.006 | 0.977 | 0.005 |
| | | 300 | 0.985 | 0.003 | 0.983 | 0.003 | 0.985 | 0.003 | 0.984 | 0.003 | 0.981 | 0.003 | 0.983 | 0.003 |
| | 100 | 50 | 0.990 | 0.002 | 0.972 | 0.008 | 0.981 | 0.005 | 0.989 | 0.003 | 0.965 | 0.010 | 0.976 | 0.006 |
| | | 100 | 0.991 | 0.002 | 0.980 | 0.004 | 0.985 | 0.003 | 0.990 | 0.002 | 0.976 | 0.005 | 0.982 | 0.004 |
| | | 300 | 0.991 | 0.001 | 0.987 | 0.002 | 0.989 | 0.002 | 0.991 | 0.001 | 0.985 | 0.002 | 0.987 | 0.002 |
| | 300 | 50 | 0.994 | 0.001 | 0.976 | 0.006 | 0.984 | 0.004 | 0.994 | 0.001 | 0.968 | 0.009 | 0.979 | 0.005 |
| | | 100 | 0.994 | 0.001 | 0.983 | 0.003 | 0.987 | 0.002 | 0.994 | 0.001 | 0.978 | 0.005 | 0.984 | 0.003 |
| | | 300 | 0.995 | 0.001 | 0.989 | 0.001 | 0.991 | 0.001 | 0.994 | 0.001 | 0.988 | 0.002 | 0.990 | 0.001 |
| 4 | 50 | 50 | 0.973 | 0.007 | 0.940 | 0.018 | 0.965 | 0.010 | 0.968 | 0.009 | 0.922 | 0.022 | 0.955 | 0.013 |
| | | 100 | 0.974 | 0.006 | 0.957 | 0.010 | 0.971 | 0.007 | 0.972 | 0.006 | 0.951 | 0.011 | 0.967 | 0.008 |
| | | 300 | 0.976 | 0.004 | 0.974 | 0.004 | 0.978 | 0.004 | 0.973 | 0.005 | 0.970 | 0.005 | 0.974 | 0.005 |
| | 100 | 50 | 0.985 | 0.003 | 0.950 | 0.014 | 0.975 | 0.006 | 0.984 | 0.003 | 0.938 | 0.017 | 0.969 | 0.008 |
| | | 100 | 0.986 | 0.003 | 0.966 | 0.007 | 0.981 | 0.003 | 0.985 | 0.003 | 0.960 | 0.008 | 0.977 | 0.005 |
| | | 300 | 0.987 | 0.002 | 0.982 | 0.003 | 0.986 | 0.003 | 0.986 | 0.002 | 0.979 | 0.003 | 0.984 | 0.003 |
| | 300 | 50 | 0.993 | 0.001 | 0.956 | 0.011 | 0.982 | 0.004 | 0.993 | 0.001 | 0.943 | 0.014 | 0.976 | 0.006 |
| | | 100 | 0.993 | 0.001 | 0.972 | 0.006 | 0.986 | 0.002 | 0.993 | 0.001 | 0.965 | 0.008 | 0.982 | 0.003 |
| | | 300 | 0.994 | 0.001 | 0.986 | 0.002 | 0.991 | 0.001 | 0.994 | 0.001 | 0.984 | 0.002 | 0.990 | 0.001 |

Table 6: Subspace recovery ($\rho_{\min}$) of $\mathbf{B}^{(2)}$ for Poisson data under missingness, with varying $n$, $p$, $r^*$, and $\rho$ over 200 repetitions.

| | | | $\rho = 0.1$ | | | | | | $\rho = 0.2$ | | | | | |
| | | | SVD | | N-Lasso | | SMART | | SVD | | N-Lasso | | SMART | |
| $r^*$ | $n$ | $p$ | mean | std | mean | std | mean | std | mean | std | mean | std | mean | std |
|---|---|---|---|---|---|---|---|---|---|---|---|---|---|---|
| 2 | 50 | 50 | 0.971 | 0.011 | 0.947 | 0.016 | 0.955 | 0.014 | 0.969 | 0.011 | 0.939 | 0.017 | 0.949 | 0.015 |
| | | 100 | 0.974 | 0.008 | 0.954 | 0.011 | 0.960 | 0.011 | 0.971 | 0.010 | 0.948 | 0.014 | 0.954 | 0.013 |
| | | 300 | 0.979 | 0.005 | 0.970 | 0.007 | 0.972 | 0.007 | 0.978 | 0.005 | 0.967 | 0.007 | 0.969 | 0.007 |
| | 100 | 50 | 0.976 | 0.008 | 0.953 | 0.013 | 0.961 | 0.011 | 0.975 | 0.009 | 0.946 | 0.014 | 0.956 | 0.012 |
| | | 100 | 0.978 | 0.007 | 0.958 | 0.010 | 0.963 | 0.009 | 0.976 | 0.008 | 0.950 | 0.012 | 0.957 | 0.011 |
| | | 300 | 0.985 | 0.004 | 0.973 | 0.006 | 0.976 | 0.006 | 0.984 | 0.004 | 0.970 | 0.007 | 0.973 | 0.007 |
| | 300 | 50 | 0.980 | 0.008 | 0.956 | 0.012 | 0.964 | 0.010 | 0.979 | 0.008 | 0.948 | 0.012 | 0.958 | 0.010 |
| | | 100 | 0.982 | 0.007 | 0.961 | 0.010 | 0.966 | 0.009 | 0.982 | 0.006 | 0.955 | 0.010 | 0.962 | 0.009 |
| | | 300 | 0.988 | 0.004 | 0.975 | 0.006 | 0.978 | 0.006 | 0.987 | 0.004 | 0.973 | 0.006 | 0.975 | 0.006 |
| 4 | 50 | 50 | 0.943 | 0.019 | 0.903 | 0.023 | 0.926 | 0.020 | 0.938 | 0.021 | 0.889 | 0.024 | 0.917 | 0.021 |
| | | 100 | 0.952 | 0.013 | 0.918 | 0.016 | 0.936 | 0.015 | 0.949 | 0.013 | 0.912 | 0.016 | 0.929 | 0.016 |
| | | 300 | 0.965 | 0.007 | 0.950 | 0.008 | 0.957 | 0.008 | 0.962 | 0.008 | 0.947 | 0.009 | 0.954 | 0.009 |
| | 100 | 50 | 0.958 | 0.013 | 0.917 | 0.020 | 0.940 | 0.015 | 0.957 | 0.014 | 0.905 | 0.020 | 0.933 | 0.017 |
| | | 100 | 0.963 | 0.010 | 0.928 | 0.013 | 0.945 | 0.012 | 0.962 | 0.010 | 0.923 | 0.015 | 0.941 | 0.014 |
| | | 300 | 0.974 | 0.007 | 0.956 | 0.009 | 0.963 | 0.008 | 0.973 | 0.007 | 0.953 | 0.009 | 0.961 | 0.009 |
| | 300 | 50 | 0.967 | 0.009 | 0.923 | 0.015 | 0.947 | 0.011 | 0.968 | 0.009 | 0.911 | 0.018 | 0.942 | 0.013 |
| | | 100 | 0.969 | 0.009 | 0.934 | 0.013 | 0.951 | 0.012 | 0.969 | 0.008 | 0.924 | 0.013 | 0.945 | 0.011 |
| | | 300 | 0.980 | 0.006 | 0.960 | 0.008 | 0.968 | 0.008 | 0.980 | 0.006 | 0.958 | 0.008 | 0.966 | 0.008 |

sampled independently from a Bernoulli distribution with a missing rate $\rho$. The observed data are defined as the sparse matrix $\mathbf{X}_{\text{obs}}^{(k)} = \mathbf{X}_{\text{full}}^{(k)} \circ \mathbf{\Omega}^{(k)}$.

We investigate the capability of the two-stage SMART method to handle missingness with sample sizes and dimensions $(n, p) \in \{50, 100, 300\}^2$, latent ranks $r^* \in \{2, 4\}$, sparsity $s = 10$, and missing rates $\rho \in \{0.1, 0.2\}$. Unobserved entries are excluded from the optimization during training.

The methodology remains identical to the full-data case, with the sole exception of the CV procedure. To accommodate missingness, we employ an entry-wise CV where only the observed indices are partitioned into folds. In each fold, the designated validation entries are masked and treated as additional missing values during the Stage I CJMLE signal recovery. This allows the model to select $\lambda$ by minimizing the prediction error specifically on observed but held-out entries. Once Stage I produces the completed matrix $\tilde{\mathbf{M}}_{\text{complete}}$, the Stage II attribution proceeds exactly as described in the complete-data framework. The results, summarizing variable selection performance and subspace recovery precision, are presented in Tables 3–6.

### D.5 Results for One-Step Estimation

Table 7: Variable selection performance ($F_1$ Score) for one-step estimation under missingness, with varying $n$, $p$, $r^*$, and $\rho$ over 200 repetitions.

| | | | $\rho = 0.1$ | | | | $\rho = 0.2$ | | | |
| | | | N-Lasso | | SMART | | N-Lasso | | SMART | |
| $r^*$ | $n$ | $p$ | mean | std | mean | std | mean | std | mean | std |
|---|---|---|---|---|---|---|---|---|---|---|
| 2 | 50 | 50 | 0.902 | 0.090 | 0.930 | 0.076 | 0.844 | 0.121 | 0.892 | 0.099 |
| | | 100 | 0.862 | 0.105 | 0.897 | 0.091 | 0.801 | 0.121 | 0.858 | 0.097 |
| | | 300 | 0.850 | 0.089 | 0.882 | 0.087 | 0.816 | 0.103 | 0.861 | 0.089 |
| | 100 | 50 | 0.898 | 0.089 | 0.919 | 0.077 | 0.866 | 0.096 | 0.904 | 0.075 |
| | | 100 | 0.878 | 0.089 | 0.910 | 0.068 | 0.817 | 0.118 | 0.866 | 0.098 |
| | | 300 | 0.853 | 0.098 | 0.889 | 0.087 | 0.818 | 0.106 | 0.856 | 0.094 |
| | 300 | 50 | 0.898 | 0.082 | 0.929 | 0.072 | 0.849 | 0.118 | 0.890 | 0.096 |
| | | 100 | 0.866 | 0.089 | 0.897 | 0.078 | 0.816 | 0.102 | 0.862 | 0.091 |
| | | 300 | 0.858 | 0.088 | 0.893 | 0.072 | 0.820 | 0.092 | 0.867 | 0.077 |
| 4 | 50 | 50 | 0.981 | 0.028 | 0.990 | 0.022 | 0.954 | 0.057 | 0.985 | 0.027 |
| | | 100 | 0.969 | 0.043 | 0.992 | 0.020 | 0.946 | 0.059 | 0.986 | 0.026 |
| | | 300 | 0.962 | 0.047 | 0.989 | 0.025 | 0.936 | 0.056 | 0.979 | 0.035 |
| | 100 | 50 | 0.984 | 0.030 | 0.996 | 0.014 | 0.969 | 0.040 | 0.993 | 0.021 |
| | | 100 | 0.977 | 0.036 | 0.994 | 0.017 | 0.956 | 0.053 | 0.987 | 0.028 |
| | | 300 | 0.968 | 0.042 | 0.991 | 0.021 | 0.939 | 0.057 | 0.983 | 0.032 |
| | 300 | 50 | 0.982 | 0.034 | 0.996 | 0.014 | 0.968 | 0.048 | 0.992 | 0.025 |
| | | 100 | 0.975 | 0.041 | 0.990 | 0.024 | 0.956 | 0.059 | 0.990 | 0.024 |
| | | 300 | 0.966 | 0.042 | 0.990 | 0.021 | 0.952 | 0.049 | 0.986 | 0.028 |

Table 8: Subspace recovery ($\rho_{\min}$) of $\mathbf{F}$ for one-step estimation under missingness, with varying $n$, $p$, $r^*$, and $\rho$ over 200 repetitions.

| | | | $\rho = 0.1$ | | | | | | $\rho = 0.2$ | | | | | |
| | | | SVD | | N-Lasso | | SMART | | SVD | | N-Lasso | | SMART | |
| $r^*$ | $n$ | $p$ | mean | std | mean | std | mean | std | mean | std | mean | std | mean | std |
|---|---|---|---|---|---|---|---|---|---|---|---|---|---|---|
| 2 | 50 | 50 | 0.996 | 0.001 | 0.996 | 0.001 | 0.996 | 0.001 | 0.996 | 0.001 | 0.996 | 0.001 | 0.996 | 0.001 |
| | | 100 | 0.998 | 0.000 | 0.998 | 0.000 | 0.998 | 0.000 | 0.998 | 0.001 | 0.998 | 0.001 | 0.998 | 0.001 |
| | | 300 | 0.999 | 0.000 | 0.999 | 0.000 | 0.999 | 0.000 | 0.999 | 0.000 | 0.999 | 0.000 | 0.999 | 0.000 |
| | 100 | 50 | 0.996 | 0.001 | 0.996 | 0.001 | 0.996 | 0.001 | 0.996 | 0.001 | 0.996 | 0.001 | 0.996 | 0.001 |
| | | 100 | 0.998 | 0.000 | 0.998 | 0.000 | 0.998 | 0.000 | 0.998 | 0.000 | 0.998 | 0.000 | 0.998 | 0.000 |
| | | 300 | 0.999 | 0.000 | 0.999 | 0.000 | 0.999 | 0.000 | 0.999 | 0.000 | 0.999 | 0.000 | 0.999 | 0.000 |
| | 300 | 50 | 0.997 | 0.001 | 0.996 | 0.001 | 0.996 | 0.001 | 0.996 | 0.001 | 0.996 | 0.001 | 0.996 | 0.001 |
| | | 100 | 0.998 | 0.000 | 0.998 | 0.000 | 0.998 | 0.000 | 0.998 | 0.000 | 0.998 | 0.000 | 0.998 | 0.000 |
| | | 300 | 0.999 | 0.000 | 0.999 | 0.000 | 0.999 | 0.000 | 0.999 | 0.000 | 0.999 | 0.000 | 0.999 | 0.000 |
| 4 | 50 | 50 | 0.994 | 0.001 | 0.994 | 0.001 | 0.994 | 0.001 | 0.993 | 0.002 | 0.993 | 0.002 | 0.993 | 0.002 |
| | | 100 | 0.997 | 0.001 | 0.997 | 0.001 | 0.997 | 0.001 | 0.997 | 0.001 | 0.997 | 0.001 | 0.997 | 0.001 |
| | | 300 | 0.999 | 0.000 | 0.999 | 0.000 | 0.999 | 0.000 | 0.999 | 0.000 | 0.999 | 0.000 | 0.999 | 0.000 |
| | 100 | 50 | 0.995 | 0.001 | 0.995 | 0.001 | 0.995 | 0.001 | 0.994 | 0.001 | 0.994 | 0.001 | 0.994 | 0.001 |
| | | 100 | 0.998 | 0.000 | 0.998 | 0.000 | 0.998 | 0.000 | 0.997 | 0.000 | 0.997 | 0.000 | 0.997 | 0.000 |
| | | 300 | 0.999 | 0.000 | 0.999 | 0.000 | 0.999 | 0.000 | 0.999 | 0.000 | 0.999 | 0.000 | 0.999 | 0.000 |
| | 300 | 50 | 0.995 | 0.001 | 0.995 | 0.001 | 0.995 | 0.001 | 0.995 | 0.001 | 0.995 | 0.001 | 0.995 | 0.001 |
| | | 100 | 0.998 | 0.000 | 0.998 | 0.000 | 0.998 | 0.000 | 0.998 | 0.000 | 0.998 | 0.000 | 0.998 | 0.000 |
| | | 300 | 0.999 | 0.000 | 0.999 | 0.000 | 0.999 | 0.000 | 0.999 | 0.000 | 0.999 | 0.000 | 0.999 | 0.000 |

Beyond the two-stage framework, we explore an efficient One-step Estimation strategy (Theorem B.1) specifically designed for continuous (Gaussian) data under the MCAR assumption. This extension performs end-to-end structured decomposition and missing value imputation simultaneously by minimizing a regularized

Table 9: Subspace recovery ($\rho_{\min}$) of $\mathbf{B}^{(1)}$ for one-step estimation under missingness, with varying $n$, $p$, $r^*$, and $\rho$ over 200 repetitions.

| | | | $\rho = 0.1$ | | | | | | $\rho = 0.2$ | | | | | |
| | | | SVD | | N-Lasso | | SMART | | SVD | | N-Lasso | | SMART | |
| $r^*$ | $n$ | $p$ | mean | std | mean | std | mean | std | mean | std | mean | std | mean | std |
|---|---|---|---|---|---|---|---|---|---|---|---|---|---|---|
| 2 | 50 | 50 | 0.992 | 0.002 | 0.985 | 0.005 | 0.989 | 0.003 | 0.991 | 0.002 | 0.981 | 0.006 | 0.986 | 0.004 |
| | | 100 | 0.992 | 0.002 | 0.989 | 0.003 | 0.991 | 0.002 | 0.991 | 0.002 | 0.987 | 0.003 | 0.989 | 0.002 |
| | | 300 | 0.993 | 0.001 | 0.993 | 0.001 | 0.994 | 0.001 | 0.992 | 0.001 | 0.993 | 0.001 | 0.993 | 0.001 |
| | 100 | 50 | 0.996 | 0.001 | 0.987 | 0.003 | 0.991 | 0.002 | 0.996 | 0.001 | 0.984 | 0.005 | 0.988 | 0.004 |
| | | 100 | 0.996 | 0.001 | 0.991 | 0.002 | 0.993 | 0.001 | 0.996 | 0.001 | 0.990 | 0.002 | 0.992 | 0.002 |
| | | 300 | 0.997 | 0.000 | 0.996 | 0.001 | 0.996 | 0.001 | 0.996 | 0.000 | 0.995 | 0.001 | 0.996 | 0.001 |
| | 300 | 50 | 0.999 | 0.000 | 0.989 | 0.003 | 0.993 | 0.002 | 0.999 | 0.000 | 0.985 | 0.004 | 0.990 | 0.003 |
| | | 100 | 0.999 | 0.000 | 0.993 | 0.001 | 0.995 | 0.001 | 0.999 | 0.000 | 0.991 | 0.002 | 0.993 | 0.001 |
| | | 300 | 0.999 | 0.000 | 0.997 | 0.001 | 0.997 | 0.000 | 0.999 | 0.000 | 0.996 | 0.001 | 0.997 | 0.001 |
| 4 | 50 | 50 | 0.989 | 0.003 | 0.978 | 0.007 | 0.988 | 0.004 | 0.987 | 0.003 | 0.971 | 0.010 | 0.984 | 0.005 |
| | | 100 | 0.990 | 0.002 | 0.985 | 0.003 | 0.991 | 0.002 | 0.988 | 0.002 | 0.983 | 0.004 | 0.989 | 0.003 |
| | | 300 | 0.991 | 0.002 | 0.991 | 0.002 | 0.993 | 0.001 | 0.989 | 0.002 | 0.990 | 0.002 | 0.992 | 0.001 |
| | 100 | 50 | 0.995 | 0.001 | 0.982 | 0.005 | 0.992 | 0.002 | 0.994 | 0.001 | 0.977 | 0.007 | 0.990 | 0.003 |
| | | 100 | 0.995 | 0.001 | 0.989 | 0.002 | 0.994 | 0.001 | 0.995 | 0.001 | 0.986 | 0.003 | 0.993 | 0.001 |
| | | 300 | 0.996 | 0.000 | 0.994 | 0.001 | 0.996 | 0.001 | 0.995 | 0.001 | 0.994 | 0.001 | 0.996 | 0.001 |
| | 300 | 50 | 0.998 | 0.000 | 0.984 | 0.004 | 0.994 | 0.001 | 0.998 | 0.000 | 0.980 | 0.005 | 0.992 | 0.002 |
| | | 100 | 0.999 | 0.000 | 0.991 | 0.002 | 0.996 | 0.001 | 0.998 | 0.000 | 0.989 | 0.002 | 0.995 | 0.001 |
| | | 300 | 0.999 | 0.000 | 0.996 | 0.001 | 0.998 | 0.000 | 0.999 | 0.000 | 0.995 | 0.001 | 0.997 | 0.000 |

Table 10: Subspace recovery ($\rho_{\min}$) of $\mathbf{B}^{(2)}$ for one-step estimation under missingness, with varying $n$, $p$, $r^*$, and $\rho$ over 200 repetitions.

| | | | $\rho = 0.1$ | | | | | | $\rho = 0.2$ | | | | | |
| | | | SVD | | N-Lasso | | SMART | | SVD | | N-Lasso | | SMART | |
| $r^*$ | $n$ | $p$ | mean | std | mean | std | mean | std | mean | std | mean | std | mean | std |
|---|---|---|---|---|---|---|---|---|---|---|---|---|---|---|
| 2 | 50 | 50 | 0.993 | 0.002 | 0.988 | 0.004 | 0.991 | 0.003 | 0.992 | 0.002 | 0.984 | 0.005 | 0.988 | 0.004 |
| | | 100 | 0.993 | 0.002 | 0.990 | 0.003 | 0.992 | 0.002 | 0.992 | 0.002 | 0.988 | 0.003 | 0.990 | 0.003 |
| | | 300 | 0.993 | 0.001 | 0.994 | 0.001 | 0.994 | 0.001 | 0.992 | 0.001 | 0.993 | 0.001 | 0.994 | 0.001 |
| | 100 | 50 | 0.997 | 0.001 | 0.990 | 0.003 | 0.993 | 0.002 | 0.996 | 0.001 | 0.986 | 0.004 | 0.990 | 0.003 |
| | | 100 | 0.997 | 0.001 | 0.992 | 0.002 | 0.994 | 0.001 | 0.996 | 0.001 | 0.991 | 0.002 | 0.993 | 0.002 |
| | | 300 | 0.997 | 0.000 | 0.996 | 0.001 | 0.996 | 0.001 | 0.996 | 0.000 | 0.995 | 0.001 | 0.996 | 0.001 |
| | 300 | 50 | 0.999 | 0.000 | 0.991 | 0.003 | 0.994 | 0.002 | 0.999 | 0.000 | 0.988 | 0.004 | 0.992 | 0.003 |
| | | 100 | 0.999 | 0.000 | 0.994 | 0.001 | 0.995 | 0.001 | 0.999 | 0.000 | 0.992 | 0.002 | 0.994 | 0.001 |
| | | 300 | 0.999 | 0.000 | 0.997 | 0.000 | 0.998 | 0.000 | 0.999 | 0.000 | 0.996 | 0.001 | 0.997 | 0.000 |
| 4 | 50 | 50 | 0.990 | 0.003 | 0.981 | 0.007 | 0.990 | 0.003 | 0.988 | 0.003 | 0.977 | 0.008 | 0.987 | 0.005 |
| | | 100 | 0.991 | 0.002 | 0.987 | 0.003 | 0.992 | 0.002 | 0.989 | 0.002 | 0.984 | 0.004 | 0.990 | 0.002 |
| | | 300 | 0.991 | 0.001 | 0.992 | 0.001 | 0.994 | 0.001 | 0.989 | 0.002 | 0.991 | 0.002 | 0.993 | 0.001 |
| | 100 | 50 | 0.995 | 0.001 | 0.985 | 0.004 | 0.993 | 0.002 | 0.995 | 0.001 | 0.981 | 0.006 | 0.991 | 0.002 |
| | | 100 | 0.996 | 0.001 | 0.990 | 0.002 | 0.995 | 0.001 | 0.995 | 0.001 | 0.988 | 0.003 | 0.994 | 0.001 |
| | | 300 | 0.996 | 0.000 | 0.995 | 0.001 | 0.997 | 0.000 | 0.995 | 0.001 | 0.994 | 0.001 | 0.996 | 0.001 |
| | 300 | 50 | 0.999 | 0.000 | 0.987 | 0.004 | 0.995 | 0.001 | 0.998 | 0.000 | 0.984 | 0.005 | 0.994 | 0.002 |
| | | 100 | 0.999 | 0.000 | 0.992 | 0.001 | 0.997 | 0.001 | 0.998 | 0.000 | 0.990 | 0.002 | 0.996 | 0.001 |
| | | 300 | 0.999 | 0.000 | 0.996 | 0.001 | 0.998 | 0.000 | 0.999 | 0.000 | 0.995 | 0.001 | 0.998 | 0.000 |

objective function directly on the observed samples:

$$\min_{\mathbf{F}, \mathbf{B}^{(1)}, \Delta\mathbf{B}} \sum_{k=1}^{2} \sum_{(i,j) \in \mathbf{\Omega}^{(k)}} \left( x_{ij}^{(k)} - m_{ij}^{(k)} \right)^2 + \lambda \sum_{j=1}^{p} \|\Delta\boldsymbol{b}_j\|_2$$

where $\mathbf{M}^{(1)} = \mathbf{F}(\mathbf{B}^{(1)})^\top$ and $\mathbf{M}^{(2)} = \mathbf{F}(\mathbf{B}^{(1)} + \Delta\mathbf{B})^\top$. This one-step approach circumvents the need for separate pre-imputation, providing a computationally efficient alternative for continuous data. Simulation results across different missing rates are presented in Tables 7–10.

## D.6 Numerical Validation of Error Bounds

We include an additional numerical check of the theoretical rates. For each configuration, we record the normalized estimation errors of $\hat{\mathbf{M}}$, $\hat{\mathbf{F}}$, $\hat{\mathbf{B}}$, and $\hat{\Delta\mathbf{B}}$ and plot them against the problem size on a log-log scale. Figure 18 reports two representative regimes: symmetric growth with $n = p$ and a regime with fixed $n = 500$ and varying $p$. The four curves are read against different theoretical orders, so we annotate them separately. The normalized errors of $\hat{\mathbf{M}}$, $\hat{\mathbf{F}}$, and $\hat{\mathbf{B}}$ decrease monotonically, matching the $1/(\sqrt{n} \wedge \sqrt{p})$ rate of Corollaries 3.4 and 3.5. The curve for $\hat{\Delta\mathbf{B}}$ is flat, and this is what the theory predicts rather than a shortfall of the bound. Under our data generating process the nonzero entries of $\Delta\mathbf{B}$ are drawn from a fixed distribution, so the per-row signal is of constant order, and Theorem 3.3 gives $s^{-1/2}\|\hat{\Delta\mathbf{B}}\mathbf{O} - \Delta\mathbf{B}^*\|_F = O_p(1)$. The row normalized quantity measures the error relative to that constant order signal and therefore does not

decrease with $n$ or $p$. The fixed-$n$ regime also illustrates the blessing-of-dimensionality behavior in factor models: increasing $p$ can improve latent-space recovery because more coordinates contribute information about the common representation.

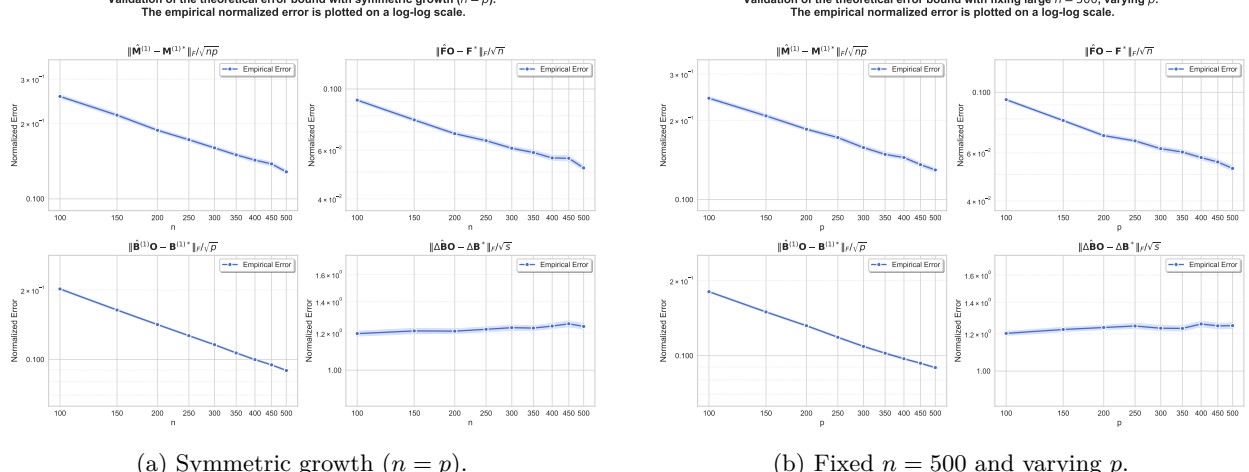

(a) Symmetric growth ($n = p$).        (b) Fixed $n = 500$ and varying $p$.

Figure 18: Numerical validation of the theoretical error-bound scaling. Empirical normalized errors are plotted on a log-log scale.

## D.7 Why Direct Tests on Observed Coordinates Are Insufficient

This subsection clarifies why SMART targets row support in $\Delta \mathbf{B}$ rather than marginal changes in $\mathbf{X}$. Consider the additive model $X_j = \boldsymbol{f}^\top \boldsymbol{b}_j + \varepsilon_j$ with $\boldsymbol{f}$ having a distribution symmetric under coordinate sign changes. Let a changed feature satisfy $\boldsymbol{b}_j^{(2)} = \boldsymbol{b}_j^{(1)} + \Delta \boldsymbol{b}_j$ with $\Delta \boldsymbol{b}_j \neq \mathbf{0}$, but choose $\boldsymbol{b}_j^{(2)}$ so that $\boldsymbol{f}^\top \boldsymbol{b}_j^{(1)}$ and $\boldsymbol{f}^\top \boldsymbol{b}_j^{(2)}$ have the same distribution. If the noise distribution is unchanged, then the entire marginal distribution of $X_j$ is identical across the two environments even though the structural mechanism has changed. Thus, mean tests, variance tests, and more general marginal distributional comparisons can miss mechanism shifts in principle.

We verified this phenomenon numerically using 100 repetitions. We compared two-sample $t$-tests for mean shifts, $F$-tests for variance shifts, their union, and SMART. Direct-test $p$-values were adjusted across variables by the Benjamini–Hochberg procedure. Table 11 shows that the direct tests have nearly zero precision and recall, whereas SMART recovers the mechanism support perfectly. Halving the noise level does not change this conclusion, indicating that the failure is structural rather than a lack of sample size or signal-to-noise ratio.

Table 11: Direct observed-coordinate tests can fail when the mechanism changes but the observed marginal distribution does not. Results are mean (standard deviation) over 100 repetitions.

| Noise level | Method | Precision | Recall |
|---|---|---|---|
| $\sigma = 1$ | $t$-test | 0.0075 (0.0750) | 0.0060 (0.0600) |
| $\sigma = 1$ | $F$-test | 0.0018 (0.0182) | 0.0010 (0.0100) |
| $\sigma = 1$ | Union | 0.0093 (0.0770) | 0.0070 (0.0607) |
| $\sigma = 1$ | SMART | 1.0000 (0.0000) | 1.0000 (0.0000) |
| $\sigma = 0.5$ | $t$-test | 0.0100 (0.1000) | 0.0100 (0.1000) |
| $\sigma = 0.5$ | $F$-test | 0.0000 (0.0000) | 0.0000 (0.0000) |
| $\sigma = 0.5$ | Union | 0.0100 (0.1000) | 0.0100 (0.1000) |
| $\sigma = 0.5$ | SMART | 1.0000 (0.0000) | 1.0000 (0.0000) |

## D.8 Multi-Environment Simulations

This subsection reports the full results for the multi-environment extension in Section 4. We vary the number of environments $K \in \{2, 3, 4, 5\}$ while keeping the same latent shift design. Table 14 shows that both one-step and two-stage estimators retain high union $F_1$ scores across all values of $K$. Tables 12 and 13 show that the one-step estimator often attains slightly smaller subspace error, but the difference becomes modest when $n$ and $p$ increase. Tables 15 and 16 confirm that model fitting time grows smoothly with $K$, and that the additional cost of the two-stage procedure mainly comes from Stage I signal recovery.

Table 12: Subspace error of **F** comparing one-step and two-stage methods, with varying $n$, $p$, and $K$ over 200 repetitions.

| | | $K = 2$ | | $K = 3$ | | $K = 4$ | | $K = 5$ | |
|---|---|---|---|---|---|---|---|---|---|
| $n$ | $p$ | one-step | two-stage | one-step | two-stage | one-step | two-stage | one-step | two-stage |
| 50 | 50 | 0.083 (0.010) | 0.083 (0.010) | **0.066** (0.009) | 0.07 (0.009) | **0.054** (0.007) | 0.071 (0.014) | **0.047** (0.006) | 0.074 (0.017) |
| | 100 | **0.058** (0.007) | 0.059 (0.007) | **0.047** (0.005) | 0.052 (0.007) | **0.04** (0.004) | 0.052 (0.010) | **0.035** (0.004) | 0.056 (0.014) |
| | 300 | **0.034** (0.003) | 0.036 (0.004) | **0.028** (0.003) | 0.032 (0.005) | **0.023** (0.003) | 0.032 (0.006) | **0.021** (0.002) | 0.036 (0.008) |
| 100 | 50 | **0.081** (0.008) | 0.082 (0.008) | **0.064** (0.007) | 0.069 (0.007) | **0.053** (0.006) | 0.068 (0.012) | **0.046** (0.005) | 0.074 (0.016) |
| | 100 | **0.057** (0.005) | 0.058 (0.005) | **0.046** (0.004) | 0.05 (0.006) | **0.039** (0.004) | 0.052 (0.010) | **0.034** (0.003) | 0.055 (0.014) |
| | 300 | **0.033** (0.003) | 0.036 (0.003) | **0.027** (0.002) | 0.034 (0.005) | **0.023** (0.002) | 0.035 (0.006) | **0.02** (0.002) | 0.038 (0.009) |
| 300 | 50 | **0.078** (0.006) | 0.079 (0.006) | **0.063** (0.006) | 0.068 (0.008) | **0.053** (0.005) | 0.068 (0.012) | **0.046** (0.004) | 0.075 (0.015) |
| | 100 | **0.056** (0.004) | 0.057 (0.004) | **0.045** (0.003) | 0.05 (0.005) | **0.038** (0.002) | 0.051 (0.010) | **0.033** (0.002) | 0.058 (0.014) |
| | 300 | **0.032** (0.002) | 0.033 (0.002) | **0.026** (0.001) | 0.028 (0.003) | **0.022** (0.001) | 0.03 (0.006) | **0.02** (0.001) | 0.034 (0.008) |

Table 13: Average subspace error of $\mathbf{B}^{(k)}$ comparing one-step and two-stage methods, with varying $n$, $p$, and $K$ over 200 repetitions.

| | | $K = 2$ | | $K = 3$ | | $K = 4$ | | $K = 5$ | |
|---|---|---|---|---|---|---|---|---|---|
| $n$ | $p$ | one-step | two-stage | one-step | two-stage | one-step | two-stage | one-step | two-stage |
| 50 | 50 | **0.144** (0.018) | 0.147 (0.017) | **0.181** (0.023) | 0.193 (0.021) | **0.194** (0.020) | 0.221 (0.020) | **0.204** (0.021) | 0.241 (0.021) |
| | 100 | **0.125** (0.014) | 0.128 (0.013) | **0.144** (0.015) | 0.158 (0.016) | **0.158** (0.014) | 0.181 (0.017) | **0.165** (0.015) | 0.201 (0.019) |
| | 300 | **0.099** (0.008) | 0.101 (0.008) | **0.103** (0.008) | 0.113 (0.009) | **0.107** (0.007) | 0.126 (0.012) | **0.111** (0.007) | 0.14 (0.016) |
| 100 | 50 | **0.129** (0.015) | 0.134 (0.015) | **0.172** (0.020) | 0.188 (0.019) | **0.19** (0.022) | 0.217 (0.021) | **0.198** (0.021) | 0.238 (0.021) |
| | 100 | **0.108** (0.010) | 0.113 (0.010) | **0.134** (0.012) | 0.149 (0.014) | **0.152** (0.013) | 0.179 (0.016) | **0.159** (0.012) | 0.196 (0.019) |
| | 300 | **0.08** (0.006) | 0.083 (0.007) | **0.091** (0.007) | 0.102 (0.010) | **0.098** (0.007) | 0.12 (0.013) | **0.103** (0.006) | 0.136 (0.015) |
| 300 | 50 | **0.119** (0.014) | 0.125 (0.014) | **0.168** (0.022) | 0.184 (0.020) | **0.184** (0.020) | 0.212 (0.020) | **0.198** (0.019) | 0.237 (0.020) |
| | 100 | **0.098** (0.009) | 0.104 (0.010) | **0.13** (0.012) | 0.147 (0.015) | **0.145** (0.012) | 0.173 (0.017) | **0.154** (0.013) | 0.195 (0.019) |
| | 300 | **0.066** (0.005) | 0.071 (0.007) | **0.083** (0.006) | 0.097 (0.009) | **0.096** (0.006) | 0.118 (0.013) | **0.104** (0.006) | 0.137 (0.015) |

Table 14: Variable selection performance (union $F_1$ score, identifying a variable if selected in at least one of the $K - 1$ shifts) comparing one-step and two-stage methods, with varying $n$, $p$, and $K$ over 200 repetitions.

| | | $K = 2$ | | $K = 3$ | | $K = 4$ | | $K = 5$ | |
|---|---|---|---|---|---|---|---|---|---|
| $n$ | $p$ | one-step | two-stage | one-step | two-stage | one-step | two-stage | one-step | two-stage |
| 50 | 50 | 0.923 (0.070) | 0.923 (0.071) | **0.978** (0.036) | 0.977 (0.037) | 0.986 (0.029) | **0.987** (0.027) | 0.979 (0.033) | **0.983** (0.030) |
| | 100 | **0.914** (0.075) | 0.912 (0.078) | **0.975** (0.039) | 0.974 (0.039) | 0.989 (0.025) | 0.989 (0.023) | 0.991 (0.019) | 0.991 (0.019) |
| | 300 | 0.909 (0.067) | 0.909 (0.068) | **0.974** (0.037) | 0.973 (0.038) | 0.988 (0.026) | 0.988 (0.025) | 0.989 (0.022) | **0.99** (0.022) |
| 100 | 50 | **0.924** (0.064) | 0.923 (0.064) | **0.984** (0.030) | 0.981 (0.033) | **0.993** (0.017) | 0.992 (0.020) | **0.999** (0.008) | 0.998 (0.009) |
| | 100 | **0.917** (0.070) | 0.916 (0.071) | **0.984** (0.031) | 0.982 (0.032) | 0.992 (0.021) | 0.992 (0.021) | 0.997 (0.012) | 0.997 (0.013) |
| | 300 | 0.919 (0.070) | **0.921** (0.067) | **0.981** (0.032) | 0.98 (0.032) | 0.992 (0.020) | 0.992 (0.020) | **0.996** (0.014) | 0.995 (0.015) |
| 300 | 50 | 0.932 (0.067) | 0.932 (0.068) | **0.985** (0.026) | 0.984 (0.027) | **0.994** (0.017) | 0.993 (0.018) | **0.999** (0.007) | 0.998 (0.009) |
| | 100 | **0.92** (0.067) | 0.919 (0.068) | **0.981** (0.035) | 0.98 (0.034) | 0.992 (0.020) | 0.992 (0.021) | 0.998 (0.010) | 0.998 (0.011) |
| | 300 | 0.922 (0.066) | 0.922 (0.066) | 0.975 (0.035) | 0.975 (0.035) | 0.99 (0.025) | **0.991** (0.022) | 0.994 (0.019) | 0.994 (0.019) |

Table 15: Computational time comparing one-step and two-stage methods, with varying $n$, $p$, and $K$ over 200 repetitions.

| | | $K = 2$ | | $K = 3$ | | $K = 4$ | | $K = 5$ | |
|---|---|---|---|---|---|---|---|---|---|
| $n$ | $p$ | one-step | two-stage | one-step | two-stage | one-step | two-stage | one-step | two-stage |
| 50 | 50 | **0.068** (0.019) | 0.084 (0.020) | **0.093** (0.038) | 0.115 (0.040) | **0.115** (0.055) | 0.147 (0.053) | **0.153** (0.072) | 0.2 (0.074) |
| | 100 | **0.038** (0.010) | 0.055 (0.012) | **0.067** (0.040) | 0.09 (0.039) | **0.09** (0.055) | 0.127 (0.055) | **0.125** (0.074) | 0.173 (0.075) |
| | 300 | **0.031** (0.007) | 0.062 (0.009) | **0.046** (0.010) | 0.095 (0.016) | **0.063** (0.014) | 0.129 (0.020) | **0.087** (0.026) | 0.176 (0.030) |
| 100 | 50 | **0.078** (0.018) | 0.096 (0.021) | **0.109** (0.049) | 0.141 (0.049) | **0.136** (0.060) | 0.185 (0.059) | **0.171** (0.075) | 0.228 (0.080) |
| | 100 | **0.05** (0.013) | 0.073 (0.013) | **0.095** (0.051) | 0.133 (0.053) | **0.115** (0.071) | 0.167 (0.070) | **0.173** (0.090) | 0.242 (0.090) |
| | 300 | **0.049** (0.009) | 0.106 (0.012) | **0.075** (0.017) | 0.161 (0.024) | **0.111** (0.028) | 0.223 (0.033) | **0.153** (0.041) | 0.292 (0.051) |
| 300 | 50 | **0.099** (0.023) | 0.129 (0.026) | **0.128** (0.050) | 0.178 (0.052) | **0.176** (0.070) | 0.239 (0.071) | **0.217** (0.089) | 0.295 (0.095) |
| | 100 | **0.075** (0.019) | 0.127 (0.021) | **0.136** (0.068) | 0.212 (0.070) | **0.209** (0.108) | 0.32 (0.112) | **0.277** (0.153) | 0.421 (0.159) |
| | 300 | **0.205** (0.037) | 0.384 (0.045) | **0.309** (0.061) | 0.588 (0.070) | **0.433** (0.101) | 0.859 (0.097) | **0.659** (0.191) | 1.262 (0.328) |

Table 16: Individual stage-wise model fitting time comparing one-step and two-stage methods, with varying $n$, $p$, and $K$ over 200 repetitions.

| | | K = 2 | | | K = 3 | | | K = 4 | | | K = 5 | | |
|---|---|---|---|---|---|---|---|---|---|---|---|---|---|
| $n$ | $p$ | one-step | two-stage stage 1 | two-stage stage 2 | one-step | two-stage stage 1 | two-stage stage 2 | one-step | two-stage stage 1 | two-stage stage 2 | one-step | two-stage stage 1 | two-stage stage 2 |
| 50 | 50 | 0.068 | 0.016 | 0.068 | 0.093 | 0.024 | **0.09** | 0.115 | 0.034 | **0.113** | **0.153** | 0.046 | 0.154 |
| | | (0.019) | (0.005) | (0.019) | (0.038) | (0.005) | (0.039) | (0.055) | (0.006) | (0.053) | (0.072) | (0.008) | (0.072) |
| | 100 | 0.038 | 0.018 | **0.037** | 0.067 | 0.027 | **0.063** | 0.09 | 0.039 | **0.088** | 0.125 | 0.048 | 0.125 |
| | | (0.010) | (0.005) | (0.010) | (0.040) | (0.006) | (0.037) | (0.055) | (0.008) | (0.053) | (0.074) | (0.009) | (0.074) |
| | 300 | 0.031 | 0.031 | 0.031 | 0.046 | 0.049 | 0.046 | 0.063 | 0.066 | 0.063 | **0.087** | 0.085 | 0.09 |
| | | (0.007) | (0.006) | (0.006) | (0.010) | (0.011) | (0.011) | (0.014) | (0.013) | (0.015) | (0.026) | (0.015) | (0.027) |
| 100 | 50 | 0.078 | 0.02 | **0.076** | 0.109 | 0.037 | **0.104** | **0.136** | 0.047 | 0.137 | 0.171 | 0.062 | **0.166** |
| | | (0.018) | (0.005) | (0.020) | (0.049) | (0.008) | (0.048) | (0.060) | (0.008) | (0.059) | (0.075) | (0.009) | (0.078) |
| | 100 | 0.05 | 0.026 | **0.048** | 0.095 | 0.042 | **0.091** | 0.115 | 0.058 | **0.11** | 0.173 | 0.073 | **0.169** |
| | | (0.013) | (0.004) | (0.012) | (0.051) | (0.008) | (0.052) | (0.071) | (0.008) | (0.069) | (0.090) | (0.009) | (0.089) |
| | 300 | 0.049 | 0.057 | **0.048** | **0.075** | 0.085 | 0.076 | 0.111 | 0.113 | 0.111 | 0.153 | 0.14 | **0.152** |
| | | (0.009) | (0.008) | (0.008) | (0.017) | (0.012) | (0.018) | (0.028) | (0.016) | (0.027) | (0.041) | (0.019) | (0.044) |
| 300 | 50 | 0.099 | 0.031 | **0.097** | **0.128** | 0.049 | 0.129 | 0.176 | 0.071 | **0.169** | 0.217 | 0.088 | **0.207** |
| | | (0.023) | (0.007) | (0.024) | (0.050) | (0.010) | (0.051) | (0.070) | (0.013) | (0.069) | (0.089) | (0.015) | (0.094) |
| | 100 | 0.075 | 0.056 | **0.072** | 0.136 | 0.085 | **0.127** | 0.209 | 0.116 | **0.204** | 0.277 | 0.147 | **0.273** |
| | | (0.019) | (0.008) | (0.017) | (0.068) | (0.012) | (0.068) | (0.108) | (0.014) | (0.110) | (0.153) | (0.020) | (0.154) |
| | 300 | 0.205 | 0.18 | **0.204** | 0.309 | 0.285 | **0.303** | **0.433** | 0.417 | 0.442 | **0.659** | 0.581 | 0.681 |
| | | (0.037) | (0.025) | (0.037) | (0.061) | (0.032) | (0.063) | (0.101) | (0.048) | (0.084) | (0.191) | (0.115) | (0.257) |

## D.9 Mixed Data Simulations

We also consider heterogeneous observations formed by combining Gaussian and Poisson coordinates under the same latent loading structure. This experiment examines whether the modular Stage I design remains effective when the observation matrix is not generated from a single likelihood family. Table 17 shows that SMART usually improves the support recovery $F_1$ score over N-Lasso, especially when sparsity is moderate or large. The additional tables show that recovery of the shared representation is essentially identical across methods, and that the loading subspaces recovered by SMART remain close to the unpenalized SVD benchmark while improving on or matching N-Lasso in most settings.

Table 17: Variable selection performance ($F_1$ Score) for mixed data, with varying $n$, $p$, $r^*$, and $s$ over 200 repetitions.

| | | | s = 5 | | | | s = 10 | | | | s = 20 | | | |
|---|---|---|---|---|---|---|---|---|---|---|---|---|---|---|
| | | | N-Lasso | | SMART | | N-Lasso | | SMART | | N-Lasso | | SMART | |
| $r^*$ | $n$ | $p$ | mean | std | mean | std | mean | std | mean | std | mean | std | mean | std |
| 2 | 50 | 50 | 0.897 | 0.147 | **0.915** | 0.122 | 0.931 | 0.087 | **0.947** | 0.073 | 0.955 | 0.049 | **0.965** | 0.038 |
| | | 100 | 0.912 | 0.125 | **0.930** | 0.105 | 0.910 | 0.112 | **0.930** | 0.092 | 0.940 | 0.066 | **0.955** | 0.052 |
| | | 300 | 0.886 | 0.137 | **0.917** | 0.114 | 0.901 | 0.100 | **0.927** | 0.080 | 0.899 | 0.098 | **0.923** | 0.075 |
| | 100 | 50 | 0.910 | 0.131 | **0.922** | 0.119 | 0.941 | 0.078 | **0.958** | 0.059 | 0.962 | 0.045 | **0.973** | 0.034 |
| | | 100 | 0.900 | 0.128 | **0.923** | 0.107 | 0.912 | 0.105 | **0.933** | 0.087 | 0.943 | 0.066 | **0.960** | 0.047 |
| | | 300 | 0.906 | 0.126 | **0.930** | 0.104 | 0.910 | 0.093 | **0.933** | 0.078 | 0.909 | 0.089 | **0.932** | 0.067 |
| | 300 | 50 | 0.912 | 0.137 | **0.927** | 0.119 | 0.941 | 0.080 | **0.961** | 0.058 | 0.962 | 0.051 | **0.972** | 0.040 |
| | | 100 | 0.898 | 0.141 | **0.923** | 0.114 | 0.910 | 0.109 | **0.931** | 0.089 | 0.944 | 0.064 | **0.960** | 0.049 |
| | | 300 | 0.896 | 0.132 | **0.925** | 0.104 | 0.902 | 0.104 | **0.924** | 0.084 | 0.907 | 0.088 | **0.929** | 0.067 |
| 3 | 50 | 50 | 0.964 | 0.078 | **0.981** | 0.050 | 0.971 | 0.048 | **0.980** | 0.034 | 0.962 | 0.072 | **0.975** | 0.029 |
| | | 100 | 0.954 | 0.088 | **0.977** | 0.052 | 0.965 | 0.057 | **0.981** | 0.036 | 0.972 | 0.034 | **0.982** | 0.024 |
| | | 300 | 0.949 | 0.090 | **0.961** | 0.094 | 0.953 | 0.066 | **0.972** | 0.052 | 0.958 | 0.050 | **0.977** | 0.032 |
| | 100 | 50 | 0.965 | 0.068 | **0.979** | 0.047 | 0.975 | 0.044 | **0.989** | 0.025 | 0.972 | 0.069 | **0.992** | 0.016 |
| | | 100 | 0.957 | 0.077 | **0.980** | 0.045 | 0.965 | 0.059 | **0.982** | 0.035 | 0.978 | 0.033 | **0.989** | 0.018 |
| | | 300 | 0.957 | 0.076 | **0.977** | 0.054 | 0.952 | 0.059 | **0.974** | 0.042 | 0.958 | 0.053 | **0.978** | 0.029 |
| | 300 | 50 | 0.961 | 0.080 | **0.978** | 0.055 | 0.978 | 0.043 | **0.991** | 0.024 | 0.984 | 0.044 | **0.995** | 0.013 |
| | | 100 | 0.957 | 0.082 | **0.976** | 0.053 | 0.967 | 0.055 | **0.982** | 0.035 | 0.981 | 0.029 | **0.991** | 0.018 |
| | | 300 | 0.955 | 0.079 | **0.975** | 0.056 | 0.954 | 0.064 | **0.978** | 0.039 | 0.961 | 0.049 | **0.981** | 0.029 |
| 4 | 50 | 50 | 0.976 | 0.057 | **0.977** | 0.050 | **0.971** | 0.041 | 0.962 | 0.048 | 0.932 | 0.114 | **0.953** | 0.047 |
| | | 100 | 0.962 | 0.070 | **0.969** | 0.057 | 0.968 | 0.045 | **0.970** | 0.044 | **0.971** | 0.030 | 0.961 | 0.036 |
| | | 300 | **0.938** | 0.086 | 0.881 | 0.135 | **0.957** | 0.051 | 0.946 | 0.057 | **0.961** | 0.034 | 0.961 | 0.036 |
| | 100 | 50 | 0.977 | 0.051 | **0.987** | 0.039 | 0.984 | 0.031 | **0.986** | 0.027 | 0.971 | 0.079 | **0.986** | 0.021 |
| | | 100 | 0.970 | 0.062 | **0.982** | 0.042 | 0.981 | 0.034 | **0.986** | 0.029 | 0.984 | 0.021 | 0.984 | 0.021 |
| | | 300 | **0.963** | 0.072 | 0.945 | 0.088 | 0.968 | 0.044 | **0.968** | 0.040 | 0.970 | 0.033 | **0.976** | 0.026 |
| | 300 | 50 | 0.982 | 0.051 | **0.993** | 0.028 | 0.989 | 0.026 | **0.994** | 0.019 | 0.983 | 0.062 | **0.995** | 0.011 |
| | | 100 | 0.974 | 0.063 | **0.992** | 0.031 | 0.982 | 0.035 | **0.995** | 0.018 | 0.991 | 0.016 | **0.995** | 0.011 |
| | | 300 | 0.974 | 0.070 | **0.985** | 0.046 | 0.974 | 0.046 | **0.991** | 0.023 | 0.982 | 0.026 | **0.991** | 0.015 |

Table 18: Subspace error of $\mathbf{F}$ for mixed data, with varying $n$, $p$, $r^*$, and $s$ over 200 repetitions.

| | | | s = 5 | | | | | | s = 10 | | | | | | s = 20 | | | | | |
| | | | SVD | | N-Lasso | | SMART | | SVD | | N-Lasso | | SMART | | SVD | | N-Lasso | | SMART | |
| $r^*$ | $n$ | $p$ | mean | std | mean | std | mean | std | mean | std | mean | std | mean | std | mean | std | mean | std | mean | std |
|---|---|---|---|---|---|---|---|---|---|---|---|---|---|---|---|---|---|---|---|---|
| 2 | 50 | 50 | 0.073 | 0.010 | 0.072 | 0.010 | 0.072 | 0.010 | 0.071 | 0.011 | 0.071 | 0.011 | 0.071 | 0.011 | 0.067 | 0.011 | 0.066 | 0.011 | 0.066 | 0.011 |
| | | 100 | 0.056 | 0.011 | 0.056 | 0.011 | 0.056 | 0.011 | 0.056 | 0.011 | 0.056 | 0.011 | 0.056 | 0.011 | 0.054 | 0.012 | 0.054 | 0.012 | 0.054 | 0.012 |
| | | 300 | 0.044 | 0.018 | 0.044 | 0.018 | 0.044 | 0.018 | 0.042 | 0.017 | 0.042 | 0.017 | 0.042 | 0.017 | 0.044 | 0.018 | 0.044 | 0.018 | 0.044 | 0.018 |
| | 100 | 50 | 0.071 | 0.008 | 0.071 | 0.008 | 0.071 | 0.008 | 0.069 | 0.008 | 0.068 | 0.008 | 0.068 | 0.008 | 0.065 | 0.010 | 0.065 | 0.010 | 0.065 | 0.010 |
| | | 100 | 0.051 | 0.005 | 0.051 | 0.005 | 0.051 | 0.005 | 0.050 | 0.005 | 0.050 | 0.005 | 0.050 | 0.005 | 0.049 | 0.007 | 0.049 | 0.007 | 0.049 | 0.007 |
| | | 300 | 0.037 | 0.011 | 0.037 | 0.011 | 0.037 | 0.011 | 0.037 | 0.012 | 0.037 | 0.012 | 0.037 | 0.012 | 0.038 | 0.013 | 0.038 | 0.013 | 0.038 | 0.013 |
| | 300 | 50 | 0.075 | 0.012 | 0.075 | 0.012 | 0.075 | 0.012 | 0.072 | 0.012 | 0.072 | 0.012 | 0.072 | 0.012 | 0.066 | 0.012 | 0.066 | 0.012 | 0.066 | 0.012 |
| | | 100 | 0.052 | 0.007 | 0.052 | 0.007 | 0.052 | 0.007 | 0.051 | 0.006 | 0.050 | 0.006 | 0.050 | 0.006 | 0.049 | 0.007 | 0.049 | 0.007 | 0.049 | 0.007 |
| | | 300 | 0.029 | 0.003 | 0.029 | 0.003 | 0.029 | 0.003 | 0.030 | 0.003 | 0.030 | 0.003 | 0.030 | 0.003 | 0.029 | 0.003 | 0.029 | 0.003 | 0.029 | 0.003 |
| 3 | 50 | 50 | 0.081 | 0.011 | 0.081 | 0.011 | 0.081 | 0.011 | 0.078 | 0.011 | 0.078 | 0.011 | 0.078 | 0.011 | 0.073 | 0.012 | 0.073 | 0.011 | 0.073 | 0.011 |
| | | 100 | 0.061 | 0.010 | 0.061 | 0.011 | 0.061 | 0.010 | 0.060 | 0.009 | 0.060 | 0.009 | 0.060 | 0.009 | 0.058 | 0.010 | 0.058 | 0.010 | 0.058 | 0.010 |
| | | 300 | **0.052** | 0.018 | 0.053 | 0.018 | 0.053 | 0.018 | **0.052** | 0.019 | 0.053 | 0.019 | 0.053 | 0.019 | **0.051** | 0.020 | 0.052 | 0.020 | 0.052 | 0.020 |
| | 100 | 50 | 0.079 | 0.010 | 0.079 | 0.010 | 0.079 | 0.010 | 0.075 | 0.010 | 0.075 | 0.010 | 0.075 | 0.010 | 0.070 | 0.010 | 0.070 | 0.010 | 0.070 | 0.010 |
| | | 100 | 0.056 | 0.006 | 0.056 | 0.006 | 0.056 | 0.006 | 0.054 | 0.006 | 0.054 | 0.006 | 0.054 | 0.006 | 0.052 | 0.006 | 0.052 | 0.006 | 0.052 | 0.006 |
| | | 300 | 0.044 | 0.013 | 0.044 | 0.013 | 0.044 | 0.013 | **0.044** | 0.012 | 0.044 | 0.013 | 0.044 | 0.013 | 0.043 | 0.013 | 0.043 | 0.013 | 0.043 | 0.013 |
| | 300 | 50 | **0.082** | 0.011 | 0.082 | 0.012 | 0.082 | 0.012 | 0.079 | 0.013 | 0.079 | 0.013 | 0.079 | 0.013 | **0.073** | 0.012 | 0.073 | 0.013 | 0.073 | 0.013 |
| | | 100 | 0.056 | 0.007 | 0.056 | 0.007 | 0.056 | 0.007 | 0.055 | 0.007 | 0.055 | 0.007 | 0.055 | 0.007 | 0.053 | 0.007 | 0.053 | 0.007 | 0.053 | 0.007 |
| | | 300 | 0.033 | 0.003 | 0.033 | 0.003 | 0.033 | 0.003 | 0.032 | 0.003 | 0.032 | 0.003 | 0.032 | 0.003 | 0.032 | 0.003 | 0.032 | 0.003 | 0.032 | 0.003 |
| 4 | 50 | 50 | 0.092 | 0.013 | 0.092 | 0.013 | 0.092 | 0.013 | 0.089 | 0.014 | 0.089 | 0.014 | 0.089 | 0.014 | 0.081 | 0.013 | 0.081 | 0.014 | 0.081 | 0.013 |
| | | 100 | 0.069 | 0.010 | 0.069 | 0.010 | 0.069 | 0.010 | **0.067** | 0.010 | 0.067 | 0.011 | 0.067 | 0.011 | 0.065 | 0.011 | 0.065 | 0.011 | 0.065 | 0.011 |
| | | 300 | **0.056** | 0.013 | 0.057 | 0.014 | 0.057 | 0.014 | **0.057** | 0.015 | 0.058 | 0.016 | 0.058 | 0.016 | **0.055** | 0.014 | 0.056 | 0.015 | 0.055 | 0.015 |
| | 100 | 50 | **0.089** | 0.012 | 0.089 | 0.013 | 0.089 | 0.013 | 0.085 | 0.012 | 0.085 | 0.013 | 0.085 | 0.012 | 0.079 | 0.012 | 0.079 | 0.012 | 0.079 | 0.012 |
| | | 100 | 0.067 | 0.009 | 0.067 | 0.009 | 0.067 | 0.009 | **0.064** | 0.008 | 0.065 | 0.008 | 0.065 | 0.008 | 0.062 | 0.009 | 0.062 | 0.009 | 0.062 | 0.009 |
| | | 300 | 0.051 | 0.009 | 0.051 | 0.009 | 0.051 | 0.009 | 0.050 | 0.009 | 0.050 | 0.009 | 0.050 | 0.009 | **0.048** | 0.009 | 0.049 | 0.009 | 0.049 | 0.009 |
| | 300 | 50 | 0.093 | 0.013 | 0.093 | 0.013 | 0.093 | 0.013 | **0.088** | 0.012 | 0.089 | 0.013 | 0.089 | 0.013 | 0.081 | 0.014 | 0.081 | 0.014 | 0.081 | 0.014 |
| | | 100 | 0.068 | 0.008 | 0.068 | 0.008 | 0.068 | 0.008 | **0.065** | 0.008 | 0.065 | 0.009 | 0.065 | 0.009 | 0.062 | 0.008 | 0.062 | 0.008 | 0.062 | 0.008 |
| | | 300 | 0.046 | 0.005 | 0.046 | 0.005 | 0.046 | 0.005 | 0.045 | 0.005 | 0.045 | 0.005 | 0.045 | 0.005 | 0.044 | 0.005 | 0.045 | 0.005 | 0.044 | 0.005 |

Table 19: Subspace error of $\mathbf{B}^{(1)}$ for mixed data, with varying $n$, $p$, $r^*$, and $s$ over 200 repetitions.

| | | | s = 5 | | | | | | s = 10 | | | | | | s = 20 | | | | | |
| | | | SVD | | N-Lasso | | SMART | | SVD | | N-Lasso | | SMART | | SVD | | N-Lasso | | SMART | |
| $r^*$ | $n$ | $p$ | mean | std | mean | std | mean | std | mean | std | mean | std | mean | std | mean | std | mean | std | mean | std |
|---|---|---|---|---|---|---|---|---|---|---|---|---|---|---|---|---|---|---|---|---|
| 2 | 50 | 50 | **0.092** | 0.018 | 0.135 | 0.023 | 0.119 | 0.019 | **0.093** | 0.018 | 0.156 | 0.028 | 0.130 | 0.024 | **0.093** | 0.018 | 0.175 | 0.039 | 0.142 | 0.029 |
| | | 100 | **0.096** | 0.021 | 0.118 | 0.014 | 0.106 | 0.013 | **0.097** | 0.021 | 0.133 | 0.019 | 0.117 | 0.017 | **0.096** | 0.021 | 0.151 | 0.024 | 0.127 | 0.021 |
| | | 300 | 0.103 | 0.028 | 0.097 | 0.011 | **0.092** | 0.011 | 0.101 | 0.026 | 0.109 | 0.012 | **0.099** | 0.011 | **0.104** | 0.029 | 0.123 | 0.014 | 0.112 | 0.013 |
| | 100 | 50 | **0.069** | 0.014 | 0.123 | 0.022 | 0.106 | 0.018 | **0.070** | 0.015 | 0.142 | 0.028 | 0.115 | 0.023 | **0.070** | 0.015 | 0.161 | 0.035 | 0.128 | 0.026 |
| | | 100 | **0.065** | 0.010 | 0.104 | 0.013 | 0.091 | 0.011 | **0.065** | 0.011 | 0.119 | 0.016 | 0.102 | 0.014 | **0.066** | 0.011 | 0.138 | 0.025 | 0.112 | 0.021 |
| | | 300 | **0.071** | 0.018 | 0.080 | 0.008 | 0.072 | 0.008 | **0.071** | 0.020 | 0.093 | 0.009 | 0.082 | 0.008 | **0.072** | 0.020 | 0.107 | 0.012 | 0.095 | 0.011 |
| | 300 | 50 | **0.075** | 0.038 | 0.119 | 0.024 | 0.102 | 0.023 | **0.074** | 0.038 | 0.138 | 0.030 | 0.111 | 0.027 | **0.071** | 0.033 | 0.155 | 0.038 | 0.123 | 0.031 |
| | | 100 | **0.055** | 0.021 | 0.096 | 0.013 | 0.084 | 0.012 | **0.054** | 0.020 | 0.112 | 0.017 | 0.094 | 0.017 | **0.055** | 0.021 | 0.130 | 0.023 | 0.103 | 0.021 |
| | | 300 | **0.041** | 0.008 | 0.067 | 0.007 | 0.058 | 0.005 | **0.042** | 0.007 | 0.081 | 0.006 | 0.070 | 0.005 | **0.042** | 0.008 | 0.097 | 0.010 | 0.083 | 0.009 |
| 3 | 50 | 50 | **0.106** | 0.021 | 0.149 | 0.023 | 0.119 | 0.018 | **0.107** | 0.022 | 0.173 | 0.032 | 0.129 | 0.023 | **0.107** | 0.022 | 0.197 | 0.042 | 0.140 | 0.026 |
| | | 100 | 0.110 | 0.023 | 0.128 | 0.017 | **0.108** | 0.014 | 0.110 | 0.022 | 0.146 | 0.020 | 0.117 | 0.016 | **0.108** | 0.021 | 0.170 | 0.027 | 0.127 | 0.020 |
| | | 300 | 0.124 | 0.033 | 0.109 | 0.012 | **0.099** | 0.013 | 0.125 | 0.033 | 0.120 | 0.013 | **0.105** | 0.014 | 0.125 | 0.035 | 0.136 | 0.016 | **0.115** | 0.015 |
| | 100 | 50 | **0.083** | 0.017 | 0.135 | 0.021 | 0.104 | 0.016 | **0.083** | 0.018 | 0.158 | 0.030 | 0.112 | 0.019 | **0.082** | 0.016 | 0.179 | 0.037 | 0.122 | 0.022 |
| | | 100 | **0.075** | 0.013 | 0.113 | 0.014 | 0.090 | 0.012 | **0.074** | 0.012 | 0.129 | 0.016 | 0.097 | 0.013 | **0.074** | 0.014 | 0.150 | 0.023 | 0.106 | 0.016 |
| | | 300 | 0.085 | 0.022 | 0.087 | 0.008 | **0.074** | 0.008 | 0.085 | 0.021 | 0.100 | 0.008 | **0.082** | 0.008 | 0.085 | 0.021 | 0.116 | 0.013 | 0.092 | 0.011 |
| | 300 | 50 | **0.089** | 0.031 | 0.129 | 0.023 | 0.097 | 0.018 | **0.093** | 0.034 | 0.154 | 0.032 | 0.108 | 0.025 | **0.092** | 0.033 | 0.172 | 0.039 | 0.118 | 0.027 |
| | | 100 | **0.063** | 0.018 | 0.104 | 0.015 | 0.080 | 0.012 | **0.063** | 0.017 | 0.120 | 0.016 | 0.087 | 0.014 | **0.064** | 0.018 | 0.141 | 0.023 | 0.096 | 0.017 |
| | | 300 | **0.049** | 0.007 | 0.073 | 0.007 | 0.056 | 0.005 | **0.049** | 0.007 | 0.087 | 0.006 | 0.066 | 0.005 | **0.049** | 0.007 | 0.102 | 0.011 | 0.077 | 0.008 |
| 4 | 50 | 50 | 0.135 | 0.031 | 0.167 | 0.030 | **0.129** | 0.021 | 0.138 | 0.035 | 0.199 | 0.035 | 0.142 | 0.026 | **0.137** | 0.032 | 0.231 | 0.047 | 0.154 | 0.029 |
| | | 100 | 0.135 | 0.028 | 0.141 | 0.018 | **0.116** | 0.016 | 0.138 | 0.029 | 0.163 | 0.021 | **0.126** | 0.018 | 0.138 | 0.030 | 0.190 | 0.028 | **0.136** | 0.021 |
| | | 300 | 0.153 | 0.034 | 0.120 | 0.014 | **0.109** | 0.015 | 0.153 | 0.034 | 0.132 | 0.015 | **0.114** | 0.015 | 0.151 | 0.032 | 0.147 | 0.015 | **0.121** | 0.015 |
| | 100 | 50 | **0.104** | 0.025 | 0.150 | 0.025 | 0.107 | 0.016 | **0.104** | 0.025 | 0.177 | 0.031 | 0.117 | 0.020 | **0.105** | 0.024 | 0.202 | 0.040 | 0.128 | 0.021 |
| | | 100 | 0.105 | 0.025 | 0.123 | 0.014 | **0.095** | 0.012 | 0.104 | 0.023 | 0.143 | 0.019 | **0.103** | 0.014 | **0.104** | 0.023 | 0.166 | 0.025 | 0.112 | 0.017 |
| | | 300 | 0.115 | 0.023 | 0.099 | 0.010 | **0.083** | 0.010 | 0.115 | 0.022 | 0.110 | 0.010 | **0.089** | 0.009 | 0.115 | 0.022 | 0.127 | 0.013 | **0.098** | 0.012 |
| | 300 | 50 | 0.100 | 0.025 | 0.139 | 0.023 | **0.097** | 0.017 | **0.100** | 0.024 | 0.167 | 0.031 | 0.106 | 0.021 | **0.098** | 0.024 | 0.190 | 0.040 | 0.118 | 0.022 |
| | | 100 | 0.083 | 0.017 | 0.110 | 0.015 | **0.080** | 0.011 | **0.081** | 0.015 | 0.131 | 0.016 | 0.088 | 0.011 | **0.081** | 0.014 | 0.152 | 0.024 | 0.095 | 0.014 |
| | | 300 | 0.079 | 0.011 | 0.081 | 0.007 | **0.061** | 0.005 | 0.079 | 0.013 | 0.094 | 0.008 | **0.069** | 0.006 | 0.080 | 0.011 | 0.111 | 0.010 | **0.079** | 0.008 |

Table 20: Subspace error of $\mathbf{B}^{(2)}$ for mixed data, with varying $n$, $p$, $r^*$, and $s$ over 200 repetitions.

| | | | s = 5 | | | | | | s = 10 | | | | | | s = 20 | | | | | |
| | | | SVD | | N-Lasso | | SMART | | SVD | | N-Lasso | | SMART | | SVD | | N-Lasso | | SMART | |
| $r^*$ | $n$ | $p$ | mean | std | mean | std | mean | std | mean | std | mean | std | mean | std | mean | std | mean | std | mean | std |
|---|---|---|---|---|---|---|---|---|---|---|---|---|---|---|---|---|---|---|---|---|
| 2 | 50 | 50 | **0.111** | 0.017 | 0.125 | 0.029 | 0.111 | 0.026 | **0.105** | 0.019 | 0.137 | 0.040 | 0.116 | 0.034 | **0.094** | 0.022 | 0.135 | 0.053 | 0.111 | 0.041 |
| | | 100 | 0.113 | 0.012 | 0.114 | 0.019 | **0.103** | 0.017 | **0.109** | 0.014 | 0.125 | 0.025 | 0.111 | 0.024 | **0.103** | 0.016 | 0.131 | 0.035 | 0.112 | 0.030 |
| | | 300 | 0.114 | 0.009 | 0.096 | 0.012 | **0.091** | 0.013 | 0.113 | 0.009 | 0.107 | 0.013 | **0.098** | 0.014 | 0.111 | 0.011 | 0.119 | 0.020 | **0.108** | 0.019 |
| | 100 | 50 | **0.078** | 0.012 | 0.114 | 0.028 | 0.098 | 0.026 | **0.074** | 0.015 | 0.122 | 0.037 | 0.100 | 0.032 | **0.068** | 0.018 | 0.121 | 0.052 | 0.097 | 0.040 |
| | | 100 | **0.080** | 0.008 | 0.101 | 0.017 | 0.089 | 0.016 | **0.078** | 0.011 | 0.112 | 0.025 | 0.096 | 0.024 | **0.074** | 0.015 | 0.120 | 0.038 | 0.098 | 0.033 |
| | | 300 | 0.080 | 0.006 | 0.079 | 0.009 | **0.072** | 0.010 | **0.079** | 0.006 | 0.093 | 0.011 | 0.082 | 0.012 | **0.078** | 0.009 | 0.103 | 0.019 | 0.092 | 0.018 |
| | 300 | 50 | **0.047** | 0.011 | 0.107 | 0.030 | 0.091 | 0.028 | **0.045** | 0.013 | 0.117 | 0.038 | 0.093 | 0.034 | **0.044** | 0.018 | 0.114 | 0.051 | 0.089 | 0.041 |
| | | 100 | **0.048** | 0.009 | 0.093 | 0.019 | 0.081 | 0.019 | **0.048** | 0.012 | 0.105 | 0.026 | 0.089 | 0.026 | **0.047** | 0.016 | 0.112 | 0.036 | 0.090 | 0.033 |
| | | 300 | **0.047** | 0.006 | 0.068 | 0.011 | 0.060 | 0.011 | **0.048** | 0.008 | 0.083 | 0.011 | 0.071 | 0.013 | **0.048** | 0.011 | 0.095 | 0.020 | 0.083 | 0.019 |
| 3 | 50 | 50 | 0.123 | 0.018 | 0.135 | 0.027 | **0.108** | 0.021 | 0.114 | 0.018 | 0.148 | 0.035 | **0.111** | 0.025 | **0.099** | 0.019 | 0.148 | 0.052 | 0.108 | 0.033 |
| | | 100 | 0.122 | 0.012 | 0.119 | 0.018 | **0.101** | 0.015 | 0.117 | 0.013 | 0.130 | 0.022 | **0.105** | 0.018 | 0.110 | 0.016 | 0.142 | 0.034 | **0.108** | 0.025 |
| | | 300 | 0.123 | 0.010 | 0.105 | 0.012 | **0.095** | 0.013 | 0.120 | 0.010 | 0.115 | 0.014 | **0.100** | 0.014 | 0.117 | 0.010 | 0.125 | 0.018 | **0.105** | 0.017 |
| | 100 | 50 | **0.084** | 0.011 | 0.120 | 0.025 | 0.091 | 0.019 | **0.078** | 0.012 | 0.131 | 0.035 | 0.092 | 0.024 | **0.069** | 0.013 | 0.134 | 0.049 | 0.091 | 0.031 |
| | | 100 | **0.084** | 0.008 | 0.105 | 0.017 | 0.084 | 0.013 | **0.081** | 0.008 | 0.116 | 0.021 | 0.088 | 0.017 | **0.075** | 0.010 | 0.125 | 0.031 | 0.088 | 0.022 |
| | | 300 | 0.083 | 0.005 | 0.084 | 0.008 | **0.071** | 0.009 | 0.082 | 0.006 | 0.095 | 0.010 | **0.077** | 0.010 | **0.080** | 0.006 | 0.107 | 0.015 | 0.085 | 0.013 |
| | 300 | 50 | **0.048** | 0.006 | 0.114 | 0.026 | 0.082 | 0.021 | **0.046** | 0.007 | 0.126 | 0.036 | 0.084 | 0.026 | **0.041** | 0.009 | 0.125 | 0.048 | 0.079 | 0.031 |
| | | 100 | **0.048** | 0.004 | 0.097 | 0.017 | 0.074 | 0.014 | **0.046** | 0.005 | 0.108 | 0.021 | 0.077 | 0.018 | **0.044** | 0.007 | 0.117 | 0.031 | 0.077 | 0.023 |
| | | 300 | **0.048** | 0.003 | 0.072 | 0.007 | 0.055 | 0.007 | **0.047** | 0.003 | 0.084 | 0.008 | 0.063 | 0.009 | **0.046** | 0.005 | 0.096 | 0.016 | 0.072 | 0.014 |
| 4 | 50 | 50 | 0.133 | 0.019 | 0.146 | 0.028 | **0.112** | 0.019 | 0.125 | 0.020 | 0.164 | 0.036 | **0.115** | 0.024 | **0.110** | 0.021 | 0.167 | 0.056 | 0.111 | 0.030 |
| | | 100 | 0.132 | 0.014 | 0.130 | 0.019 | **0.106** | 0.015 | 0.127 | 0.014 | 0.145 | 0.023 | **0.109** | 0.016 | 0.118 | 0.016 | 0.155 | 0.035 | **0.109** | 0.023 |
| | | 300 | 0.130 | 0.011 | 0.116 | 0.013 | **0.103** | 0.014 | 0.128 | 0.011 | 0.124 | 0.015 | **0.106** | 0.015 | 0.123 | 0.011 | 0.133 | 0.016 | **0.108** | 0.015 |
| | 100 | 50 | 0.091 | 0.011 | 0.130 | 0.025 | **0.090** | 0.016 | **0.085** | 0.013 | 0.147 | 0.035 | 0.094 | 0.022 | **0.074** | 0.013 | 0.148 | 0.050 | 0.090 | 0.028 |
| | | 100 | 0.089 | 0.007 | 0.112 | 0.017 | **0.086** | 0.012 | **0.085** | 0.008 | 0.124 | 0.021 | 0.087 | 0.014 | **0.079** | 0.010 | 0.135 | 0.032 | 0.087 | 0.020 |
| | | 300 | 0.087 | 0.006 | 0.093 | 0.009 | **0.078** | 0.009 | 0.086 | 0.006 | 0.103 | 0.011 | **0.082** | 0.009 | 0.083 | 0.006 | 0.114 | 0.015 | 0.086 | 0.012 |
| | 300 | 50 | **0.051** | 0.006 | 0.119 | 0.025 | 0.078 | 0.018 | **0.048** | 0.007 | 0.136 | 0.034 | 0.080 | 0.022 | **0.043** | 0.008 | 0.136 | 0.048 | 0.076 | 0.027 |
| | | 100 | **0.050** | 0.004 | 0.100 | 0.017 | 0.071 | 0.012 | **0.049** | 0.004 | 0.115 | 0.020 | 0.074 | 0.014 | **0.045** | 0.005 | 0.124 | 0.030 | 0.073 | 0.019 |
| | | 300 | **0.048** | 0.002 | 0.078 | 0.007 | 0.058 | 0.006 | **0.048** | 0.003 | 0.088 | 0.009 | 0.063 | 0.007 | **0.047** | 0.003 | 0.100 | 0.013 | 0.069 | 0.010 |

## D.10 Additional Comparison Between One-Step and Two-Stage Estimation

We also conducted additional comparisons between the one-step estimator and the two-stage SMART estimator for Gaussian data. The two methods use the same rowwise group penalty and the same CV protocol for $\lambda$; the difference is whether signal recovery is performed before attribution. The results support a simple practical guideline. When the data are Gaussian with low noise, one-step estimation can slightly reduce subspace error because it avoids the extra Stage I estimation step. As the noise level increases, the performance gap becomes negligible. For example, under $n = 100$ and $p = 100$, the subspace error of $\mathbf{B}^{(2)}$ is 0.082 for one-step and 0.096 for two-stage at $\sigma = 0.1$, whereas it becomes 0.142 and 0.151 at $\sigma = 2.0$. The $F_1$ scores remain similar across noise levels.

The computational comparison is also consistent with the methodological trade-off. The one-step method is faster in the Gaussian setting because it avoids Stage I; for instance, under $n = 300$, $p = 300$, and $\sigma = 1.0$, the model fitting time is 0.394 seconds for one-step and 0.831 seconds for two-stage in our implementation. However, the two-stage approach is more flexible: it covers Poisson, Binary, mixed data, and settings with missing values through the choice of Stage I estimator, while the one-step formulation in Section B is specialized to additive Gaussian observations. Additional simulations varying $r^*$, $s$, signal strength $c$, and the number of environments $K$ show qualitatively similar conclusions.

Tables 21–25 report the full Gaussian comparison under varying noise levels. Together they support the role of the two-stage estimator as the default, broadly applicable version of SMART, while the one-step estimator is a useful efficient specialization for low noise additive data.

Table 21: Subspace error of $\mathbf{F}$ comparing one-step and two-stage methods, with varying $n$, $p$, and $\sigma$ over 200 repetitions.

| | | $\sigma = 0.1$ | | $\sigma = 0.2$ | | $\sigma = 0.5$ | | $\sigma = 1.0$ | | $\sigma = 1.5$ | | $\sigma = 2.0$ | |
| $n$ | $p$ | one-step | two-stage | one-step | two-stage | one-step | two-stage | one-step | two-stage | one-step | two-stage | one-step | two-stage |
|---|---|---|---|---|---|---|---|---|---|---|---|---|---|
| 50 | 50 | **0.008** (0.001) | 0.017 (0.010) | **0.016** (0.002) | 0.022 (0.007) | **0.042** (0.006) | 0.044 (0.007) | **0.084** (0.011) | 0.085 (0.011) | 0.123 (0.015) | 0.123 (0.015) | 0.168 (0.023) | **0.168** (0.022) |
| | 100 | **0.006** (0.001) | 0.015 (0.008) | **0.012** (0.001) | 0.018 (0.005) | **0.029** (0.003) | 0.033 (0.005) | **0.059** (0.007) | 0.061 (0.007) | **0.088** (0.011) | 0.089 (0.011) | **0.120** (0.013) | 0.121 (0.013) |
| | 300 | **0.003** (0.000) | 0.013 (0.004) | **0.007** (0.001) | 0.014 (0.004) | **0.017** (0.002) | 0.021 (0.004) | **0.034** (0.004) | 0.036 (0.004) | **0.051** (0.006) | 0.053 (0.006) | **0.068** (0.007) | 0.069 (0.007) |
| 100 | 50 | **0.008** (0.001) | 0.017 (0.011) | **0.016** (0.002) | 0.023 (0.008) | **0.040** (0.004) | 0.043 (0.005) | **0.080** (0.009) | 0.081 (0.008) | 0.120 (0.012) | 0.120 (0.012) | 0.162 (0.015) | 0.162 (0.015) |
| | 100 | **0.006** (0.000) | 0.013 (0.007) | **0.011** (0.001) | 0.016 (0.006) | **0.028** (0.002) | 0.030 (0.003) | **0.057** (0.005) | 0.058 (0.005) | 0.085 (0.007) | 0.085 (0.007) | **0.115** (0.010) | 0.116 (0.010) |
| | 300 | **0.003** (0.000) | 0.017 (0.005) | **0.007** (0.001) | 0.018 (0.005) | **0.016** (0.001) | 0.023 (0.004) | **0.033** (0.003) | 0.036 (0.004) | **0.050** (0.004) | 0.052 (0.005) | **0.066** (0.005) | 0.067 (0.006) |
| 300 | 50 | **0.008** (0.001) | 0.018 (0.010) | **0.016** (0.001) | 0.023 (0.008) | **0.040** (0.003) | 0.042 (0.005) | **0.079** (0.006) | 0.080 (0.006) | 0.118 (0.009) | 0.118 (0.009) | 0.157 (0.014) | **0.156** (0.014) |
| | 100 | **0.006** (0.000) | 0.014 (0.008) | **0.011** (0.001) | 0.018 (0.007) | **0.028** (0.002) | 0.030 (0.004) | **0.056** (0.003) | 0.057 (0.003) | 0.084 (0.005) | 0.084 (0.005) | 0.111 (0.007) | 0.111 (0.007) |
| | 300 | **0.003** (0.000) | 0.006 (0.004) | **0.006** (0.000) | 0.009 (0.003) | **0.016** (0.001) | 0.017 (0.002) | **0.032** (0.002) | 0.033 (0.002) | 0.048 (0.002) | 0.048 (0.002) | 0.064 (0.003) | 0.064 (0.003) |

Table 22: Subspace error of $\mathbf{B}^{(1)}$ comparing one-step and two-stage methods, with varying $n$, $p$, and $\sigma$ over 200 repetitions.

| | | $\sigma = 0.1$ | | $\sigma = 0.2$ | | $\sigma = 0.5$ | | $\sigma = 1.0$ | | $\sigma = 1.5$ | | $\sigma = 2.0$ | |
|---|---|---|---|---|---|---|---|---|---|---|---|---|---|
| $n$ | $p$ | one-step | two-stage | one-step | two-stage | one-step | two-stage | one-step | two-stage | one-step | two-stage | one-step | two-stage |
| 50 | 50 | **0.104** (0.016) | 0.105 (0.016) | 0.105 (0.016) | 0.105 (0.016) | **0.114** (0.019) | 0.115 (0.019) | 0.138 (0.021) | 0.138 (0.021) | **0.168** (0.024) | 0.169 (0.024) | **0.209** (0.033) | 0.210 (0.033) |
| | 100 | **0.091** (0.010) | 0.092 (0.010) | 0.093 (0.012) | 0.093 (0.012) | **0.101** (0.014) | 0.102 (0.013) | **0.124** (0.014) | 0.125 (0.014) | **0.155** (0.019) | 0.156 (0.019) | **0.191** (0.021) | 0.192 (0.021) |
| | 300 | 0.059 (0.006) | 0.059 (0.006) | 0.061 (0.007) | 0.061 (0.007) | 0.071 (0.007) | 0.071 (0.007) | 0.101 (0.009) | 0.101 (0.009) | 0.136 (0.011) | 0.136 (0.011) | 0.174 (0.013) | 0.174 (0.013) |
| 100 | 50 | **0.101** (0.014) | 0.102 (0.014) | **0.100** (0.014) | 0.101 (0.014) | **0.104** (0.014) | 0.105 (0.014) | 0.118 (0.016) | 0.118 (0.016) | 0.138 (0.018) | **0.138** (0.017) | 0.161 (0.023) | 0.161 (0.023) |
| | 100 | **0.089** (0.009) | 0.090 (0.009) | 0.090 (0.009) | 0.090 (0.009) | 0.094 (0.010) | 0.094 (0.010) | **0.106** (0.011) | 0.107 (0.011) | 0.124 (0.013) | 0.124 (0.013) | **0.148** (0.016) | 0.149 (0.016) |
| | 300 | 0.059 (0.006) | 0.059 (0.006) | 0.059 (0.006) | 0.059 (0.006) | 0.064 (0.006) | 0.064 (0.006) | 0.080 (0.006) | 0.080 (0.006) | 0.102 (0.007) | 0.102 (0.007) | **0.127** (0.009) | 0.128 (0.009) |
| 300 | 50 | **0.100** (0.012) | 0.101 (0.012) | **0.100** (0.012) | 0.101 (0.012) | **0.103** (0.013) | 0.104 (0.013) | **0.106** (0.013) | 0.107 (0.013) | 0.113 (0.014) | 0.113 (0.014) | 0.124 (0.017) | 0.124 (0.017) |
| | 100 | **0.088** (0.008) | 0.089 (0.008) | **0.089** (0.009) | 0.091 (0.009) | **0.089** (0.008) | 0.091 (0.008) | **0.094** (0.008) | 0.095 (0.008) | **0.102** (0.011) | 0.103 (0.011) | **0.111** (0.011) | 0.112 (0.011) |
| | 300 | 0.057 (0.004) | 0.057 (0.004) | 0.057 (0.004) | 0.057 (0.004) | **0.059** (0.005) | 0.060 (0.005) | 0.065 (0.005) | 0.065 (0.005) | 0.074 (0.005) | 0.074 (0.005) | 0.086 (0.006) | 0.086 (0.006) |

Table 23: Subspace error of $\mathbf{B}^{(2)}$ comparing one-step and two-stage methods, with varying $n$, $p$, and $\sigma$ over 200 repetitions.

| | | $\sigma = 0.1$ | | $\sigma = 0.2$ | | $\sigma = 0.5$ | | $\sigma = 1.0$ | | $\sigma = 1.5$ | | $\sigma = 2.0$ | |
|---|---|---|---|---|---|---|---|---|---|---|---|---|---|
| $n$ | $p$ | one-step | two-stage | one-step | two-stage | one-step | two-stage | one-step | two-stage | one-step | two-stage | one-step | two-stage |
| 50 | 50 | **0.092** (0.016) | 0.103 (0.017) | **0.092** (0.017) | 0.102 (0.017) | **0.102** (0.018) | 0.111 (0.019) | **0.126** (0.021) | 0.133 (0.021) | **0.157** (0.024) | 0.163 (0.024) | **0.196** (0.029) | 0.201 (0.028) |
| | 100 | **0.085** (0.010) | 0.095 (0.016) | **0.087** (0.013) | 0.095 (0.013) | **0.093** (0.013) | 0.104 (0.014) | **0.118** (0.014) | 0.126 (0.016) | **0.150** (0.017) | 0.157 (0.017) | **0.183** (0.020) | 0.189 (0.019) |
| | 300 | **0.056** (0.006) | 0.066 (0.012) | **0.058** (0.007) | 0.066 (0.012) | **0.069** (0.007) | 0.077 (0.012) | **0.099** (0.008) | 0.104 (0.009) | **0.135** (0.011) | 0.138 (0.011) | **0.172** (0.013) | 0.175 (0.013) |
| 100 | 50 | **0.090** (0.015) | 0.100 (0.016) | **0.089** (0.014) | 0.099 (0.016) | **0.093** (0.014) | 0.104 (0.015) | **0.106** (0.017) | 0.115 (0.018) | **0.124** (0.019) | 0.134 (0.020) | **0.148** (0.020) | 0.157 (0.020) |
| | 100 | **0.082** (0.009) | 0.096 (0.014) | **0.083** (0.010) | 0.095 (0.014) | **0.087** (0.010) | 0.098 (0.013) | **0.100** (0.011) | 0.110 (0.014) | **0.118** (0.013) | 0.127 (0.014) | **0.142** (0.016) | 0.151 (0.015) |
| | 300 | **0.056** (0.006) | 0.064 (0.011) | **0.056** (0.005) | 0.067 (0.012) | **0.061** (0.006) | 0.070 (0.012) | **0.077** (0.006) | 0.085 (0.010) | **0.101** (0.007) | 0.106 (0.009) | **0.125** (0.008) | 0.130 (0.010) |
| 300 | 50 | **0.089** (0.013) | 0.100 (0.014) | **0.089** (0.013) | 0.101 (0.015) | **0.091** (0.014) | 0.103 (0.016) | **0.095** (0.014) | 0.105 (0.015) | **0.102** (0.015) | 0.112 (0.016) | **0.113** (0.018) | 0.125 (0.019) |
| | 100 | **0.081** (0.008) | 0.094 (0.013) | **0.082** (0.008) | 0.097 (0.015) | **0.082** (0.008) | 0.095 (0.013) | **0.087** (0.008) | 0.100 (0.014) | **0.095** (0.010) | 0.109 (0.015) | **0.104** (0.011) | 0.115 (0.014) |
| | 300 | **0.054** (0.004) | 0.067 (0.011) | **0.055** (0.004) | 0.069 (0.012) | **0.057** (0.005) | 0.070 (0.011) | **0.062** (0.005) | 0.076 (0.011) | **0.072** (0.004) | 0.083 (0.010) | **0.084** (0.005) | 0.093 (0.009) |

Table 24: Variable selection performance ($F_1$ Score) comparing one-step and two-stage methods, with varying $n$, $p$, and $\sigma$ over 200 repetitions.

| | | $\sigma = 0.1$ | | $\sigma = 0.2$ | | $\sigma = 0.5$ | | $\sigma = 1.0$ | | $\sigma = 1.5$ | | $\sigma = 2.0$ | |
|---|---|---|---|---|---|---|---|---|---|---|---|---|---|
| $n$ | $p$ | one-step | two-stage | one-step | two-stage | one-step | two-stage | one-step | two-stage | one-step | two-stage | one-step | two-stage |
| 50 | 50 | **0.946** (0.059) | 0.945 (0.059) | **0.944** (0.059) | 0.943 (0.060) | **0.949** (0.063) | 0.948 (0.063) | **0.941** (0.063) | 0.941 (0.064) | **0.866** (0.074) | 0.864 (0.075) | **0.702** (0.077) | 0.690 (0.078) |
| | 100 | **0.918** (0.073) | 0.917 (0.073) | **0.911** (0.077) | 0.910 (0.078) | **0.908** (0.077) | 0.906 (0.077) | 0.920 (0.071) | 0.920 (0.071) | **0.874** (0.086) | 0.872 (0.086) | **0.649** (0.092) | 0.642 (0.097) |
| | 300 | 0.912 (0.074) | **0.913** (0.071) | **0.923** (0.071) | 0.921 (0.071) | 0.917 (0.065) | 0.917 (0.065) | 0.916 (0.080) | **0.916** (0.077) | **0.836** (0.140) | 0.824 (0.149) | **0.457** (0.139) | 0.446 (0.137) |
| 100 | 50 | **0.946** (0.055) | 0.944 (0.055) | **0.948** (0.063) | 0.947 (0.063) | **0.952** (0.056) | 0.950 (0.057) | **0.948** (0.056) | 0.948 (0.057) | **0.944** (0.060) | 0.942 (0.059) | **0.886** (0.082) | 0.880 (0.080) |
| | 100 | **0.918** (0.070) | 0.917 (0.071) | **0.918** (0.068) | 0.915 (0.072) | **0.915** (0.073) | 0.913 (0.073) | **0.912** (0.069) | 0.912 (0.070) | **0.913** (0.072) | 0.912 (0.073) | **0.906** (0.070) | 0.902 (0.072) |
| | 300 | **0.915** (0.067) | 0.914 (0.066) | 0.914 (0.071) | 0.914 (0.071) | **0.918** (0.068) | 0.917 (0.068) | 0.913 (0.065) | **0.914** (0.065) | 0.910 (0.072) | **0.910** (0.071) | **0.874** (0.109) | 0.862 (0.123) |
| 300 | 50 | **0.943** (0.063) | 0.941 (0.066) | **0.944** (0.062) | 0.942 (0.064) | **0.945** (0.062) | 0.942 (0.063) | **0.943** (0.056) | 0.942 (0.059) | **0.946** (0.058) | 0.945 (0.059) | 0.944 (0.055) | **0.945** (0.054) |
| | 100 | **0.921** (0.072) | 0.919 (0.073) | **0.926** (0.070) | 0.922 (0.071) | **0.915** (0.072) | 0.913 (0.073) | **0.925** (0.064) | 0.923 (0.063) | **0.923** (0.070) | 0.922 (0.070) | **0.920** (0.074) | 0.919 (0.075) |
| | 300 | **0.926** (0.059) | 0.923 (0.061) | **0.914** (0.066) | 0.911 (0.068) | **0.923** (0.066) | 0.920 (0.066) | **0.916** (0.069) | 0.914 (0.070) | 0.915 (0.074) | **0.916** (0.071) | **0.914** (0.066) | 0.913 (0.067) |

Table 25: Computational time comparing one-step and two-stage methods, with varying $n$, $p$, and $\sigma$ over 200 repetitions.

| | | $\sigma = 0.1$ | | $\sigma = 0.2$ | | $\sigma = 0.5$ | | $\sigma = 1.0$ | | $\sigma = 1.5$ | | $\sigma = 2.0$ | |
|---|---|---|---|---|---|---|---|---|---|---|---|---|---|
| $n$ | $p$ | one-step | two-stage | one-step | two-stage | one-step | two-stage | one-step | two-stage | one-step | two-stage | one-step | two-stage |
| 50 | 50 | **0.150** (0.009) | 0.182 (0.015) | **0.164** (0.017) | 0.197 (0.021) | **0.163** (0.015) | 0.196 (0.017) | **0.177** (0.019) | 0.211 (0.022) | **0.184** (0.025) | 0.219 (0.028) | **0.196** (0.045) | 0.235 (0.052) |
| | 100 | **0.104** (0.033) | 0.146 (0.040) | **0.108** (0.036) | 0.154 (0.040) | **0.112** (0.044) | 0.160 (0.055) | **0.125** (0.053) | 0.171 (0.057) | **0.130** (0.054) | 0.178 (0.068) | **0.216** (0.093) | 0.282 (0.109) |
| | 300 | **0.096** (0.037) | 0.205 (0.070) | **0.098** (0.040) | 0.205 (0.072) | **0.099** (0.038) | 0.208 (0.071) | **0.101** (0.040) | 0.206 (0.073) | **0.101** (0.048) | 0.203 (0.076) | **0.088** (0.054) | 0.200 (0.080) |
| 100 | 50 | **0.262** (0.095) | 0.327 (0.119) | **0.259** (0.092) | 0.316 (0.111) | **0.257** (0.085) | 0.313 (0.098) | **0.263** (0.091) | 0.297 (0.089) | **0.258** (0.081) | 0.310 (0.090) | **0.206** (0.027) | 0.254 (0.033) |
| | 100 | **0.120** (0.026) | 0.182 (0.037) | **0.122** (0.029) | 0.178 (0.034) | **0.125** (0.030) | 0.179 (0.034) | **0.130** (0.038) | 0.186 (0.042) | **0.127** (0.036) | 0.178 (0.034) | **0.135** (0.036) | 0.189 (0.043) |
| | 300 | **0.125** (0.025) | 0.294 (0.052) | **0.128** (0.027) | 0.293 (0.047) | **0.125** (0.024) | 0.291 (0.050) | **0.125** (0.029) | 0.292 (0.046) | **0.126** (0.029) | 0.298 (0.046) | **0.127** (0.032) | 0.289 (0.046) |
| 300 | 50 | **0.310** (0.046) | 0.403 (0.060) | **0.314** (0.052) | 0.411 (0.070) | **0.328** (0.064) | 0.414 (0.070) | **0.307** (0.038) | 0.391 (0.046) | **0.302** (0.042) | 0.391 (0.042) | **0.303** (0.034) | 0.396 (0.044) |
| | 100 | **0.220** (0.049) | 0.371 (0.054) | **0.227** (0.054) | 0.377 (0.059) | **0.225** (0.055) | 0.379 (0.058) | **0.222** (0.049) | 0.382 (0.065) | **0.228** (0.052) | 0.377 (0.061) | **0.225** (0.048) | 0.370 (0.055) |
| | 300 | **0.389** (0.053) | 0.810 (0.080) | **0.395** (0.063) | 0.840 (0.097) | **0.398** (0.053) | 0.844 (0.092) | **0.394** (0.060) | 0.831 (0.092) | **0.384** (0.051) | 0.790 (0.087) | **0.372** (0.049) | 0.740 (0.097) |

# E  Additional Results on Real Data

## E.1  Regularization Parameter Selection

In real-world applications, identifying the most interpretable model structure is often prioritized over minimizing pure prediction error. Consequently, we adopted a heuristic approach rather than CV. Specifically, we determined the search range for $\lambda$ via preliminary experiments to cover the full solution path. We then selected 100 points uniformly within this range as the candidate grid of $\lambda$ values. For the latent rank $r$, we performed a preliminary assessment based on $IC_2$ criterion (range 1–8). After fixing $r$, we plotted the number of non-zero features against $\lambda$ and adopted the "elbow method" to select the $\lambda$ where the curve's slope shifts from a sharp decline to a gradual plateau. As shown in Figure 19 and Figure 20, both datasets exhibit clear elbow structures.

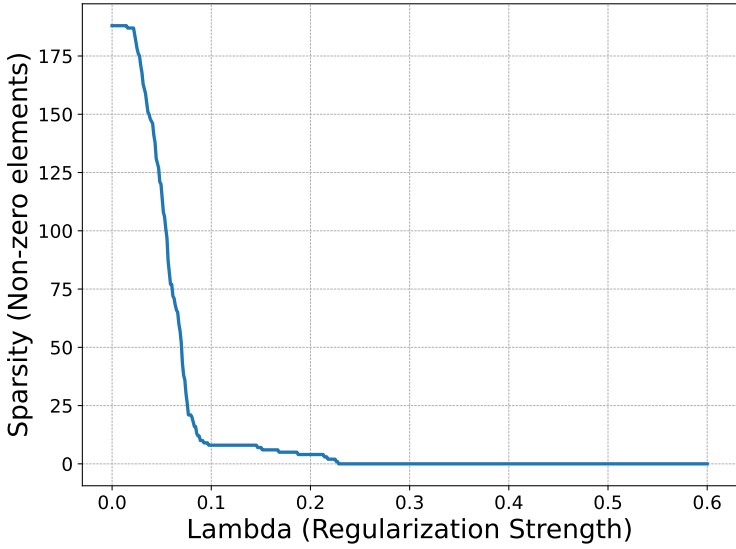

Figure 19: Sparsity varying with $\lambda$ for the NBA dataset: The number of non-zero elements as a function of regularization strength.

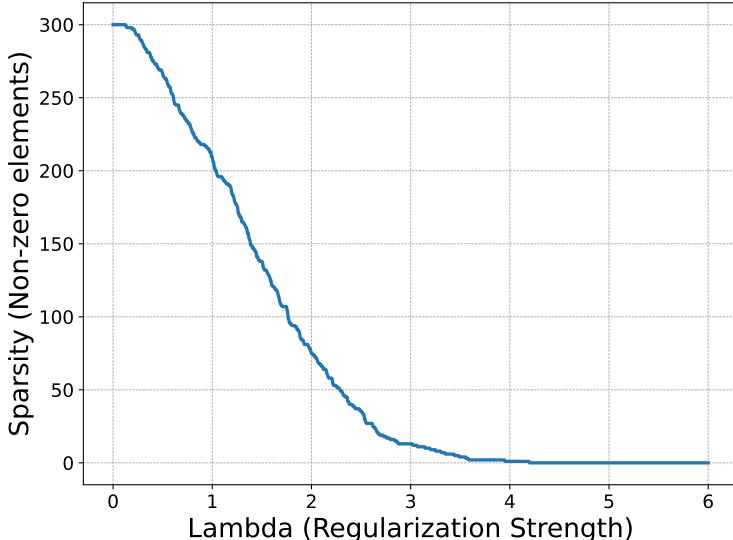

Figure 20: Sparsity varying with $\lambda$ for the TCGA dataset: The number of non-zero elements as a function of regularization strength.

## E.2 Additional TCGA Data Analysis

The remaining selected genes provide a broader biological validation of the structural shifts detected by SMART. Several markers have tumor suppression or epigenetic roles. `WIF1` is a frequent target of epigenetic silencing in breast cancer; its inactivation removes inhibition of the Wnt signaling pathway and thereby promotes oncogenesis Ai et al. (2006). `CLCA4` has been characterized as a tumor suppressor whose loss facilitates cancer progression and metastasis Yu et al. (2013), while `CMTM5` is reported as a putative suppressor in breast tissue Zhou et al. (2020). `LINC01198` has also been documented to play a dual role as both a tumor suppressor and a promoter of anti-tumor immune responses Fonseca-Montaño et al. (2023).

SMART also selects genes linked to malignancy and metabolic adaptation. `FABP7` supports breast cancer cell survival and proliferation within the brain microenvironment Cordero et al. (2019). The aberrant activation of the `DLK1-DIO3` locus, particularly its noncoding RNAs, is closely linked to epithelial mesenchymal transition, metastasis, and poor prognosis Budkova et al. (2020). Broadly altered expression profiles were also detected for `NEUROG2` Laisné et al. (2021) and `KCNJ16`, the latter of which shows significant alterations in pan-cancer analyses Zhu et al. (2022). Collectively, these findings confirm that the rows selected in $\Delta\mathbf{B}$ correspond to biologically meaningful drivers and indicators of breast cancer pathology.

## E.3 NBA 2023-24 Season Analysis

Professional basketball (NBA) represents a complex, high stakes dynamic system where team strategies and player incentives evolve significantly over the course of a season. The All-Star break traditionally serves as a structural inflection point, marking the transition from the regular season to a more intense playoff environment. We apply SMART to continuous player performance metrics to validate the model's ability to detect these latent strategic shifts ("gameplay evolution").

We analyzed paired data comparing the environments before and after the All-Star break. Following the selection strategy, we set the latent rank $r = 3$ and the regularization parameter $\lambda = 0.1$. We evaluated the model's goodness of fit for both periods. Figure 21 presents the diagnostic plots for the period after the All-Star break. The tight clustering of points along the $y = x$ line indicates high predictive accuracy, while the symmetric residual distribution centered at zero confirms that the model assumptions hold without significant systematic bias. Detailed diagnostic plots for the period before the All-Star break (Figure 22) exhibit similarly robust fit characteristics. The consistency of these diagnostic patterns across both temporal environments confirms that the SMART framework effectively captures the underlying data structure without overfitting to a specific period. Through sparse regularization on $\Delta\mathbf{B}$, the model identifies nine key nonzero rows (features) in the difference loading matrix (Figure 23), which we map to three distinct mechanisms driving late season gameplay transformations:

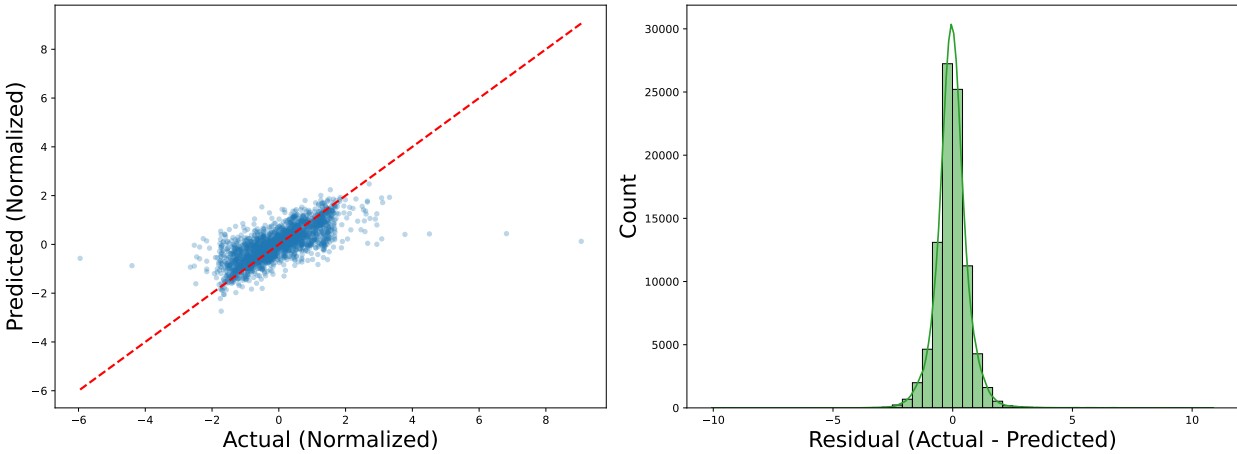

Figure 21: Model diagnostic plots for the period after the All-Star break. Left: Goodness of fit (predicted against actual values); Right: Residual distribution.

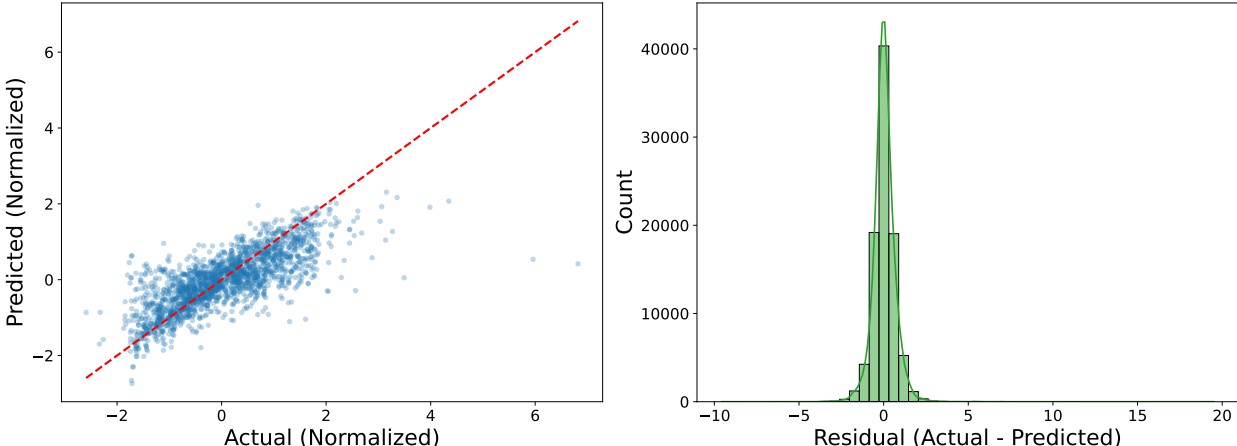

Figure 22: Model diagnostic plots for the NBA period before the All-Star break. Left: Goodness of fit (predicted against actual values); Right: Residual distribution.

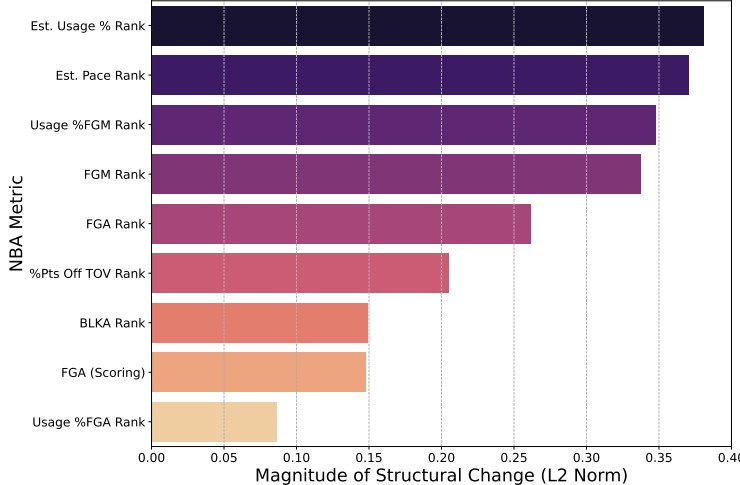

Figure 23: Non-zero features identified in the NBA dataset and their magnitudes of structural change.

Possession Distribution (Mechanism I): The structural shift is most heavily weighted towards offensive usage and volume metrics. Specifically, the model identifies *Est. Usage % Rank*, *Usage %FGM Rank*, and *Usage %FGA Rank*, alongside aggregate volume indicators like *FGM Rank*, *FGA Rank*, and *FGA (Scoring)*. The concentration of these six features confirms a consolidation of offensive responsibilities, consistent with usage

patterns centered on star players predicted by Tournament Theory Price et al. (2010): as teams vie for playoff positions, the marginal value of each possession increases, intensifying individual incentives.

Game Pace (Mechanism II): The second mechanism is characterized by *Est. Pace Rank* and *%Pts Off TOV Rank*. The prominence of *Est. Pace Rank* (the second-highest magnitude feature) quantifies the slowdown in gameplay. Concurrently, the shift in *%Pts Off TOV Rank* reflects the downstream effect of this slowdown: fewer possessions and tighter defensive transition discipline reduce the frequency of opportunistic transition scoring.

Physical Intensity (Mechanism III): The identification of *BLKA Rank* (Blocked Attempts) captures the escalation in defensive physicality. Unlike simple foul counts, which are subject to whistle variability, a structural shift in blocked shots indicates a tangible increase in shot contention and rim protection.

These latter two mechanisms, Pace and Physicality, exhibit strong concordance with established shifts in officiating standards. Empirical observations indicate a distinct variation in whistle frequency between the periods before and after the All-Star break. As noted by Laxdal et al. (2022), officiating logic functions as a primary exogenous variable that directly regulates both the speed of play (Mechanism II) and the permissible level of physical contact (Mechanism III). The ability of SMART to isolate these specific evolutions, which map directly to these institutional adjustments, strongly corroborates the empirical validity of our structural shift detection.

