# OpenReview forum: "SMART: A Modular Two-Stage Framework for Structural Representation Attribution"
_TMLR — Under review for TMLR_

### Review · Reviewer_nJev · 2026-06-08

**Summary Of Contributions:**

This paper introduces SMART (Sparse Mechanism Attribution for RepresenTation), a two-stage statistical framework designed to identify and localize structural shifts within a latent generative process across multiple environments. The authors started with a two-environment setting and then extended it to a multi-environment setting. The authors provided non-asymptotic error bounds for the estimated structures, and further introduced a consistent information criterion (IC) for rank selection.

**Audience:**

Yes

**Audience Explanation:**

The problem of learning from multiple environments, domain adaptation, and isolating localized mechanism adjustments is of high interest to the machine learning community. However, the paper must clarify its theoretical goals and fill major gaps in its technical guarantees before it can serve as a reliable reference for the community.

**Claims And Evidence:**

No

**Claims Explanation:**

While the authors provide initial simulations and numeric results, the overall claims regarding the necessity and purpose of the proposed regularization are mathematically and empirically disjointed:

1. Theoretical Mismatch (Theorem 3.3 vs Corollaries 3.4/3.5): If the primary goal of the theoretical tracking is to bound the errors of $F$ and $B$, the rates do not improve over simpler benchmarks. In fact, by column-stacking $M = [M^{(1)}, M^{(2)}]$ and minimizing the Frobenius norm directly under identical constraints, standard Davis-Kahan perturbations would completely omit the regularized inflation term $\sqrt{s\lambda}$ found in Theorem 3.3.

2. Empirical Disconnect (Table 2 / Subspace Recovery): The synthetic experiments verify that the proposed regularized framework does not outperform an unpenalized SVD projection in structural reconstruction fidelity. If the implicit purpose is instead to recover the sparse support profile of $\Delta B$, the paper fails to back this up with the appropriate corresponding mathematical evidence—namely, a strict proof of model selection consistency.

3. Information Criterion. First, why do you need a new IC, given that ICs already exist? In particular, does the term involving sparsity improve the selection of the dimensionality? Second, the $\tilde M^{(1)}$ and $\tilde M^{(2)}$ here are supposed to be rank-dependent. How are they obtained, under $r_{max}$?

**Requested Changes:**

1. Clarify the Core Objective & Theoretical Motivation: The authors must explicitly state what the primary goal of the SMART analysis is. If the goal is strictly the recovery of the support of $B^{(2)} - B^{(1)}$ (i.e., localizing the feature changes), the presentation should be pivoted around variable selection rather than matrix reconstruction rates.

2. Provide Model Selection Consistency Results: Given that the simulation emphasizes variable selection as the defining advantage over standard SVD projection, the authors should provide a rigorous theoretical proof demonstrating the support recovery consistency of $\Delta B$ (i.e., proving that $\mathbb{P}(\hat{\mathcal{S}} = \mathcal{S}) \to 1$ under standard coherence/irrepresentable-type conditions).

3. Address the Stacked-SVD Rate Comparison: Provide an explicit theoretical or discussion-based justification comparing the SMART rates to a simple baseline method where $M^{(1)}$ and $M^{(2)}$ are stacked by column ($M = [M^{(1)}, M^{(2)}]$) and minimized directly via Frobenius norm. Explain why the $\sqrt{s\lambda}$ term introduced in Theorem 3.3 is an acceptable or necessary trade-off for downstream tasks.

4. Justification and Clarification of the Information Criterion (IC): Provide a theoretical justification explaining why existing low-rank information criteria are insufficient, and demonstrate how incorporating the sparsity level ($\hat{s}$) explicitly improves the selection performance of the dimensionality $r$.

5. Explicitly clarify how the preliminary estimators $\tilde{M}^{(1)}$ and $\tilde{M}^{(2)}$ are obtained when selecting $r$. Since $\tilde{M}^{(k)}$ are supposed to be rank-dependent or act as regularizers, a circularity/bias issue arises if they are fixed prior to searching over $1 \le r \le r_{\max}$.

6. Fix Typographical Reference Errors: In Corollary 3.5 (and any corresponding locations in the multi-environment setting), correct the typo where it refers to "Theorem 3.4". Based on context, this should point to Corollary 3.4.

7. Extend Guarantees to Multi-Environment Settings: Ensure that the foundational corrections requested above (clarifying support recovery focus, addressing simpler stacking baselines, and detailing IC structures) are mirrored and resolved in the Multi-Environment extensions in Section 4.

---

> ### Author Response · Authors · 2026-07-21
> **Support recovery as the primary goal: new consistency theory, stacked SVD, and IC clarifications**
>
> We sincerely thank Reviewer nJev for the detailed review. Supplementary figures are available at the anonymous link: https://anonymous.4open.science/r/SMART2026-3DF2
>
> **[Q1] Primary goal.** The revision makes row support recovery, $\mathcal{S}^\ast=\lbrace j:\Vert\Delta\mathbf b_j^\ast\Vert_2>0\rbrace$, the primary goal. The abstract and introduction list localizing shifted features as the main contribution, the error bounds become supporting results, and the theory section centers on the new support theorem.
>
> **[Q2] Model selection consistency.** The revision adds a selection consistency theorem for the support $\hat{\mathcal{S}}$ of the estimated loading difference. With $\beta_{\min}=\min_{j\in\mathcal{S}^\ast}\Vert\Delta\mathbf b_j^\ast\Vert_2$, if $$\max(n\rho\_n,\ \sqrt{n}\mathrm{err}/\sigma\_B,\ ns/\sigma\_B^2)\ll\lambda\ll\min(n\beta\_{\min},\ (n\vee p)/s),$$ then $\Pr(\hat{\mathcal{S}}=\mathcal{S}^\ast)\to 1$, where the new quantity $\rho_{n,p}:=\max_j\Vert n^{-1}(\mathbf{F}^\ast)^\top\mathbf e_{\Delta,j}\Vert_2=O_p(\rho_n)$ controls the maximal projected noise. These conditions are natural. The minimal signal $\beta_{\min}$ makes truly shifted rows detectable, the projection term $\rho_n$ keeps noise from masking the signal, and the $\lambda$ window balances these two forces. The interval is provably nonempty under mild conditions, which closes the gap between rotation invariance of the support and its exact recovery.
>
> **[Q3] Stacked SVD and $\sqrt{s\lambda}$.** For low-rank reconstruction the truncated SVD is Frobenius optimal, so its bound lacks $\sqrt{s\lambda}$. The term comes from the sparsity regularization in our bound and is not a cost every support method must pay. When $s\lambda\ll n\vee p$ it is lower order, so SMART matches the first order reconstruction rate of the stacked SVD while gaining the localization guaranteed by the new support theorem. The revision keeps the stacked SVD as the reconstruction benchmark and adds a row-level support baseline (see our reply to Reviewer VGpA, Q3). A representative Poisson F1 figure is at the anonymous link.
>
> **[Q4] Information criterion.** Existing low-rank criteria charge only about $(n+p)r$ leading parameters, while SMART also estimates sparse loading differences, so under the $(\mathbf{B}^{(1)},\Delta\mathbf{B})$ parameterization the leading complexity is about $(n+p+s)r$, and ignoring the extra term underestimates it when the shift is nonnegligible. We use $IC(r)=\mathrm{Loss}(r)+r\eta(n,p)$, with consistency under the general condition $(\mathrm{err})^2+s\lambda\ll\eta(n,p)\ll n\sigma_B^2$ and the concrete choice $\eta_r=2(n+p+\hat s_r)g(n,p)$, $g\in\lbrace\log\frac{np}{n+p},\log(n\wedge p)\rbrace$. The $\hat s_r$ term charges the sparse shift degrees of freedom, so that when they are nonnegligible relative to $n+p$ the criterion is not underpenalized and rank selection stays stable. The revision separates the general condition from this concrete choice.
>
> **[Q5] Stage I and circularity.** Stage I denoising imposes no low-rank constraint, so $\tilde{\mathbf{M}}^{(k)}$ does not depend on the candidate rank and there is no circularity. Simulations compute Stage I once with the true factor number. Real data run it once with a preset $r_{\max}$ and fix it, and only Stage II is refitted for each candidate $r$. Chen and Li (2022, Theorem 1) gives uniform error control over $r^\ast\le r\le r_{\max}$, and Fan et al. (2013) shows that mild overestimation is more admissible than underestimation.
>
> **[Q6] Cross reference error.** We corrected this and checked all theorem and corollary labels and cross references, and all corollaries now display correctly.
>
> **[Q7] Multi-environment extension.** For the multi-environment case the target is the full sequence of shifts. With $\Delta\mathbf{B}^{(k)\ast}=\mathbf{B}^{(k)\ast}-\mathbf{B}^{(k-1)\ast}$, we define $\mathcal{A}^\ast=\lbrace(k,j):2\le k\le K,\ \Vert\Delta\mathbf b_j^{(k)\ast}\Vert_2>0\rbrace$, which records both the shifted features and the environments where they shift, rather than only their union. We estimate the corresponding support $\hat{\mathcal{A}}$ and prove that, for fixed $K$ under conditions analogous to the two-environment case (with $s_K=\vert\mathcal{A}^\ast\vert$ and $\beta_{\min,K}=\min_{(k,j)\in\mathcal{A}^\ast}\Vert\Delta\mathbf b_j^{(k)\ast}\Vert_2$), $\Pr(\hat{\mathcal{A}}=\mathcal{A}^\ast)\to 1$. The multi-environment criterion uses leading complexity $(n+p+\hat s_{K,r})r$, and Section 4 will be updated accordingly.
>
> We hope these clarifications address the reviewer's concerns.
>
> References:
>
> [1] Chen, Y. and Li, X. (2022). Determining the number of factors in high-dimensional generalized latent factor models. Biometrika.
>
> [2] Fan, J., Liao, Y. and Mincheva, M. (2013). Large covariance estimation by thresholding principal orthogonal complements. JRSS-B.

---

> > ### Comment · Reviewer_nJev · 2026-07-21
> >
> > I thank the authors for the revision. I think all the comments have been addressed.

---

> > > ### Author Response · Authors · 2026-07-21
> > >
> > > We sincerely thank you for the time and effort devoted to our paper, and are glad to hear that all the comments have been addressed. Your constructive feedback has substantially strengthened the paper.

---

### Review · Reviewer_GU9D · 2026-06-30

**Summary Of Contributions:**

This paper studies latent signal recovery in multi-environment via representation learning. This paper proposes SMART, a two-stage framework with theoretical guarantee for structural representation attribution. Specifically, the first stage performs signal recovery and the second stage learns a shared representation and attributes mechanism shifts across environments through sparse changes in the loading matrix. The authors provide theoretical results for the estimation errors (convergence rate) and a consistent information criterion to determine the latent dimensionality. Experimental results show the effectiveness of the proposed method.

**Audience:**

Yes

**Audience Explanation:**

Representation learning is an important issue that the audience of TMLR would be interested in.

**Claims And Evidence:**

Yes

**Claims Explanation:**

The theoretical results are generally comprehensive, as well as the experimental results.

**Requested Changes:**

### Weakness/Question

1. SMART appears to have a strong connection to existing identifiable causal representation learning (CRL) methods [a, b]? Typically, CRL assumes an invariant mechanism from latent to observations $x$, i.e. $p(x|z)$, and an environment-dependent latent distribution $p(z|u)$, where identifiability comes from the sufficient variability of the environment $u$. In SMART, however, the sample-level latent representation $F$ is assumed invariant while the loading matrix $B$, which is analogous to $p(x|z)$ in CRL, is assumed variant across environments.  The assumed variant/invariant structures seem to be almost opposite, and the paper should explain clearly why its invariance assumption is appropriate for the intended applications.
2. The linear factorization $FB^{T}$ and the shared $F$ assumptions are somewhat strong. Could the author clarify that?
3. The one-step estimator is only developed for additive noise models. Although the least-square loss for non-additive observations may be inappropriate, replacing the first-stage estimate with the original observation is still natural. Could the author clarify whether a distribution-specific one-step likelihood formulation is possible for other distributions such as Poisson and Binary?
4. The theoretical results show the second-stage error inherits the first-stage signal recovery error. It may suggest that the final support recovery could be sensitive to the first-stage estimator, including model misspecification, representation instability, and rank selection error. It would be better to conduct more experiments comparing different first-stage estimators.
5. The used baselines are relatively simple. The authors should compare against stronger and more directly related methods, such as multi-environment factor analysis or identifiable causal representation learning methods, if applicable.


[a] Khemakhem, I., Kingma, D., Monti, R., & Hyvarinen, A. (2020, June). Variational autoencoders and nonlinear ica: A unifying framework. In *International conference on artificial intelligence and statistics* (pp. 2207-2217). PMLR.

[b] Li, Z., Cai, R., Chen, G., Sun, B., Hao, Z., & Zhang, K. (2023). Subspace identification for multi-source domain adaptation. *Advances in Neural Information Processing Systems*, *36*, 34504-34518.

---

> ### Author Response · Authors · 2026-07-21
> **Relation to identifiable CRL, modeling assumptions, likelihood extensions, and baselines**
>
> We sincerely thank Reviewer GU9D for the thoughtful review. Supplementary figures are available at the anonymous link: https://anonymous.4open.science/r/SMART2026-3DF2
>
> **[Q1] Relation to identifiable CRL.** The two frameworks place change and invariance in opposite parts of the model. Identifiable CRL (Khemakhem et al., 2020; Li et al., 2023) assumes an invariant mechanism $p(x\mid z)$ and identifies latent variables or subspaces from sufficient variation of $p(z\mid u)$. SMART studies a complementary problem: for paired or aligned samples, $\mathbf{M}^{(u)}=\mathbf{F}(\mathbf{B}^{(u)})^\top$ uses a shared $\mathbf{F}$ as a common coordinate system and localizes the nonzero rows of $\mathbf{B}^{(u)}-\mathbf{B}^{(v)}$, attributing observed changes to specific factor-to-observation mechanisms.
>
> The shared $\mathbf{F}$ makes this comparison meaningful: separate $\mathbf{F}^{(u)}$ would confound coordinate changes with loading changes, whereas a shared $\mathbf{F}$ leaves only a single global transform that preserves the zero row set.
>
> This matches applications where aligned samples are measured across conditions, the latent profile stays comparable, and its expression on a few features changes. When environmental change acts mainly on $p(\mathbf{F}\mid u)$, standard identifiable CRL is the natural tool, and SMART does not claim to cover that case.
>
> Because CRL solves the harder latent identification problem, its guarantees require the number of auxiliary values or domains to grow with the latent dimension, whereas SMART targets the rotation-invariant row support and needs only two paired environments. The revision replaces "latent generative process" with "loading structures".
>
> **[Q2] Shared $\mathbf{F}$ and the linear low-rank assumption.** SMART targets matrix data with paired or aligned samples, where $\mathbf{F}$ captures cross-environment heterogeneity and $\mathbf{B}^{(u)}$ its environment-specific association with features. With aligned samples the shared $\mathbf{F}$ is empirically checkable when the samples donot change. The linear decomposition approximates the Stage I parameter matrix and keeps each loading row tied to one feature. It does not require Gaussian observations, as Example 2 handles Gaussian, Poisson, and Binary data via exponential family likelihoods on the natural parameter matrix.
>
> **[Q3] One-step extension to Poisson or Binary likelihoods.** Replacing the Stage I estimate by raw observations is feasible, using the Poisson or Bernoulli negative log likelihood with the fused group penalty. But our one-step theory relies on additive noise and squared loss, so coupling a complex likelihood with the structural penalty requires redesigning both optimization and theory. The modular two-stage design decouples distribution-specific recovery from attribution, and we list this extension as future work.
>
> **[Q4] Sensitivity to Stage I.** The final accuracy inherits Stage I quality. Our bounds and the new support theorem keep the Stage I error $\mathrm{err}$ explicit (e.g. the $\sqrt{n}\mathrm{err}/\sigma_B$ term), so any change in Stage I propagates to support recovery at the order of $\mathrm{err}$, and better Stage I estimators plug in without changing Stage II. For rank selection, real data run Stage I once with a preset $r_{\max}$, for which Chen and Li (2022, Theorem 1) gives uniform error control over $r^\ast\le r\le r_{\max}$ and Fan et al. (2013) shows mild overestimation is admissible. Simulations use the true factor number, so comparisons isolate Stage II.
>
> **[Q5] Multi-environment factor analysis and CRL baselines.** The added baseline obtains a common low-rank estimate of the denoised matrices, a factor analytic estimate, then ranks $\Vert\hat b_j^{(2)}-\hat b_j^{(1)}\Vert_2$ with a unified non-oracle rule (see our reply to Reviewer VGpA, Q3). It shares SMART's input, rank, and metrics, and represents the multi-environment factor analysis route for this target. We did not include CRL methods in the support F1 comparison, because their inputs are samples with auxiliary variables or unpaired multi-domain data, their identification rests on an invariant observation map (the part SMART lets change), and their outputs are latent variables or subspaces, not a row support like $\mathcal{S}^\ast$.
>
> We hope these clarifications address the reviewer's concerns.
>
> References:
>
> [1] Khemakhem, I. et al. (2020). Variational autoencoders and nonlinear ICA: a unifying framework. AISTATS.
>
> [2] Li, Z. et al. (2023). Subspace identification for multi-source domain adaptation. NeurIPS.
>
> [3] Chen, Y. and Li, X. (2022). Determining the number of factors in high-dimensional generalized latent factor models. Biometrika.
>
> [4] Fan, J., Liao, Y. and Mincheva, M. (2013). Large covariance estimation by thresholding principal orthogonal complements. JRSS-B.

---

### Review · Reviewer_VGpA · 2026-07-09

**Summary Of Contributions:**

This paper proposes SMART, a two-stage method for identifying which features drive a distributional shift across two or more environments whose observations are given as matrices. Stage I denoises each environment's observation matrix $\mathbf{M}_1, \mathbf{M}_2$ into a signal estimate using any representation learner. Stage II fits each $\mathbf{M}$ using a latent matrix $\mathbf{F}$, assumed invariant across environments, together with environment-specific loading matrices $\mathbf{B}_1, \mathbf{B}_2$, and applies a row-sparse group penalty to the loading difference matrix $\Delta \mathbf{B} = \mathbf{B}_2 - \mathbf{B}_1$. Hence $\mathbf{M}_k$ are both approximated as low-rank matrices $\mathbf{F}\mathbf{B}_k$, with the nonzero rows of $\Delta \mathbf{B}$ identifying the shifted features. Because the row support of $\Delta \mathbf{B}$ is invariant under the rotational ambiguity of latent factor models, the attribution target is well-defined even though the loading matrices themselves are not identifiable. The authors prove an estimation error rate for Stage II dependent on the error from Stage I, and also provide a consistent information criterion for selecting the latent dimension. Synthetic experiments comparing SMART against its $\ell_1$ penalized version and unpenalized SVD, together with two real-data applications, illustrate the method.

**Strengths:** The problem is well motivated. The core idea is sound, and the proofs seem to be correct. I also appreciate the authors for providing a method to infer the latent dimension, which makes the proposed method highly practical.

**Weaknesses:** The theory only establishes $\ell_2$ error bounds on the estimator but does not establish row support recovery (attribution) guarantees, which is an important property the method is disclosed to deliver. Also, the experiments evaluating row support recovery compares with the Lasso variant of SMART, which cannot deliver high F1 score by design; I am not entirely convinced that this is the correct choice as the baseline.

**Audience:**

Yes

**Audience Explanation:**

Transfer-learning techniques in statistical methods have recently drawn attention in the ML and stat community. While previous research have focused on estimating vectors from different environments and their differences with sparse profiles, this research contributes by proposing what can be thought of as its matrix variant, and furthermore presenting both theory and experimental results of their algorithm. I believe readers interested in either the methodology or its theory would find it relevant.

**Claims And Evidence:**

No

**Claims Explanation:**

The claims are only addressed partially.

1.  I am uncertain how the theory demonstrates the effectiveness of the proposed method. The main effect of the row-sparse penalty is to identify the support of $\Delta \mathbf{B}$, yet the paper provides no theoretical guarantee on row support recovery. Moreover, the Stage II error rates in Theorems 3.2 and 3.3 are minimized at $\lambda = 0$, so the analyzed bound becomes strictly worse for any positive penalty (even for $\lambda=0$, the theory establishes that the Stage II error cannot be worse than Stage I in terms of its rate, but not on the constants). It therefore shows only that regularization does not degrade the $\ell_2$​ estimation rate in a weak sense, with no analysis of its intended benefit (support recovery).  Every benefit of the row-sparse penalty over the unpenalized estimator is demonstrated only empirically.

2.  The N-Lasso comparison in terms of F1 score may be biased toward SMART. The F1 score is computed on the row support, but N-Lasso applies an entrywise $\ell_1$​ penalty, which zeros individual entries rather than rows. It can eliminate a row only by independently zeroing **all** of its entries, which should be very difficult for N-Lasso. The metric therefore rewards row-level sparsity that the baseline's penalty is not designed to produce. I believe a proper comparison would be with a method that is actually aimed at row selection.

3.  The paper describes SMART as a "unified framework," but it is not clear what is being unified. From my understanding, the method addresses a specific problem of estimating the row-wise differences between loading matrices from two (or more) related environments, under a shared latent representation. The generality suggested by "unified framework" is not reflected in the method.

**Requested Changes:**

Critical points:

C1.  Please clarify the interpretation of the given guarantees; they do not show that Stage II estimates anything more accurately than Stage I, yet not addressing the accuracy of the row support of $\Delta \mathbf{B}$, which is the inferential target. Could the authors explain what they are meant to demonstrate?

C2.  The condition $s\lambda \ll n \vee p$ in Corollary 3.4 controls the penalty's contribution to the $\ell_2$ error and forces it to be asymptotically negligible. However, a sparse row support is produced by the group-Lasso step (Step 2 of subsection 3.2.1), where a null row is zeroed only when the factor-residual correlation falls below $\lambda$ according to the KKT conditions. This should impose a separate lower bound requirement on $\lambda$. Could the authors state the condition under which rows are zeroed, and confirm it is compatible with $s\lambda \ll n \vee p$? This would clarify whether the analyzed regime is one in which the penalty is actually in play.

C3.  Please add a baseline that selects at the row level, so the comparison is on the same footing as the metric. One natural choice, I believe, is to rank rows by $\|\|\mathbf{b}_j^{(2)} - \mathbf{b}_j^{(1)}\|\|_2$ using results from the unpenalized SVD estimator, selecting the top $k$ rows with $k$ decided by, say, the elbow method in Appendix D.

C4.  Please clarify what "unified framework" refers to.

C5.  In Appendix C.6, it is stated that "The empirical curves show clear monotone scaling trends, consistent with the rate statements in Theorems 3.2 and 3.3," yet the figure appears to show a plateaued error on $\Delta \mathbf{B}$. Please address this, and clarify whether a slower rate for $\Delta \mathbf{B}$ relative to the other quantities is to be expected.

Other requested changes :
1.  Page 10 "We further validate two aspects raised by the reviewer feedback" (and similar phrasing referring to reviewer feedback / revision experiments, e.g. Table 27) appears to be leftover from a previous review round and should be removed.
2.  In the Appendix, all Corollaries (3.4, 3.5, 3.7) in the main text are referred to as Theorems and should be relabelled correctly.
3.  While I appreciate Tables 1 and 2 being exhaustive across several experimental settings, I find them a little difficult to compare by eye. A summary plot would help, for instance F1 against $p$ with one line per method and error bands, with separate panels for $s$ and $r$ (This is only a suggestion and I leave it to the authors' discretion).

---

> ### Author Response · Authors · 2026-07-21
> **New support selection consistency theory, explicit range for lambda, and a row-level SVD baseline**
>
> We sincerely thank Reviewer VGpA for the careful and constructive review. Supplementary figures are available at the anonymous link: https://anonymous.4open.science/r/SMART2026-3DF2
>
> **[Q1] Guarantees, support recovery, and $\lambda=0$.** The primary goal of SMART is not only to estimate the shared low-rank structure but to recover the row support $\mathcal{S}^\ast=\lbrace j:\Vert\Delta\mathbf b_j^\ast\Vert_2>0\rbrace$. Beyond the original error bounds, the revision adds a support selection consistency theorem showing that the row support $\hat{\mathcal{S}}$ of the estimated loading difference satisfies $\Pr(\hat{\mathcal{S}}=\mathcal{S}^\ast)\to 1$ under a minimal signal condition and the requirement on $\lambda$ in Q2. This upgrades the guarantee from overall error control to exact support recovery.
>
> On $\lambda=0$: the observation is correct. For Frobenius low-rank reconstruction the unpenalized SVD is optimal, and a positive $\lambda$ worsens the constant in our bound. However, when $s\lambda\ll n\vee p$ the extra term $\sqrt{s\lambda}$ is lower order, so regularization does not sacrifice the rate. This modest cost buys the ability the SVD lacks, namely attributing the shift to specific features with a consistent support guarantee. We do not claim SMART beats the SVD in pure reconstruction. The revision evaluates reconstruction and support recovery separately.
>
> **[Q2] Bounds for $\lambda$.** The lower bound removes unchanged rows and prevents false positives, and the upper bound keeps truly shifted rows and preserves the error bounds. The new theorem requires $$\max(n\rho\_n,\ \sqrt{n}\mathrm{err}/\sigma\_B,\ ns/\sigma\_B^2)\ll\lambda\ll\min(n\beta\_{\min},\ (n\vee p)/s),$$ where $\beta_{\min}=\min_{j\in\mathcal{S}^\ast}\Vert\Delta\mathbf b_j^\ast\Vert_2$, $\mathrm{err}$ is the Stage I Frobenius error, and $\rho_{n,p}:=\max_j\Vert n^{-1}(\mathbf{F}^\ast)^\top\mathbf e_{\Delta,j}\Vert_2=O_p(\rho_n)$ controls the maximal projection of the noise difference onto the true factor space. The other two terms control the leakage of the Stage I error and of the sparsity penalty. The CV choice in experiments is a practical surrogate. The interval is provably nonempty under mild conditions. A new corollary shows that with the refined GFM estimator of Chen and Li (2024) in Stage I ($\sigma_B\asymp\sqrt{p}$, no missing data, $r=r^\ast$), the requirement simplifies to $n(\log np)^2/\sqrt{n\wedge p}\ll\lambda\ll\min(n\beta_{\min},(n\vee p)/s)$, compatible with $s\lambda\ll n\vee p$.
>
> **[Q3] A fair row-level baseline.** We agree, and added a row-level selection baseline built on the unpenalized SVD. Let $d_j=\Vert\hat b_j^{(2)}-\hat b_j^{(1)}\Vert_2$, sorted as $d_{(1)}\ge\cdots\ge d_{(p)}$, and select the top $\hat s_{\mathrm{diff}}$ variables, where $\hat s_{\mathrm{diff}}=\min\arg\max_{1\le k\le p-1}(d_{(k)}-d_{(k+1)})$ is an elbow rule on successive gaps, with $\hat s_{\mathrm{diff}}=0$ if $d_{(1)}\le 10^{-5}$. All methods share the same Stage I input, rank, and row-level metrics. On Poisson data SMART attains the highest or tied highest mean F1 in almost all settings with the narrowest error bands. The SVD baseline is competitive for small $s$, but its recall and F1 degrade markedly as $s$ grows, while its precision stays near 1. Gaussian and Binary data yield similar results, reported in the revision. A representative Poisson F1 figure is at the anonymous link.
>
> **[Q4] "Unified framework".** We agree the term was too strong and now use "modular two-stage framework". Stage I accommodates heterogeneous data types, Stage II performs structural attribution under the shared representation, and Theorem 3.2 links them through error inheritance. All occurrences are replaced.
>
> **[Q5] The $\Delta\mathbf{B}$ error plateau. **By Theorem 3.3, under our data generating process the row-normalized error satisfies $s^{-1/2}\Vert\hat{\Delta\mathbf{B}}\mathbf{O}-\Delta\mathbf{B}^\ast\Vert_F=O_p(1)$. This measures the error relative to the per-row signal magnitude, which is of constant order, so it does not decrease with $n$ or $p$, matching the flat curves. Under subspace ($\sqrt{p}$) normalization the error does converge, and the normalized errors of $\hat{\mathbf{M}}$, $\hat{\mathbf{F}}$, $\hat{\mathbf{B}}$ decrease monotonically. The revision removes the monotonicity claim and annotates each order.
>
> **[Q6] Other edits.** We removed leftover revision language and an outdated section reference, fixed the cross references that displayed Corollaries 3.4/3.5/3.7 as "Theorem", and replaced the large tables with summary figures reporting means and one standard deviation bands over 200 runs. We provide one representative illustration through the anonymous link, while the other figures are displayed in the revision document.
>
> We hope these clarifications address the reviewer's concerns.
>
> References:
>
> [1] Chen, Y. and Li, X. (2024). A note on entrywise consistency for mixed-data matrix completion. JMLR.